# Interventional Processes For Causal Uncertainty Quantification

Hugh Dance [1]  Peter Orbanz [1]  Arthur Gretton [1]

## Abstract

Reliable uncertainty quantification for causal effects is crucial in high-stakes applications, but remains challenging when the target is an entire function rather than a scalar estimand. In this work, we introduce a GP-based approach for uncertainty quantification of interventional functions. The central idea is to build on recent work representing *interventional* functions as an inner-product of *observational* functions in a reproducing kernel Hilbert space (RKHS), by constructing appropriate GP priors for such functions and inferring posteriors from observational data. Our approach yields closed-form posterior moments and tractable training and inference, while avoiding pathologies of previous GP prior constructions for RKHS functions. We further derive a practical procedure for posterior coverage calibration. Across synthetic benchmarks, causal Bayesian optimization tasks, and a large-scale real dataset, our method improves uncertainty quantification while remaining competitive in causal effect estimation.

## 1. Introduction

Decisions in healthcare, economics, and public policy often hinge on estimating the effects of interventions from observational data. In such high-stakes settings, point estimates alone are often insufficient: decision-makers need reliable uncertainty estimates to assess when causal conclusions may be unstable or weakly supported.

For scalar average treatment effects with discrete treatments, doubly robust and debiased estimators provide a principled route to asymptotically valid confidence intervals (Hines et al., 2022; Kennedy, 2024; Chernozhukov et al., 2018). For function-valued causal targets, such as conditional average treatment effects and continuous-treatment dose-response

functions, the situation is more delicate. Pointwise evaluation of such functions is generally not a bounded linear functional in the usual $L^2$ sense. As a consequence, the Riesz representers used in debiased inference need not exist without additional structure (see Sec 6 in (Singh et al., 2024)). Existing frequentist approaches therefore use parametric, smoothness, or functional-form assumptions (Kennedy et al., 2017; Oprescu et al., 2019; Semenova & Chernozhukov, 2021; Kennedy, 2023). These approaches are also known to be sensitive to weak overlap in finite samples. Taking a Bayesian approach is also difficult, since placing priors on the conditional laws which identify interventional functions can make posterior inference computationally intractable.

Recently, Singh et al. (2024) showed that many causal functions can be represented using functions in a reproducing kernel Hilbert space (RKHS). These RKHS functions can be estimated from observational data using non-parametric regression, avoiding full density estimation or propensity weighting while yielding estimators with strong theoretical guarantees. However, they do not address the problem of uncertainty quantification around such estimators. By reducing causal function estimation to nonparametric regression, this approach suggests a natural route to functional uncertainty quantification via Gaussian processes.

Gaussian processes (GPs) are a standard tool for functional uncertainty quantification in machine learning (Williams & Rasmussen, 2006) and are closely connected to RKHSs (Kanagawa et al., 2018). A natural idea is therefore to place GP priors on the RKHS functions and infer posteriors on the interventional function using observational data. However, constructing GP priors for functions in a prescribed RKHS is known to be nontrivial (Lukić & Beder, 2001). Previous work has attempted to do so using bespoke covariance constructions (Flaxman et al., 2016; Chau et al., 2021b). Unfortunately, the resulting kernels are only closed-form in special cases, and we later show they can induce *underfitting* of causal functions (see Figure 3), and *variance collapse* outside the training support (see Figure 4).

**Our Contributions** In this work, we introduce a novel GP-based approach for quantifying uncertainty over causal functions via RKHS representations. To avoid the pathologies of earlier GP prior constructions for RKHS functions, we expand the relevant function classes and use spectral

---

[1]Gatsby Unit, University College London, London, United Kingdom. Correspondence to: Hugh Dance <uct-phwd@ucl.ac.uk>.

*Proceedings of the 43rd International Conference on Machine Learning*, Seoul, South Korea. PMLR 306, 2026. Copyright 2026 by the author(s).

representations of RKHS elements, allowing standard GP priors to be used. Crucially, the posterior construction is chosen so that its posterior mean recovers the original kernel causal estimator in Singh et al. (2024), thereby targeting uncertainty around this established point estimate. Our approach yields closed-form posterior moments, tractable moment-matched credible regions, and practical algorithms for hyperparameter training and posterior calibration. Empirically, our method improves uncertainty quantification across synthetic benchmarks, causal Bayesian optimization tasks, and a large-scale real dataset, while remaining competitive in causal effect estimation.

## 2. Background

### 2.1. Causal Effects as Interventional Functions

We study the estimation of causal effects of a possibly continuous treatment $A \in \mathbb{R}$ on an outcome $Y \in \mathbb{R}$, and *epistemic uncertainty* in these effects. We adopt the framework of causal graphical models (Pearl, 2009a), where causal relations are represented by a directed acyclic graph (DAG) over observed variables $O \in \mathcal{O}$ and latent variables $U \in \mathcal{U}$. Two examples are shown in Figure 1. In the observational distribution, each variable is generated from its conditional distribution given its parents in the graph. A do-intervention, denoted $\mathrm{do}(A = a)$, simulates the effect of fixing $A = a$ by changing the conditional distribution for $A$ to a point mass at $a$ (cutting all incoming edges to $A$). The induced interventional distribution of outcomes is $\mathbb{P}(Y \mid \mathrm{do}(A = a))$.

The effects we consider include average treatment effects $\mathrm{ATE}(a) = \mathbb{E}[Y \mid \mathrm{do}(A = a)]$, conditional average treatment effects $\mathrm{CATE}(a, z) = \mathbb{E}[Y \mid \mathrm{do}(A = a), Z = z]$ (where $Z$ is a pre-treatment covariate), and effects on the treated $\mathrm{ATT}(a, a') = \mathbb{E}[Y \mid \mathrm{do}(A = a), A = a']$. Under standard graphical identification criteria, such as the back-door and front-door criteria reviewed in Appendix A.2, these quantities can be written in the form

$$\gamma(w, z) = \int_{\mathcal{V}} \mathbb{E}[Y \mid W = w, V = v] \, \mathbb{P}_{V \mid Z}(dv \mid z), \quad (1)$$

where $W, V, Z$ are (possibly empty) subsets of the observed variables $O$, that depend on the estimand and identifying formula. For example, in the graph in Figure 1 left, the conditional average effect of $A$ on $Y$ given $C$ is given under the back-door criterion:

$$\mathrm{CATE}(a, c) = \int \mathbb{E}[Y \mid A = a, B = b, C = c] \mathbb{P}_{B \mid C}(db \mid c),$$

in which case we set $W = (A, C)$, $V = B$, and $Z = C$. Similarly, in the graph in Figure 1 right, the average effect of setting $A = a$ for units given $A = a'$ is given by the front-door criterion:

$$\mathrm{ATT}(a, a') = \int \mathbb{E}[Y \mid A = a', M = m] \, \mathbb{P}_{M \mid A}(dm \mid a),$$

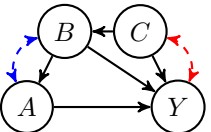
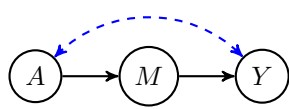

*Figure 1.* Left: causal graph where $O = (A, B, C, Y)$ and $(B, C)$ satisfies the back-door criterion w.r.t. $(A, Y)$. Right: Causal graph where $O = (A, M, Y)$ and $M$ satisfies the front-door criterion w.r.t. $(A, Y)$. Definitions of these criteria are in Appendix A.2. Dashed edges = effect of unobserved confounders.

in which case we set $W = A$, $V = M$, and $Z = A$.

We therefore take $\gamma$ in (1) as the central object of analysis, and refer to it as an *interventional* or *causal* function.

### 2.2. Kernel Methods for Interventional Functions

Recently, Singh et al. (2024) reduced the problem of estimating (1) to non-parametric regression. The idea is to represent the function $(w, v) \mapsto \mathbb{E}[Y \mid W = w, V = v]$ using an element $f$ of the reproducing kernel Hilbert space (RKHS) $\mathcal{H} := \mathcal{H}_W \otimes \mathcal{H}_V$ (Steinwart & Christmann, 2008):

$$\mathbb{E}[Y \mid W = w, V = v] = \langle f, \psi_W(w) \otimes \psi_V(v) \rangle_{\mathcal{H}}. \quad (2)$$

Here $\psi_W : \mathcal{W} \to \mathcal{H}_W$ and $\psi_V : \mathcal{V} \to \mathcal{H}_V$ are feature maps associated with the positive definite, bounded kernels $k_W : \mathcal{W}^2 \to \mathbb{R}$, $k_V : \mathcal{V}^2 \to \mathbb{R}$, via $k_W(w, w') = \langle \psi_W(w), \psi_W(w') \rangle_{\mathcal{H}_W}$. Substituting (2) into (1) gives the following representation of the causal function

$$\gamma(w, z) = \langle f, \psi_W(w) \otimes \mu(z) \rangle_{\mathcal{H}} \quad (3)$$

where

$$\mu : z \mapsto \mathbb{E}[\psi(V) \mid Z = z]$$

is an embedding of $\mathbb{P}_{V \mid Z}$ in $\mathcal{H}_V$, under the usual assumption that the kernels are *characteristic* (Park & Muandet, 2020).

Given $n$ observations $\mathcal{X}^n := \{(Y_i, V_i, W_i, Z_i)\}_{i=1}^n$, one can estimate $f$ and $\mu$ using scalar-valued and vector-valued kernel ridge regressions respectively (Grünewälder et al., 2012), which results in a closed form estimator for $\gamma$,

$$\hat{\gamma}(w, z) = \boldsymbol{\beta}(z)^\top K_V \boldsymbol{\alpha}(w) \quad (4)$$

Here $\boldsymbol{\beta}(z), \boldsymbol{\alpha}(w) \in \mathbb{R}^n$ are kernel regression weights and $K_V$ is the gram matrix of kernel evaluations with $(i, j)$ entry $k_V(V_i, V_j)$. This approach is fully non-parametric but avoids the need to estimate $\mathbb{P}_{V \mid Z}$ or use propensity weighting, therefore side-stepping the finite-sample instabilities of such methods (Li et al., 2023). It also benefits from minimax-optimal rates for $f$ and $\mu$ (Fischer & Steinwart, 2020; Li et al., 2024), and has yielded state-of-the-art performance under continuous treatments (Singh et al., 2024).

### 2.3. GPs for Uncertainty Quantification

A Gaussian process (GP) is a collection of random variables $(Y(x))_{x \in \mathcal{X}}$ such that any finite subcollection is jointly Gaus-

sian (Kallenberg, 1997). We call $m : x \mapsto \mathbb{E}Y(x)$ the mean function and $k : (x, x') \mapsto \mathbb{C}ov(Y(x), Y(x'))$ the covariance function of the GP and write $Y \sim \mathcal{GP}(m, k)$. GPs are a popular tool for modeling functional uncertainty in machine learning, offering good generalization properties with closed-form training and inference in many settings (Williams & Rasmussen, 2006).

Since, under the model (3), estimating the causal function $\gamma$ can be reduced to estimating the regression functions

$$(w, v) \mapsto \langle f, \psi_W(w) \otimes \psi_V(v) \rangle \quad \text{and} \quad z \mapsto \mu(z),$$

corresponding to the observational conditional expectations $\mathbb{E}[Y|W, V]$ and $\mathbb{E}[\psi_V(V)|Z]$, a natural idea to derive uncertainty estimates around the estimator (4) is to model these functions using GPs. However, constructing such priors is non-trivial. The sample paths of a GP with kernel $k_W \otimes k_V$ are almost surely too rough to admit an RKHS representation of the form $\langle f, \psi_W(\bullet) \otimes \psi_V(\bullet) \rangle$, despite being defined from the same kernels (Kanagawa et al., 2018). Moreover, $\mu$ is an infinite-dimensional *RKHS-valued* function rather than a scalar function, so standard GP priors do not apply.

# 3. From Interventional Functions to Interventional Processes

We now turn the interventional function $\gamma$ into an *interventional process*: a stochastic process over intervention and conditioning values induced by priors on the regression objects $f$ and $\mu$. Rather than enforcing the original RKHS constraints on these objects directly, we enlarge the relevant function classes and use spectral representations of RKHS elements. This allows standard GP priors to be used while preserving the causal representation of $\gamma$. As shown in Section 4, this construction also precisely ensures that the posterior mean later recovers the kernel estimator in (4).

## 3.1. Relaxing the RKHS Constraint for the $f$ Prior

The model for $\mathbb{E}[Y|W, V]$ in (2) can be expressed as a linear functional $f : \mathcal{H}_W \otimes \mathcal{H}_V \to \mathbb{R}$ of the features of $(W, V)$:
$$\mathbb{E}[Y|W, V] = f(\psi_W(W) \otimes \psi_V(V)).$$

The RKHS constraint in (2) enforces that $f$ is a *bounded* functional. Rather than constructing a bespoke GP prior to enforce this constraint, we instead place a simple linear-kernel GP prior directly on $f$:
$$f \sim \mathcal{GP}(0, \mathcal{K}), \quad \mathcal{K}(h, h') = \langle h, h' \rangle_{\mathcal{H}_W \otimes \mathcal{H}_V}. \quad (5)$$

Here $h, h'$ are generic elements of $\mathcal{H}_W \otimes \mathcal{H}_V$. Sample paths of such a GP are almost surely *not* bounded functionals (Dudley, 2010), meaning the prior effectively enlarges the function class. A key benefit of this relaxation is it enables us to derive closed-form posteriors on $f$ with standard GP machinery. In particular, the induced prior on the observational conditional expectation is simply

$$\mathbb{E}[Y \mid W = \bullet, V = \bullet] \sim \mathcal{GP}(0, k_W \otimes k_V),$$

where $k_W$ and $k_V$ are the kernels associated with $\psi_W, \psi_V$. When coupled with a Gaussian noise model for $Y$ (see Section 4), the posterior for $f$ is also a GP, and its mean will coincide exactly with the kernel regression estimator for $f$ in Singh et al. (2024). This reflects the general equivalence between GP posterior means and kernel estimators (Kanagawa et al., 2018).

## 3.2. Using the Spectral Representation for the $\mu$ Prior

We next construct GP priors for $\mu = \mathbb{E}[\psi_V(V) \mid Z = \bullet]$. Since this function takes values in an infinite-dimensional RKHS, our key idea is to use a *spectral representation* of the feature map. This reduces $\mu$ to a sequence of scalar-valued coordinates, on which we can place standard GP priors without constraint.

In particular, by Mercer's theorem (Sun, 2005), if the kernel $k_V$ is continuous and bounded and $\mathcal{V}$ is $\sigma$-compact[1], the feature map $\psi_V$ admits the expansion

$$\psi_V(v) = k_V(\bullet, v) = \sum_{i=1}^{\infty} \lambda_{V,i} \, \phi_{V,i}(v) \, \phi_{V,i}(\bullet).$$

with convergence absolute and uniform on compact sets. Here $(\lambda_{V,i}, \phi_{V,i})$ are eigenpairs of the operator

$$(T_{k_V, \nu} g)(v) = \int k_V(v, v') g(v') \, d\nu(v'),$$

and $\nu$ is a full-support probability measure on $\mathcal{V}$, which we call the *spectral* measure. Note that the operator and eigensystem depend on $\nu$. This expansion lets us reduce $\mu$ to a collection of *scalar-valued* conditional expectations:

$$\underbrace{\mathbb{E}[\psi_V(V)|Z]}_{\mu(Z)} = \sum_{i=1}^{\infty} \lambda_{V,i} \underbrace{\mathbb{E}[\phi_{V,i}(V)|Z]}_{:=\mu_i(Z)} \phi_{V,i}(\bullet), \quad (6)$$

Using this representation, we can simply place standard GP priors on these scalar-valued functions,

$$\mu_i \sim \mathcal{GP}(0, k_Z), \quad \forall i \in \mathbb{N}. \quad (7)$$

Note this prior ensures that $\mu(z) \in \mathcal{H}_V$ almost surely[2]. In Appendix A.5, we show this prior also induces a posterior mean for $\mu(z)$ that agrees exactly with the vector-valued kernel ridge regression estimator for $\mu(z)$ used in Singh et al. (2024), under the likelihood we later use for tractability.

Although the eigensystem is typically unknown, specifying the prior this way will not impede tractable training and inference, as in practice almost all terms will reduce to kernel evaluations. Specifying the prior on spectral co-ordinates also enables us to derive a novel closed-form objective for hyperparameter optimization in Section 5.

---

[1] e.g., if $\mathcal{V}$ is compact, countably discrete, or Euclidean.
[2] By the reproducing property, $\|\mu(z)\|_{\mathcal{H}_V}^2 = \sum_{i=1}^{\infty} \lambda_{V,i} \mu_i(z)^2$. Since $\mathbb{E}[\mu_i(z)^2] = 1$ and $\sum_i \lambda_{V,i} < \infty$, the series has finite mean by Fubini-Tonelli, so $\|\mu(z)\|_{\mathcal{H}_V}^2 < \infty$ a.s.

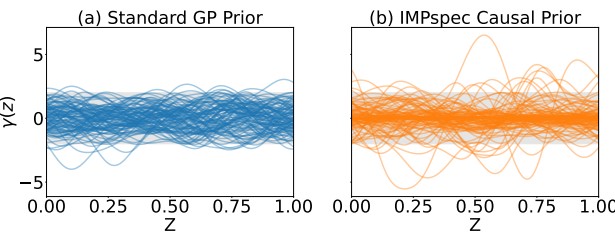

*Figure 2.* Samples from a standard GP prior (left) and IMPSPEC prior on $\gamma$ (right) for fixed $W = w$.

**IMPSPEC: The Spectral Interventional Mean Process**
We now combine the priors on $f$ and $\mu$ to obtain the induced prior over the causal function $\gamma$. Although these prior constructions enlarge the original RKHS-constrained function classes, they preserve the linear structure of the identifying formula. Indeed, substituting the outcome-regression representation into (1) gives

$$\gamma(w, z) = \mathbb{E}_V \left[ f\big(\psi_W(w) \otimes \psi_V(V)\big) \,\big|\, Z = z \right], \quad (8)$$

where the expectation is only over $V$. Passing the expectation over the linear functional[3] $f$ gives the (spectral) interventional mean process, (**IMPSPEC** for short).

$$\gamma(w, z) = f\big(\psi_W(w) \otimes \mu(z)\big). \quad (9)$$

To illustrate how the IMPSPEC prior differs from a standard GP prior, we draw (approximate) prior samples of $\gamma(w, \boldsymbol{z})$ at $\boldsymbol{z} = (z_1, \ldots, z_n)$ for fixed $w$ using a finite spectral truncation of the feature space[4], via the following steps:

(i) draw the first $m$ coordinates $\mu_i(\boldsymbol{z}) \sim \mathcal{N}(\mathbf{0}, K_Z)$, where $[K_Z]_{i,j} = k_Z(z_i, z_j)$;

(ii) form the $n \times n$ kernel matrix $K_\mu$, where $[K_\mu]_{j,k} = \sum_{i=1}^m \lambda_{V,i}\, \mu_i(z_j)\, \mu_i(z_k)$;

(iii) draw $\gamma(w, \boldsymbol{z}) \mid \mu \sim \mathcal{N}(\mathbf{0}, k_W(w, w) K_\mu)$

Figure 2 compares these samples, using Gaussian kernels $k_Z, k_W$ and eigenvalues $\lambda_{V,i} \propto e^{-\alpha i}$, with samples from a standard GP with a Gaussian kernel. Since $\gamma$ is an 'inner product' of GPs, it produces heavier-tailed sample paths.

## 4. Posterior Inference for IMPspec

We now derive our posterior inference scheme for $\gamma$. To ensure that (i) posterior moments are exactly tractable, and (ii) the posterior mean we derive precisely coincides with the kernel estimator of Singh et al. (2024), we use the following likelihood models:

$$Y = f(\psi_W(W) \otimes \psi_V(V)) + U, \quad U \sim \mathcal{N}(0, \sigma^2), \quad (10)$$

$$\phi_{V,i}(V) = \mu_i(Z) + \xi_i, \quad \xi_i \sim \mathcal{N}(0, \eta^2), \quad i \in \mathbb{N}. \quad (11)$$

---

[3]As exchanging the expectation with an *unbounded* linear functional $f$ is not immediate, in Appendix C, we show the constructions (8) and (9) converge in $L^2(\mathbb{P}_f)$.

[4]This truncation is used only to visualize prior samples.

These models should be interpreted as probabilistic surrogates for posterior computation, rather than structural assumptions on the causal data-generating process. (10) is the standard additive Gaussian observation model used in GP regression. (11) is less standard, but is essentially a coordinate-wise additive Gaussian model for the spectral features of $V$. It is exact under a linear kernel when $V = h(Z) + \xi$ with isotropic Gaussian noise, and is a reasonable local approximation when the spectral coordinates $\phi_{V,i}$ are smooth and the conditional noise in $V$ is small and approximately Gaussian. In Appendix A.4, we analyze the sensitivity of the derived posterior moments to mis-specification in this likelihood. Our posterior calibration procedure in Section 5.2 is designed to minimize any resulting coverage error in practice.

### 4.1. Posterior Moments for the Interventional Process

The priors in Section 3 and likelihoods (10)–(11) make posterior inference tractable by factorizing the problem into standard GP regression updates. In particular, (10) updates the posterior for the outcome-regression functional $f$ from the data $(Y, W, V)$, while (11) updates the posterior on each spectral coordinate $\mu_i$ from observations of $(\phi_{V,i}(V), Z)$. Thus, rather than performing inference directly over the interventional function $\gamma$, we perform ordinary scalar GP updates for the components that determine it.

The interventional process is then obtained by composing these posterior objects through the representation (9). In particular, its posterior mean and covariance can be derived by combining the Gaussian posterior updates for $f$ and the $\mu_i$'s, using the laws of total expectation and covariance to integrate over the posterior uncertainty in $\mu(z)$. The resulting expressions are closed form and stated below; the full covariance formula and derivation are given in Appendix C.

**Theorem 4.1.** *Under* (5)–(11)*, the posterior mean and variance of $\gamma(w, z) \mid \mathcal{X}^n$ are given by*

$$\mathbb{E}[\gamma(w, z) | \mathcal{X}^n] = \boldsymbol{\beta}(z)^\top K_V \boldsymbol{\alpha}(w) \quad (12)$$

$$\mathbb{V}ar[\gamma(w, z) | \mathcal{X}^n] = S_1 + S_2 + S_3 \quad (13)$$

*where*

$$S_1 = \boldsymbol{\beta}(z)^\top K_V(k(w, w)I - A(w)K_V)\boldsymbol{\beta}(z) \quad (14)$$

$$S_2 = \hat{k}(z, z)Tr[\tilde{K}_V(\boldsymbol{\alpha}(w)\boldsymbol{\alpha}(w)^\top - A(w))] \quad (15)$$

$$S_3 = \tau(k(z, z) - \boldsymbol{k}(z)^\top \boldsymbol{\beta}(z))k(w, w) \quad (16)$$

$$\boldsymbol{\alpha}(w) = D(w)(K_W \odot K_V + \sigma^2 I)^{-1}\boldsymbol{Y} \quad (17)$$

$$A(w) = D(w)(K_W \odot K_V + \sigma^2 I)^{-1}D(w) \quad (18)$$

$$\boldsymbol{\beta}(z) = (K_Z + \eta^2 I)^{-1}\boldsymbol{k}(z) \quad (19)$$

*and, for $x \in \{w, v, z\}$ we use the definitions*

$$D(w) := \mathrm{diag}(\boldsymbol{k}(w)) \quad \boldsymbol{k}(x) := [k_X(x, X_i)]_{i=1}^n,$$

$$\tilde{K}_V := \int \boldsymbol{k}(v)\boldsymbol{k}(v)^\top d\nu(v) \quad \tau := \sum_{i \in \mathbb{N}} \lambda_i,$$

Note that the only terms in Theorem 4.1 which may not have closed-form are the eigenvalue sum $\tau$ in $S_3$ and the matrix $\tilde{K}_V$ in $S_2$. The former is equal to $k_V(0,0)$ for any stationary $k_V$. The latter is only computed once at inference time, and can be approximated to arbitrary precision using a very large number of samples from the spectral measure $\nu$ (e.g., $m \gg 10^6$) at a cost of $\mathcal{O}(m)$ and error of $\mathcal{O}_p(m^{-1/2})$—or computed in closed form if available. In Appendix C we also present equivalent formulae for incremental effects and averages of such effects.

We also note the posterior mean (12) matches the estimator for $\gamma$ in (Singh et al., 2024), (4), with the noise variances playing the role of the regularization hyperparameters. As such, our method can be viewed as characterizing uncertainty around their estimator. This would not be possible without the function class expansion used to specify our priors. Had we instead constructed GP priors enforcing the original RKHS constraints, the posterior mean may be over-smoothed; see Sec 4. Kanagawa et al. (2018)).

## 4.2. Posterior Credible Interval Construction

To meaningfully measure uncertainty over $\gamma(w,z)$, we require posterior credible intervals (i.e., posterior intervals of a given probability mass), not just posterior variances. Since these cannot be derived in closed form, we approximate them by constructing a posterior GP for $\hat{\gamma}$ with mean and covariance given by the true moments.

$$\hat{\gamma} \sim \mathcal{GP}(\hat{m}, \hat{\kappa})$$
$$\hat{m}(w,z) = \mathbb{E}[\gamma(w,z) \mid \mathcal{X}^n],$$
$$\hat{\kappa}((w,z),(w',z')) = \mathbb{C}ov[\gamma(w,z), \gamma(w',z') \mid \mathcal{X}^n].$$

Equivalently, $\hat{\gamma}$ is the forward-KL projection of the true posterior process onto the family of Gaussian processes on $\mathcal{W} \times \mathcal{Z}$:

$$\hat{\gamma} \in \arg\min_{\tilde{\gamma} \sim \mathcal{GP}_{\mathcal{W} \times \mathcal{Z}}} \mathrm{KL}\left(P_{\gamma | \mathcal{X}^n} \,\|\, \mathbb{P}_{\tilde{\gamma}}\right). \quad (20)$$

Indeed, for each finite set of evaluation points, the Gaussian distribution minimizing $\mathrm{KL}(\mathbb{P}_{\gamma | \mathcal{X}^n} \| \mathbb{Q})$ over Gaussian $\mathbb{Q}$ is the one with matching mean and covariance. Thus, our credible intervals are obtained from a moment-matched Gaussian approximation to the posterior, analogous to variational inference but using the forward KL direction.

Algorithm 1 outlines how to construct credible intervals containing the central $\alpha\%$ posterior mass of $\gamma(w,z)$ at a grid of test points. The main cost is the usual $\mathcal{O}(n^3)$ from GP matrix inversions, but these can be amortized outside the loop over test points.

# 5. Hyperparameter Training and Calibration

In this section, we derive hyperparameter training and posterior calibration algorithms for IMPSPEC.

---

**Algorithm 1** $\alpha$-Credible Interval for $\gamma$

---

**Require:** Data $\mathcal{D}_n = \{(w_i, v_i, z_i, y_i)\}_{i=1}^n$; kernels $k_W, k_V, k_Z$; noise variances $\sigma^2, \eta^2$; test points $\{(w_m, z_m)\}_{m=1}^M$; probability level $\alpha$; spectral measure $\nu$; number of Monte Carlo samples $S$

**Ensure:** Posterior means $\{\hat{\mu}_m\}$ and credible intervals $\{\mathrm{CI}_m\}$

1: Compute $n \times n$ kernel matrices $K_V, K_W, K_Z$
2: Sample $(v_s)_{s=1}^S \sim \nu$
3: Compute $n \times S$ matrix $K_{VV_s}$ with entries $[K_{VV_s}]_{i,s} = k_V(v_i, v_s)$
4: $\tilde{K}_V \leftarrow \frac{1}{S} K_{VV_s} K_{VV_s}^\top, \quad \tau \leftarrow k_V(0,0)$
5: **for** $m = 1$ **to** $M$ **do**
6:     Compute $\boldsymbol{\alpha}(w_m)$ (17), $A(w_m)$ (18), $\boldsymbol{\beta}(z_m)$ (19)
7:     Compute $S_1$ (14), $S_2$ (15), and $S_3$ (16)
8:     $\hat{\mu}_m \leftarrow$ posterior mean at $(w_m, z_m)$ via (12)
9:     $\hat{\sigma}_m^2 \leftarrow$ posterior variance at $(w_m, z_m)$ via (13)
10:     $c \leftarrow \Phi^{-1}\left(\frac{1+\alpha}{2}\right)$
11:     $\mathrm{CI}_m \leftarrow [\hat{\mu}_m \pm c\, \hat{\sigma}_m]$
12: **end for**

---

## 5.1. Hyperparameter Tuning via Marginal Likelihood

Kernel hyperparameters and noise variances strongly determine both the generalization and uncertainty quantification performance of a GP, and are commonly selected by maximizing the marginal likelihood. For the outcome-regression model (10), this is standard: the model in (10) and GP prior on $f$ induces the usual GP regression (GPR) of $Y$ on $W$, $V$. The hyperparameters $(\theta_W, \theta_V)$ of $(k_W, k_V)$ and noise variance $\sigma^2$ can therefore be optimized via the standard marginal log-likelihood (MLL):

$$\log p(\boldsymbol{Y}|\boldsymbol{W}, \boldsymbol{V}) = \log \mathcal{N}(\boldsymbol{Y}|\boldsymbol{0}, K_W \odot K_V + \sigma^2 I) \quad (21)$$

Tuning the hyperparameters $\theta_Z$ of $k_Z$ and the feature-noise variance $\eta^2$ in (11) is less straight forward. Directly maximizing the MLL for each coordinate $\phi_{V,i}$ is infeasible because the coordinates are generally not available in closed form. However, for any stationary $k_V$, the spectral decomposition lets us aggregate these coordinate likelihoods into a single tractable weighted objective. To see this, note that for a fixed coordinate, (11) gives

$$\log p(\boldsymbol{\phi}_{V,i} \mid \boldsymbol{Z}) = c - \frac{1}{2} \log |\bar{K}_Z| - \frac{1}{2} \mathrm{Tr}\left[\bar{K}_Z^{-1} \boldsymbol{\phi}_{V,i} \boldsymbol{\phi}_{V,i}^\top\right],$$

where $\boldsymbol{\phi}_{V,i} = (\phi_{V,i}(V_1), \ldots, \phi_{V,i}(V_n))$, $\bar{K}_Z = K_Z + \eta^2 I$, and $c = -n \log(2\pi)/2$. Scaling by $\lambda_{V,i}$ and summing over coordinates gives the weighted log-likelihood:

$$\mathrm{WLL}_{\boldsymbol{\phi}}(\theta_Z, \eta) := \sum_{i=1}^\infty \lambda_{V,i} \log p(\boldsymbol{\phi}_{V,i} \mid \boldsymbol{Z})$$
$$= \tau c - \frac{1}{2}\tau \log |\bar{K}_Z| - \frac{1}{2}\mathrm{Tr}\left[\bar{K}_Z^{-1} K_V\right], \quad (22)$$

Note, the last line used the fact that $K_V = \sum_{i=1}^{\infty} \lambda_{V,i} \phi_{V,i} \phi_{V,i}^{\top}$ and $\tau := k_V(0,0) = \sum_i \lambda_{V,i}$.

Both objectives (21) and (22) have a closed form and so can be optimized using any gradient descent algorithm. As they have a complexity of $\mathcal{O}(n^3)$, for large datasets (e.g. $n \geq 10^4$) we estimate stochastic gradients on data minibatches, which has been successfully used in previous work using GPs and kernels (Chen et al., 2020; Jankowiak & Pleiss, 2021; Dance & Paige, 2022; Dance & Bloem-Reddy, 2024).

### 5.2. Calibration via Spectral Representation Learning

Since our posterior intervals rely on surrogate likelihoods and a moment-matched Gaussian approximation, their nominal credible level need not match their frequentist coverage. We therefore calibrate[5] the posterior directly by tuning the spectral measure $\nu$, which remains a free parameter of IMP-SPEC and controls posterior variance through $\widetilde{K}_V$.

Let $\text{CI}_{w,z,\alpha}(\nu)$ be the $\alpha$-credible interval for $\gamma(w,z)$ under spectral measure $\nu$, and $M = \{\nu_\beta : \beta \in \mathcal{B}\}$ be a parametric model for $\nu$. Our goal is to choose $\beta^*$ to minimise the *average posterior calibration error* across a grid of test values $T = (w_m, z_m)_{m=1}^M$ and coverage levels $I = (\alpha_l)_{l=1}^L$,

$$\sum_{(w,z) \in T} \sum_{\alpha \in I} |\mathbb{P}_{\mathcal{X}^n}(\gamma(w,z) \in \text{CI}_{w,z,\alpha}(\nu_\beta)) - \alpha| \quad (23)$$

For instance, if purely interested in 95% credible regions one could just fix $I = \{0.95\}$ and vary $w, z$ discretely and uniformly over a range of interest. Our experiments focus on the case where $V$ is continuous and so we use the parameterisation $\nu_\beta = \mathcal{N}(\bar{V}, \beta^2 I)$, where $\bar{V} = \frac{1}{n}\sum_{i=1}^n V_i$. For discrete variables, one could specify $\beta \in \triangle_k$ for the finite case and $\text{Poiss}(\beta)$ for the countably infinite case.

**Estimating the Calibration Loss** Following the framework of Syring & Martin (2019) for posterior calibration, we estimate (23) using the empirical bootstrap estimator for $\mathbb{P}_{\mathcal{X}^n}$ and a plug-in estimator for $\gamma(w,z)$. For the latter we use the estimator in Singh et al. (2024), as it is equivalent to the posterior mean of IMPSPEC. We then minimize this estimator for (23) via grid-search over $\beta$. Algorithm 2 summarizes the steps.

In Appendix C, we prove that under smoothness conditions and a sample-splitting criterion, our calibration loss estimator is consistent whenever the bootstrap and plug-in estimators are individually consistent. Under standard argmin-consistency conditions for M-estimation, this implies that grid search asymptotically selects the best-calibrated $\nu_\beta$ within the candidate family.

---

[5] A posterior $\alpha$-credible interval $C_\alpha(\mathcal{X}^n)$ for parameter $\theta$ is *calibrated* if it has the asserted coverage in repeated experiments, i.e. $\mathbb{P}_{\mathcal{X}_n}(\theta \in C_\alpha(\mathcal{X}^n)) = \alpha$ (Rubin, 1984).

---

**Algorithm 2** Calibration of the Spectral Measure

**Require:** Data $\mathcal{D}_n$; spectral parameter grid $\mathcal{B}$; test-point grid $T$; interval grid $I$; number of bootstrap replications $B$

**Ensure:** Calibrated spectral parameter $\beta^*$
1: Compute fixed plug-in estimates $\hat{\gamma}_{w,z}$ for all $(w,z) \in T$
2: **for** $\beta \in \mathcal{B}$ **do**
3:    **for** $(w, z, \alpha) \in T \times I$ **do**
4:       **for** $b = 1$ **to** $B$ **do**
5:          Draw bootstrap sample $\mathcal{D}_n^{(b)}$ from $\mathcal{D}_n$
6:          Compute $\text{CI}_{w,z,\alpha}^{(b)}(\nu_\beta)$ using Alg 1
7:       **end for**
8:       $\widehat{\Delta}_{w,z,\alpha,\beta} \leftarrow \left| \frac{1}{B}\sum_{b=1}^B \mathbb{1}\left\{\hat{\gamma}_{w,z} \in \text{CI}_{w,z,\alpha}^{(b)}(\nu_\beta)\right\} - \alpha \right|$
9:    **end for**
10:   $\widehat{L}(\beta) \leftarrow \sum_{(w,z,\alpha) \in \Theta} \widehat{\Delta}_{w,z,\alpha,\beta}$
11: **end for**
12: $\beta^* \leftarrow \arg\min_{\beta \in \mathcal{B}} \widehat{L}(\beta)$

---

## 6. Related Work

**Bayesian Methods for Uncertainty Quantification** Our work is situated in the wider literature on Bayesian methods for causal uncertainty quantification (Alaa & Van Der Schaar, 2017; Hahn et al., 2020; Witty et al., 2020; Oganisian & Roy, 2021; Chau et al., 2021b). Several works use GPs for causal uncertainty: Alaa & Van Der Schaar (2017) use multi-task GPs for individualized treatment-effect uncertainty, while Witty et al. (2020) use GP priors in a structural causal model with latent confounding. These works target *individual-level* counterfactual uncertainty, whereas we quantify uncertainty over identified interventional functions describing *conditional average* effects.

The closest method to ours is Chau et al. (2021b) ("Bayes-IMP"), who also use GPs to quantify uncertainty around causal functions in RKHSs. BayesIMP constructs bespoke covariance kernels to make the GP prior compatible with the original RKHS. While elegant, this introduces practical difficulties: the modified kernels are closed-form only in special cases, can become pathologically nonstationary for radial base kernels leading to underfitting, and require finite-dimensional variance approximations that can collapse outside the training support. We analyze these effects in Appendix A.6 and observe them empirically in Figures 3 and 4. Our method instead expands the relevant function classes and uses spectral representations, enabling standard GP priors and closed-form posterior moments.

**Frequentist Methods for Causal Effect Functions** Several frequentist methods have been developed for uncertainty quantification of causal effect functions, such as CATEs and dose-response curves (Kennedy et al., 2017; Oprescu et al., 2019; Semenova & Chernozhukov, 2021; Kennedy, 2023).

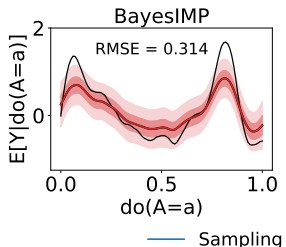 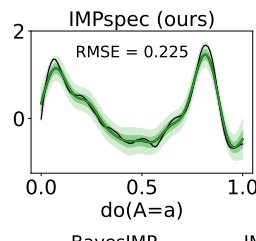 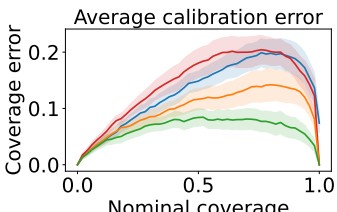 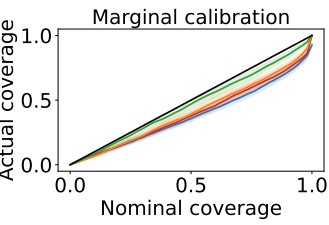

Sampling GP — BayesIMP — IMPspec (without calibration) — IMPspec (with calibration)

*Figure 3.* **Toy Example**. Left and Middle Left: Estimated dose-response curve $\mathbb{E}[Y|\mathrm{do}(A = a)]$ and average RMSE, using BayesIMP (Chau et al., 2021b) (red) and our method IMPSPEC (green). True dose-response curve = black line, posterior mean = colored lines, light shading = 90 % credible intervals, dark shading = interquartile range. All quantities are averaged over 50 trials. BayesIMP underfits the causal function, plausibly due to the non-stationarity of the kernel used (see Section 6 and Appendix A). Right and Middle Right: Calibration plots of different GP-based methods over 50 trials, with 95% confidence intervals (shaded regions) estimated using 100 bootstrap replications. IMPSPEC is best calibrated and is significantly improved by optimizing the spectral representation.

These approaches typically introduce smoothness, parametric, or functional-form assumptions to obtain valid inference despite the lack of standard Riesz representers. Our method instead uses spectral GP priors to build posterior uncertainty around RKHS estimators of such functions.

**Causal Bayesian Optimization**  Our work is also related to causal Bayesian optimization, where observational data and causal structure guide intervention search (Aglietti et al., 2020; Chau et al., 2021b). In this setting, IMPSPEC's posterior GP from observational data can be used as an informative surrogate initialization for intervention search.

## 7. Experiments

We now evaluate IMPSPEC in several synthetic experiments and a real-world dataset.[6] Implementation details for all methods and experiments are in Appendix B.

### 7.1. A Toy Example

**Experiment**  We first compare IMPSPEC against related GP-based methods (BayesIMP (Chau et al., 2021b) and the sampling-based GP of Witty et al. (2020)) in a toy simulation. We generate two separate datasets, $\mathcal{D}_1 = \{M_i^{(1)}, Y_i^{(1)}\}_{i=1}^{100}$ and $\mathcal{D}_2 = \{A_j^{(2)}, M_j^{(2)}\}_{j=1}^{100}$, from a structural model $M = f(A) + \xi$, $Y = g(M) + U$ with independent noise $\xi \sim \mathcal{N}(0, \sigma_\xi^2)$, $U \sim \mathcal{N}(0, \sigma_U^2)$ and aim to estimate the posterior over $\mathbb{E}[Y \mid \mathrm{do}(A = a)]$ for different values of $a$. Although in this model $\mathbb{E}[Y \mid \mathrm{do}(A = a)] = \mathbb{E}[Y \mid A = a]$, since we only observe $(M, Y)$ and $(A, M)$ separately each method must use the two-step formula $\mathbb{E}[Y \mid \mathrm{do}(A = a)] = \int \mathbb{E}[Y \mid M = m] \, dp(m \mid a)$. Thus, for IMPSPEC we set $(W, V, Z) := (\emptyset, M, A)$.

**Results**  We evaluate methods by (i) root mean squared error (RMSE) of posterior means relative to the true $\mathbb{E}[Y \mid \mathrm{do}(A = a)]$, (ii) calibration error of $\alpha$-credible intervals as

| Method | RMSE | Cal. error | IS95 |
|---|---|---|---|
| Sampling GP | $0.28 \pm 0.06$ | $0.13 \pm 0.01$ | $2.20 \pm 0.16$ |
| BayesIMP | $0.31 \pm 0.08$ | $0.14 \pm 0.01$ | $2.50 \pm 0.36$ |
| IMPspec-nocal | $\mathbf{0.22 \pm 0.05}$ | $0.10 \pm 0.01$ | $1.36 \pm 0.07$ |
| IMPspec | $\mathbf{0.22 \pm 0.05}$ | $\mathbf{0.07 \pm 0.01}$ | $\mathbf{1.16 \pm 0.05}$ |

*Table 1.* (Toy Example) Mean $\pm$ std. dev. RMSE, calibration error, and 95% interval score for ATE estimation from 50 trials. Bold = best performing method.

defined by (23), and (iii) the 95% interval score (defined in Appendix B.1), which penalizes both interval width and miscoverage. All metrics are averaged over a grid of intervention levels $(a_i)_{i=1}^m \subset [0, 1]$. The mean and standard deviations over 50 trials are reported in Figure 3 and Table 1.[7] IMPSPEC obtains the best RMSE, calibration error, and IS95, with spectral calibration substantially improving both coverage and interval score. By contrast, BayesIMP systematically underfits, which we attribute to the kernel non-stationarity discussed in Section 6 and Appendix A. The sampling GP performs similarly to BayesIMP. In Appendix B, we further show that IMPSPEC remains robust under heavy-tailed outcome noise and limited treatment support, and across calibration and kernel choices.

### 7.2. Synthetic Benchmark Causal Graph

**Experiment**  We next implement the same methods, along with the frequentist doubly-robust estimator of Kennedy et al. (2017) for continuous treatments, and orthogonal random forests (ORF) (Oprescu et al., 2019), on a well-known synthetic benchmark DAG studied in Aglietti et al. (2020); Chau et al. (2021b) (see Figure 8, middle in Appendix B for graph). This design contains observed variables $A, B, C, D, E$, an outcome $Y$ and unobserved confounders $U_1, U_2$. We aim to estimate the conditional average treatment effect, $\mathbb{E}[Y|\mathrm{do}(D = d), B = b] = \int_{\mathcal{C}} \mathbb{E}[Y|D = d, B = b, C = c]\mathbb{P}_{C|B}(dc|b)$, where the formula is derived using the back-door criterion. Thus, the variable mapping

---

[6]Code: https://github.com/HWDance/impspec.

[7]The mean and standard deviation of the calibration error and IS95 are computed by bootstrapping over the distribution of trials.

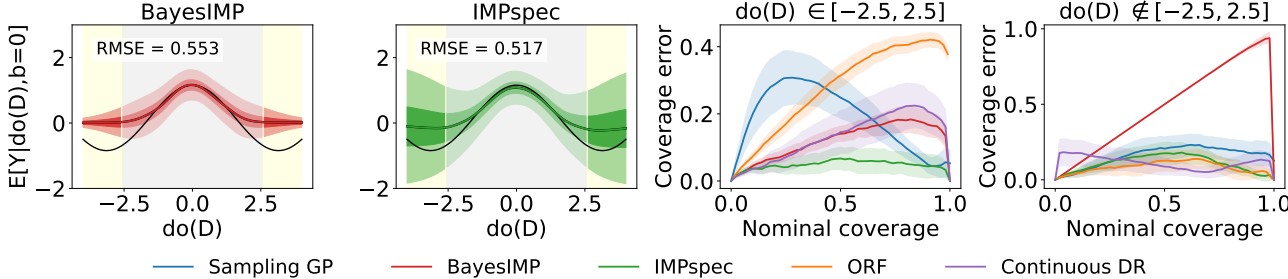

*Figure 4.* **Synthetic Benchmark**. Left + Middle-Left = trial-averaged posterior mean and credible intervals (50% and 95%) for conditional dose-response curve $\mathbb{E}[Y|\mathrm{do}(D=d), b=0]$ (black) using BayesIMP (red) and our IMPSPEC (green). Grey region = support of $\mathbb{P}_D$. Right + Middle-Right = calibration error of different methods in- and out-of-distribution. BayesIMP's posterior variance collapses out-of-distribution due to the finite-dimensional approximations used. IMPSPEC is best calibrated overall.

| Method | RMSE | Cal. error | IS95 |
|---|---|---|---|
| ORF | $0.84 \pm 0.16$ | $0.16 \pm 0.07$ | $10.34 \pm 0.95$ |
| Continuous DR | $0.61 \pm 0.32$ | $0.13 \pm 0.02$ | $8.37 \pm 13.53$ |
| Sampling GP | $0.40 \pm 0.15$ | $0.19 \pm 0.03$ | $6.67 \pm 5.90$ |
| BayesIMP | $0.38 \pm 0.13$ | $0.11 \pm 0.01$ | $2.99 \pm 2.38$ |
| IMPspec (ours) | $\mathbf{0.36 \pm 0.16}$ | $\mathbf{0.05 \pm 0.02}$ | $\mathbf{2.27 \pm 2.03}$ |

*Table 2.* (Synthetic Benchmark) Mean ± std. dev. RMSE, calibration error, and 95% interval score for CATE estimation from 50 trials for in-sample range $\mathrm{do}(D) \in [-2.5, 2.5]$. Bold is best.

| Method | Front-door | Back-door | Healthcare |
|---|---|---|---|
| BO | $1.753 \pm 1.326$ | $0.760 \pm 0.519$ | $0.064 \pm 0.029$ |
| CBO | $0.656 \pm 0.604$ | $0.383 \pm 0.427$ | $0.008 \pm 0.018$ |
| BayesIMP | $0.698 \pm 0.825$ | $0.570 \pm 0.555$ | $0.003 \pm 0.010$ |
| IMPspec (ours) | $\mathbf{0.247 \pm 0.284}$ | $\mathbf{0.335 \pm 0.420}$ | $\mathbf{0.000 \pm 0.000}$ |

*Table 3.* (CBO) mean ± std. dev cumulative regret for $\mathbb{E}[Y|\mathrm{do}(B), b=0]$ (front-door), $\mathbb{E}[Y|\mathrm{do}(D), b=0]$ (back-door) on synthetic benchmark, and $\mathbb{E}[\mathrm{Vol}|\mathrm{do}(\mathrm{Statin})]$ on healthcare application from (Aglietti et al., 2020) (50 trials). Bold is best.

used for IMPSPEC is $(W, V, Z) = ((D, B), C, B)$. Implementation details for each method are in Appendix B.3.

**Results** Table 2 reports the average RMSE, calibration error, and 95% interval score, defined as in the toy example, over 50 trials with $n = 100$ samples. We fix $B = 0$ and vary the intervention value $d$ across a grid spanning the support of $\mathbb{P}_D$, i.e., $[-2.5, 2.5]$. IMPSPEC achieves the lowest RMSE, calibration error, and IS95, indicating improved effect estimation and uncertainty quantification. ORF performs poorly in this benchmark, as expected, since the treatment-effect profile is nonlinear whereas ORF uses a locally linear treatment-effect model. Figure 4 also shows trial-averaged posterior means and credible intervals for BayesIMP and IMPSPEC (left panels), and calibration curves in- and out-of-support (middle-right/right), where out-of-support corresponds to $d \in [-4, -2.5] \cup [2.5, 4]$. BayesIMP's finite-dimensional approximation causes posterior variances to collapse outside the support of $\mathbb{P}_D$ (see Appendix A.6 for details), leading to severe out-of-support miscalibration. The other methods achieve similar out-of-support calibration error to IMPSPEC, but worse in-support calibration error.

### 7.3. Causal Bayesian Optimization Tasks

**Experiment** We now showcase the usefulness of IMP-SPEC in improving causal Bayesian optimization (CBO) (Aglietti et al., 2020). CBO aims to identify the intervention $a^*$ that either maximizes or minimizes a causal effect such as $\mathbb{E}[Y \mid \mathrm{do}(A = a)]$, in as few interventional queries

as possible, by maintaining a GP surrogate for the causal effect. At each iteration, a BO algorithm selects a treatment level $a_i$, which in practice corresponds to running a large-scale experiment to estimate $\mathbb{E}[Y \mid \mathrm{do}(A = a_i)]$, updates the GP posterior using the new interventional observation $(a_i, \hat{\mathbb{E}}[Y \mid \mathrm{do}(A = a_i)])$, and uses it to determine where to search next via an acquisition function. Unlike standard BO, CBO uses observational data to construct the interventional GP prior. We therefore use IMP-SPEC's posterior GP $\hat{\gamma}$ to define the BO prior, adding an additional prior covariance kernel $k$ as is standard (Aglietti et al., 2020): $\hat{\gamma} \sim \mathcal{GP}(\hat{m}, \hat{\kappa} + k)$. For comparison, we implement BayesIMP (implemented analogously to IMPSPEC), classical CBO (Aglietti et al., 2020) and standard BO which ignores observational data. We evaluate all methods on three tasks: (i) minimizing the CATE $\mathbb{E}[Y \mid \mathrm{do}(D = d), B = 0]$ in the synthetic benchmark, (ii) maximizing the ATT $\mathbb{E}[Y \mid \mathrm{do}(B = b), B = 0]$ in the synthetic benchmark, and (iii) minimizing the average effect of Statin dosage on Cancer volume $\mathbb{E}[\mathrm{Vol} \mid \mathrm{do}(\mathrm{Statin} = s)]$ using a simulator based on real healthcare data and a known causal graph (see Figure 8, right, in Appendix B.4). Appendix B.4 contains the designs and implementation details.

**Results** Table 3 displays cumulative regret scores (i.e., the sum of differences between the observed vs. optimal estimand quantity) after 10 iterations for two different causal effects on the synthetic benchmark (middle causal graph of Figure 8) and for the average causal effect of `statin` on `cancer_vol` in the healthcare example. Figure 5 in

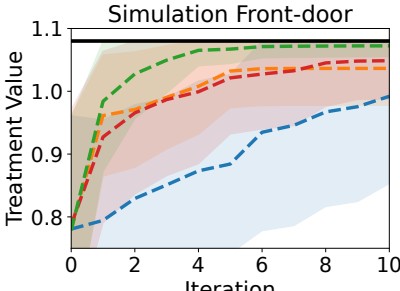 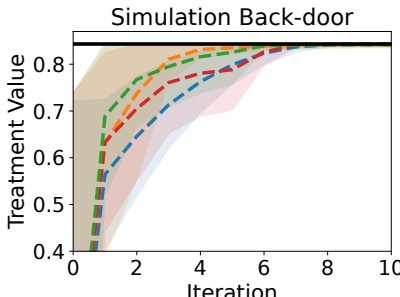 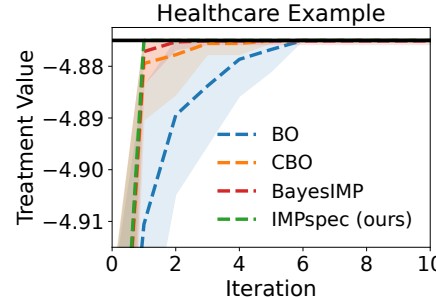

*Figure 5.* **CBO Experiments:** Best values per iteration (50 trials). Left: Synthetic benchmark targeting $\max_b \mathbb{E}[Y|\mathrm{do}(b)], B = 0$ via front-door criterion, Middle: Synthetic benchmark targeting $\min_d \mathbb{E}[Y|\mathrm{do}(d)], B = 0$ via back-door criterion, Right: Healthcare example from Aglietti et al. (2020) targeting $\min_s \mathbb{E}[\mathrm{Vol}|\mathrm{do}(\mathrm{Statin} = s)]$ via back-door criterion. Dashed lines = mean, shading = standard deviation, black line = optimal treatment value. Our method (IMPSPEC) converged fastest on average in all three experiments.

| Stage | Batch size | Time (s) | Peak GPU Memory (GB) |
|---|---|---|---|
| Training | 512 | 8.83 | 0.0 |
| Calibration | 512/9915† | 54.50 | 14.1 |
| Inference | 9915 | 5.03 | 11.1 |

*Table 4.* Runtime and GPU memory for IMPSPEC on 401(k). Training uses minibatch SGD; inference uses full-dataset computations. †Calibration includes retraining and posterior inference, and therefore uses both batch sizes.

Appendix B.4 displays convergence profiles. IMPSPEC converged fastest and obtained the lowest cumulative regret across all three tasks. In one case, only IMPSPEC could find the approximate optimum value within the maximum number of iterations, demonstrating the usefulness of its uncertainty estimates in improving optimal decision making.

### 7.4. 401k Dataset: Estimating CATE Uncertainty

We use the well-known 401(k) dataset, previously studied in Chernozhukov & Hansen (2004) and commonly used as a benchmark for treatment-effect estimation. We take 401(k) pension plan eligibility as the treatment $A$, net financial assets as the outcome $Y$, and income $I$ as the conditioning variable. The remaining demographic and financial covariates $X_{-I}$ are used for adjustment, including age, marital status, two-earner status, IRA participation, and home ownership. Thus, in the notation of (1), we set $(W, V, Z) = ((A, I), X_{-I}, I)$ and estimate the income-dependent CATE $z \mapsto \gamma((1, z), z) - \gamma((0, z), z)$. Since there is no available ground-truth causal effect, we use this primarily as a practical case study for assessing whether IMPSPEC produces plausible uncertainty-aware effect estimates at a realistic scale. We also report runtime and memory requirements to assess computational feasibility.

The estimated CATE increases with income, and the 95% credible band lies above zero for incomes above roughly $30,000. This suggests a positive effect of 401(k) eligibility on net financial assets, with stronger evidence and larger effects for higher-income individuals, consistent with previ-

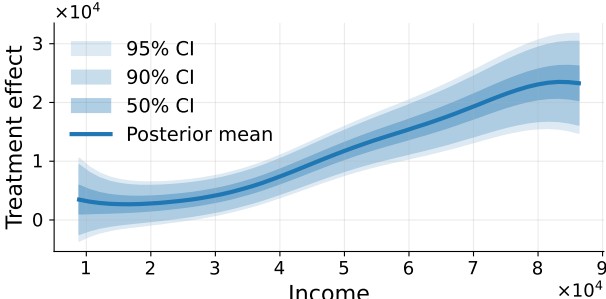

*Figure 6.* IMPSPEC posterior mean and central 50%, 90%, and 95% credible intervals for the CATE of 401(k) eligibility on net financial assets as a function of income.

ous studies of this dataset (Chernozhukov & Hansen, 2004). By using minibatch gradients for hyperparameter training, the method runs in around a minute on a NVIDIA RTX 4500 GPU, with peak allocated GPU memory below 16GB, demonstrating feasibility at this dataset scale (see Table 4).

## 8. Discussion

**Conclusion**  We introduced IMPSPEC, a GP-based framework for uncertainty quantification of causal functions admitting RKHS inner-product representations. IMPSPEC provides tractable uncertainty estimates for function-valued causal targets and a calibration algorithm targeting frequentist coverage of posterior credible intervals. Across experiments, IMPSPEC improved calibration and interval scores while remaining competitive in effect estimation.

**Limitations and Future Work**  IMPSPEC uses surrogate Gaussian likelihoods and a moment-matched GP posterior approximation to achieve tractability and recover the RKHS estimator of Singh et al. (2024) as the posterior mean. While calibration adjusts uncertainty scales within a chosen spectral family, richer posterior approximations or spectral families may further improve coverage when the true uncertainty is strongly non-Gaussian. Future work could also incorporate sparse kernel methods for much larger datasets.

## Impact Statement

This work aims to improve uncertainty quantification for causal effect estimation. Reliable uncertainty estimates can help practitioners assess the risks of candidate interventions and guide safer decision-making in applications such as healthcare, policy evaluation, and intervention search. Its reliability depends on the appropriateness of the underlying identification assumptions, modeling choices, and calibration procedure. In high-stakes applications, inaccurate uncertainty estimates or invalid causal assumptions could lead to misleading confidence in harmful interventions, so the method should be used alongside domain expertise, sensitivity analysis, and appropriate validation.

## Acknowledgements

The authors declare the support of the Gatsby Charitable Foundation.

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

# Appendix Contents

# A. Additional Details

## A.1. Mathematical Assumptions

Here we list the full set of assumptions on the variable spaces and kernels used in the main text.

**Assumption A.1** (Topological assumptions). Let $Y \in \mathcal{Y}, V \in \mathcal{V}, W \in \mathcal{W}, Z \in \mathcal{Z}$. We assume

1. $\mathcal{X}$ is a standard Borel space with Borel $\sigma$-algebra $\mathcal{B}(\mathcal{X})$ for $\mathcal{X} \in \{\mathcal{Y}, \mathcal{W}, \mathcal{V}, \mathcal{Z}\}$.

2. $\mathcal{V}$ is $\sigma$-compact (i.e., $\mathcal{V} = \cup_{j \in I} \mathcal{V}_j$ with $\mathcal{V}_j$ compact and $I$ countable).

3. $\mathcal{Y} = \mathbb{R}$.

*Remark* A.2. In practice, at least one of $V, W, Z$ correspond to the treatment variable $(A)$ in different causal set-ups. Two examples are given in the main text (in both of those cases $Z := A$). Although our method is motivated by, and designed for, the continuous treatment case (i.e., $\mathcal{A} = \mathbb{R}$), it can in principle be applied to the scenario where $A$ is binary or discrete. Thus, we do not formally list this as a topological assumption.

**Assumption A.3** (Kernel assumptions). Let $k_V : \mathcal{V}^2 \to \mathbb{R}, k_W : \mathcal{W}^2 \to \mathbb{R}$ and $k_Z : \mathcal{Z}^2 \to \mathbb{R}$ be positive definite kernels. We assume

1. $k_V, k_W, k_Z$ are Borel measurable

2. $k_V, k_W, k_Z$ are continuous, bounded and characteristic

**Assumption A.4** (Mercer's theorem). Let $\mathcal{T}_{k,\nu} : L^2(\nu) \to L^2(\nu), \varphi \mapsto \int k_V(v, \bullet)\varphi(v)\nu(dv)$ with $\nu$ finite and non-degenerate. We assume

1. $\mathcal{T}_{k,\nu}$ has eigendecomposition $(\lambda_{V,i}, \phi_{V,i})_{i \in I}$ with $I$ countable.

2. $\sum_{i \in I} \lambda_{V,i} < \infty$.

3. $(e_i)_{i \in I} := (\lambda_{V,i}^{\frac{1}{2}} \phi_{V,i})_{i \in I}$ is an orthonormal basis (ONB) of the RKHS $\mathcal{H}_\mathcal{V}$ associated with $k$.

4. $k_V(v, v') = \sum_{i \in I} e_i(v)e_i(v')$ where the convergence is absolute and uniform on any compact $A \times B \subset \mathcal{V}^2$.

5. $(e_i(v))_{i \in I} \in \ell_2(I), \quad \forall v \in \mathcal{V}$.

6. $\lambda_{V,i} > 0, \quad \forall i \in I$.

*Remark* A.5. Conditions 1-5 hold in Assumption A.4 under Assumption A.1 and Assumption A.3 (see Proposition 1 and Theorem 3 in Sun (2005). Condition 6 holds for popular characteristic kernels $k$ (e.g. Gaussian, Matérn, Rational quadratic, and Gamma exponential).

*Remark* A.6. Without loss of generality we will take $I = \mathbb{N}$ in this work for concreteness. In cases where $\mathcal{T}_{k,\nu}$ is a finite rank operator, we restrict the index set $I = \text{Rank}(\mathcal{T}_{k,\nu})$, so that the assumptions hold.

## A.2. Background on Causal Graphical Models and Identification Criteria

Here we describe some key concepts used in the causal graphical modelling literature (Pearl, 2009b).

**Graphs:** A Graph $\mathcal{G}$ is a set of directed edges $E$ and indices $[m] = \{1, ..., m\}$. The edges of $\mathcal{G}$ can be encoded by a parent function pa $: [m] \to 2^{[m]}$ in the sense that $l \in \text{pa}(j) \Leftrightarrow l \to j$ in $\mathcal{G}$. If $j$ has no incoming edges then pa$(j) = \emptyset$. In this case, one can write $\mathcal{G} = (E, \text{pa})$.

**Causal DAGs:** a graph $\mathcal{G} = (\boldsymbol{X}, \text{pa})$ is a causal DAG over random variables $\boldsymbol{X} := (X_i)_{i=1}^m \sim \mathbb{P}$, if the factorisation $\mathbb{P} = \prod_i \mathbb{P}_{X_i|X_{\text{pa}(i)}}$ holds and, for each $i \in \{1, \ldots, m\}$, $X_{\text{pa}(i)}$ are all direct causes of $X_i$.

**Definition A.7** (Back-door criterion (Pearl, 1995)). Take a causal DAG $\mathcal{G}$ over $(X_i)_{i=1}^m$, and let $(A, V, Y) \subseteq (X_i)_{i=1}^m$. $V$ satisfies the back-door criterion with respect to $(A, Y)$ if

1. No node in $V$ is a descendent of $A$.

2. $V$ blocks every path between $A$ and $Y$ that contains an edge pointing into $A$

**Definition A.8** (Front-door criterion (Pearl, 1995)). Take a causal DAG $\mathcal{G}$ over $(X_i)_{i=1}^m$, and let $(A, M, Y) \subseteq (X_i)_{i=1}^m$. $M$ satisfies the front-door criterion with respect to $(A, Y)$ if

1. $M$ intercepts all directed paths from $A$ to $Y$.

2. There is no back-door path between $A, M$.

3. Every back-door path between $M$ and $Y$ is blocked by $A$.

In the following we present the known formulae for all causal effects of interest in this work. All these effects take the form of Equation (1) in the main text, or a marginal expectation of it. Details for ATE and CATE can be found in (Pearl, 2009b) and for ATT in (Shpitser & Pearl, 2009).

**Effects under the back-door criterion:** In the following $A$ is the treatment, $Y$ is the outcome, and $(V, Z)$ satisfy the back-door criterion w.r.t. $(A, Y)$ (see Figure 1, left).

$$\text{ATE}_{BD}(a) = \int_{\mathcal{V} \times \mathcal{Z}} \mathbb{E}[Y|A = a, V = v, Z = z] \mathbb{P}_{V,Z}(dv \times dz)$$

$$\text{CATE}_{BD}(a, z) = \int_{\mathcal{V}} \mathbb{E}[Y|A = a, V = v, Z = z] \mathbb{P}_{V|Z}(dv|z)$$

$$\text{ATT}_{BD}(a, a') = \int_{\mathcal{V}} \mathbb{E}[Y|A = a, V = v, Z = z] \mathbb{P}_{V|A}(dv|a')$$

**Effects under the front-door criterion:** In the following $A$ is the treatment, $Y$ is the outcome, and $M$ satisfies the front-door criterion w.r.t. $(A, Y)$ (see Figure 1, right).

$$\text{ATE}_{FD}(a) = \int_{\mathcal{A}} \int_{\mathcal{M}} \mathbb{E}[Y|A = a', M = m] \mathbb{P}_{M|A}(dm|a) \mathbb{P}_A(da')$$

$$\text{ATT}_{FD}(a, a') = \int_{\mathcal{M}} \mathbb{E}[Y|A = a', M = m] \mathbb{P}_{M|A}(dm|a)$$

### A.3. IMPSpec Modifications for Causal Data Fusion Setting

Our method can be used both in situations where $\mathcal{X}^n$ is a single dataset of i.i.d. observations from a causal graph, or in certain situations where $\mathcal{X}^n$ is comprised of two (i.i.d.) datasets $\mathcal{D}_1, \mathcal{D}_2$ of partial observations from the causal graph. Following Chau et al. (2021b), we refer to the latter situation as the *causal data fusion* setting. In general, to estimate a causal effect of the form $\gamma(w, z)$ as defined in Equation (1) in the main text, one must have access to observation sets $\mathcal{D}_1 = (Y_i^{(1)}, W_i^{(1)}, V_i^{(1)})_{i=1}^{m_1}$ and $\mathcal{D}_2 = (Z_i^{(2)}, V_i^{(2)})_{i=1}^{m_2}$ (in the case where $W = \emptyset$ it suffices to have $\mathcal{D}_1 = (Y_i^{(1)}, V_i^{(1)})_{i=1}^{m_1}$). In such cases, we derive the posteriors on $f \mid \mathcal{D}_1$ and $\mu \mid \mathcal{D}_2$ and combine them (as in the proof of Theorem 4.1) to recover the posterior moments of $\gamma$ following the same calculation steps as in the proof of Theorem 4.1. The resulting posterior moments are then as follows.

$$\mathbb{E}[\gamma(w, z)|\mathcal{X}^n] = \boldsymbol{\beta}^{(2)}(z)^\top K_{V^{(2)}V^{(1)}} \boldsymbol{\alpha}^{(1)}(w)$$

$$\mathbb{V}ar[\gamma(w, z)|\mathcal{X}^n] = S_1 + S_2 + S_3$$

where

$$S_1 = \boldsymbol{\beta}^{(2)}(z)^\top K_{V^{(2)}} \boldsymbol{\beta}^{(2)}(z) k_W(w, w) - \boldsymbol{\beta}^{(2)}(z)^\top K_{V^{(2)}V^{(1)}} A^{(1)}(w) K_{V^{(1)}V^{(2)}} \boldsymbol{\beta}^{(2)}(z)$$

$$S_2 = \hat{k}^{(2)}(z, z) Tr[\tilde{K}_{V^{(1)}}(\boldsymbol{\alpha}^{(1)}(w)\boldsymbol{\alpha}^{(1)}(w)^\top - A^{(1)}(w))]$$

$$S_3 = \left(\sum_{i \in I} \lambda_i\right) k_W(w, w) \hat{k}^{(2)}(z, z))$$

and the associated quantities are defined as

$$\boldsymbol{\alpha}^{(1)}(w) = D^{(1)}(w) \left(K_{W^{(1)}V^{(1)}} + \sigma^2 I\right)^{-1} \boldsymbol{Y}^{(1)}, \qquad A^{(1)}(w) = D^{(1)}(w) \left(K_{W^{(1)}V^{(1)}} + \sigma^2 I\right)^{-1} D^{(1)}(w),$$

$$\boldsymbol{\beta}^{(2)}(z) = (K_{Z^{(2)}} + \eta^2 I)^{-1} \boldsymbol{k}^{(2)}(z), \qquad \hat{k}^{(2)}(z,z) = k_Z(z,z) - \boldsymbol{k}^{(2)}(z)^\top \boldsymbol{\beta}^{(2)}(z),$$

$$D^{(1)}(w) = \mathrm{diag}(\boldsymbol{k}^{(1)}(w)), \qquad \tilde{K}_{V^{(1)}} = \int \boldsymbol{k}^{(1)}(v)\boldsymbol{k}^{(1)}(v)^\top d\nu(v),$$

$$(K_{W^{(1)}})_{ij} = k_W(W_i^{(1)}, W_j^{(1)}), \qquad (K_{V^{(1)}})_{ij} = k_V(V_i^{(1)}, V_j^{(1)}),$$

$$(K_{V^{(2)}})_{ij} = k_V(V_i^{(2)}, V_j^{(2)}), \qquad (K_{V^{(2)}V^{(1)}})_{ij} = k_V(V_i^{(2)}, V_j^{(1)}),$$

$$K_{V^{(1)}V^{(2)}} = (K_{V^{(2)}V^{(1)}})^\top, \qquad K_{W^{(1)}V^{(1)}} = K_{W^{(1)}} \odot K_{V^{(1)}},$$

$$\boldsymbol{k}^{(1)}(w) = [\, k_W(w, W_1^{(1)}), \ldots, k_W(w, W_{m_1}^{(1)})\,]^\top, \qquad \boldsymbol{k}^{(2)}(z) = [\, k_Z(z, Z_1^{(2)}), \ldots, k_Z(z, Z_{m_2}^{(2)})\,]^\top.$$

Thus, in any experiments which require causal data fusion with $\mathcal{D}_1$ and $\mathcal{D}_2$, we use the above posterior moments instead of those given in the main text for practical implementation. Algorithm 3 below gives the resulting algorithm for computing posterior credible intervals in this setting.

*Remark* A.9 (Case $W = \emptyset$). In the special case where $W = \emptyset$, the terms in the posterior moments involving $k_W$ and $K_{W^{(1)}}$ drop out of the model. Concretely, $K_{W^{(1)}} = I$, $k_W(w,w) = 1$, and $\boldsymbol{k}^{(1)}(w) = \boldsymbol{1}$, so that $D^{(1)}(w) = I$ and $K_{W^{(1)}V^{(1)}}$ reduces to $K_{V^{(1)}}$. Consequently, $\boldsymbol{\alpha}^{(1)}(w)$ and $A^{(1)}(w)$ simplify to

$$\boldsymbol{\alpha}^{(1)} = (K_{V^{(1)}} + \sigma^2 I)^{-1} \boldsymbol{Y}^{(1)}, \qquad A^{(1)} = (K_{V^{(1)}} + \sigma^2 I)^{-1},$$

and the remaining equations for $\boldsymbol{\beta}^{(2)}(z)$, $\hat{k}^{(2)}(z,z)$, and the variance terms $S_1, S_2, S_3$ remain unchanged. This modification of Algorithm 3 is used in the Experiments for the Toy Example (Figure 8, left) and in the Healthcare Simulator (Figure 8, right), which are described in detail in Appendix A.

---

**Algorithm 3** $\alpha$-Credible Interval for $\gamma$ in Causal Data Fusion Setting

---

**Require:** Datasets $\mathcal{D}_1 = \{(Y_i^{(1)}, W_i^{(1)}, V_i^{(1)})\}_{i=1}^{m_1}$, $\mathcal{D}_2 = \{(Z_i^{(2)}, V_i^{(2)})\}_{i=1}^{m_2}$; kernels $k_W, k_V, k_Z$; noise variances $\sigma^2, \eta^2$;
   test points $\{(w_m, z_m)\}_{m=1}^M$; confidence level $\alpha$; spectral measure $\nu$; Monte Carlo samples $S$
**Ensure:** Posterior means $\{\hat{\mu}_m\}$ and credible intervals $\{\text{CI}_m\}$
1: Compute kernel matrices $K_{W^{(1)}}, K_{V^{(1)}}, K_{V^{(2)}}, K_{V^{(2)}V^{(1)}}, K_{V^{(1)}V^{(2)}}, K_{W^{(1)}V^{(1)}}$
2: Approximate $\tilde{K}_{V^{(1)}}$ using $(v_s)_{s=1}^S \sim \nu$: $\tilde{K}_{V^{(1)}} \leftarrow S^{-1} K_{V^{(1)}V_s} K_{V^{(1)}V_s}^\top$, where $[K_{V^{(1)}V_s}]_{i,s} := k_V(V_i^{(1)}, v_s)$
3: **for** $m = 1$ **to** $M$ **do**
4:     *Compute posterior mean and variance components*
5:     $\boldsymbol{k}^{(1)}(w_m) \leftarrow [k_W(w_m, W_1^{(1)}), \ldots, k_W(w_m, W_{m_1}^{(1)})]^\top$
6:     $D^{(1)}(w_m) \leftarrow \text{diag}(\boldsymbol{k}^{(1)}(w_m))$
7:     $\boldsymbol{\alpha}^{(1)}(w_m) \leftarrow D^{(1)}(w_m)(K_{W^{(1)}V^{(1)}} + \sigma^2 I)^{-1} \boldsymbol{Y}^{(1)}$
8:     $A^{(1)}(w_m) \leftarrow D^{(1)}(w_m)(K_{W^{(1)}V^{(1)}} + \sigma^2 I)^{-1} D^{(1)}(w_m)$
9:     $\boldsymbol{k}^{(2)}(z_m) \leftarrow [k_Z(z_m, Z_1^{(2)}), \ldots, k_Z(z_m, Z_{m_2}^{(2)})]^\top$
10:    $\boldsymbol{\beta}^{(2)}(z_m) \leftarrow (K_{Z^{(2)}} + \eta^2 I)^{-1} \boldsymbol{k}^{(2)}(z_m)$
11:    $\hat{k}^{(2)}(z_m, z_m) \leftarrow k_Z(z_m, z_m) - \boldsymbol{k}^{(2)}(z_m)^\top \boldsymbol{\beta}^{(2)}(z_m)$
12:    *Compute posterior moments of $\gamma(w_m, z_m)$*
13:    $\hat{\mu}_m \leftarrow \boldsymbol{\beta}^{(2)}(z_m)^\top K_{V^{(2)}V^{(1)}} \boldsymbol{\alpha}^{(1)}(w_m)$
14:    $S_1 \leftarrow \boldsymbol{\beta}^{(2)}(z_m)^\top K_{V^{(2)}} \boldsymbol{\beta}^{(2)}(z_m) k_W(w_m, w_m) - \boldsymbol{\beta}^{(2)}(z_m)^\top K_{V^{(2)}V^{(1)}} A^{(1)}(w_m) K_{V^{(1)}V^{(2)}} \boldsymbol{\beta}^{(2)}(z_m)$
15:    $S_2 \leftarrow \hat{k}^{(2)}(z_m, z_m) \text{Tr}\Big[\tilde{K}_{V^{(1)}} \big\{ \boldsymbol{\alpha}^{(1)}(w_m) \boldsymbol{\alpha}^{(1)}(w_m)^\top - A^{(1)}(w_m) \big\}\Big]$
16:    $S_3 \leftarrow \left(\sum_{i \in I} \lambda_i\right) k_W(w_m, w_m) \hat{k}^{(2)}(z_m, z_m)$
17:    $\hat{\sigma}_m^2 \leftarrow S_1 + S_2 + S_3$
18:    *Compute credible interval*
19:    $c \leftarrow \text{CDF}_{\mathcal{N}(0,1)}^{-1}\left(\frac{1+\alpha}{2}\right)$
20:    $\text{CI}_m \leftarrow [\hat{\mu}_m \pm c\,\hat{\sigma}_m]$
21: **end for**

---

## A.4. IMPSpec Posterior Approximation Error Induced by Surrogate Likelihood

Here we study how mis-specification of the surrogate observation model in Eq. (11) of the main text,

$$\phi_{V,i}(V) = \mu_i(Z) + \xi_i, \qquad \xi_i \sim N(0, \eta^2),$$

can affect the posterior approximation error for the pointwise causal quantity $\gamma(w, z)$.

### A.4.1. SET-UP.

Throughout, we fix a realized dataset $D = \{(V_i, Z_i, W_i, Y_i)\}_{i=1}^n$. All statements below are conditional on this observed dataset, so the following analysis is finite-sample rather than asymptotic.

**Causal Effect of Interest.** Using the notation of the main paper, recall that the causal effects of interest can be expressed as

$$\gamma(w, z) = f\big(\psi_W(w) \otimes \mu(z)\big),$$

where

$$\mu(z) := \mathbb{E}[\psi_V(V) \mid Z = z] \in \mathcal{H}_V$$

is the conditional mean embedding of $V$ given $Z$. By the Mercer expansion of $\psi_V(v) = k_V(\cdot, v)$, this admits the spectral representation

$$\mu(z) = \sum_{i \in I} \lambda_{V,i} \mu_i(z) \phi_{V,i}(\cdot), \qquad \mu_i(z) := \mathbb{E}[\phi_{V,i}(V) \mid Z = z].$$

Thus, for each fixed $(w, z)$, the scalar quantity $\gamma(w, z)$ is determined by the latent pair $(f, \mu)$. We therefore define

$$T_{w,z}(f, \mu) := f\big(\psi_W(w) \otimes \mu(z)\big),$$

so that $\gamma(w, z) = T_{w,z}(f, \mu)$.

**Probabilistic Model.** Let $\Pi$ denote the prior distribution of the latent pair $(f, \mu)$ induced by the GP priors in the main text. Given fixed $D$, let $\hat{L}(f, \mu) := \hat{p}(D|f, \mu)$ denote the likelihood induced by the surrogate observation model in Eq. (11), and let $L(f, \mu)$ denote a *well-specified*[8] benchmark likelihood. The purpose of the following derivation is to quantify the posterior mis-specification error induced by replacing the well-specified likelihood $L$ with the surrogate likelihood $\hat{L}$.

Define the corresponding posteriors

$$P(df, d\mu) := \frac{L(f, \mu)\Pi(df, d\mu)}{Z}, \qquad \hat{P}(df, d\mu) := \frac{\hat{L}(f, \mu)\Pi(df, d\mu)}{\hat{Z}}.$$

with normalizing constants $Z := \int L(f, \mu)\,\Pi(df, d\mu)$, $\hat{Z} := \int \hat{L}(f, \mu)\,\Pi(df, d\mu)$, assumed finite and strictly positive. Now, define the likelihood ratio,

$$R(f, \mu) := \frac{\hat{L}(f, \mu)}{L(f, \mu)}.$$

and note that we can express the normalizing constant ratio as follows

$$\frac{\hat{Z}}{Z} = \int \frac{\hat{L}(f, \mu)}{L(f, \mu)}\,P(df, d\mu) = \int R(f, \mu)\,P(df, d\mu) = \mathbb{E}_P[R].$$

Hence $\hat{P} \ll P$ and the Radon-Nikodym derivative can be expressed in terms of $R$.

$$\frac{d\hat{P}}{dP}(f, \mu) = \frac{R(f, \mu)}{\mathbb{E}_P[R]}.$$

Since the dataset $D$ is fixed throughout, the quantities $L(f, \mu), \hat{L}(f, \mu), R(f, \mu)$ are treated as deterministic (measurable) functions of the latent pair $(f, \mu)$. Expectations such as $\mathbb{E}_P[R]$ are therefore taken with respect to the posterior $P(df, d\mu)$ on $(f, \mu)$, i.e., for any measurable function $g \in L^1(P)$,

$$\mathbb{E}_P[g] := \int g(f, \mu)\,P(df, d\mu).$$

The above identities let us bound divergences/metrics of interest in terms of functionals of $R$,

$$\chi^2(\hat{P}\|P) := \int \left(\frac{d\hat{P}}{dP} - 1\right)^2 dP = \mathbb{E}_P\left[\left(\frac{R}{\mathbb{E}_P[R]} - 1\right)^2\right].$$

$$\text{TV}(P, \hat{P}) = \frac{1}{2}\int \left|\frac{d\hat{P}}{dP} - 1\right| dP = \frac{1}{2}\mathbb{E}_P\left[\left|\frac{R}{\mathbb{E}_P[R]} - 1\right|\right].$$

In what follows, we will use these identities to bound how likelihood differences propagate to posterior differences.

A.4.2. JOINT POSTERIOR BOUNDS UNDER LIKELIHOOD-RATIO MOMENT CONTROL

Assume the likelihood ratio satisfies the following integrated perturbation bound for some $\varepsilon > 0$:

$$\frac{\mathbb{E}_P[R^2]}{\mathbb{E}_P[R]^2} \leq 1 + \varepsilon.$$

This condition can be used to control the posterior divergences above. Indeed, since

$$\frac{d\hat{P}}{dP}(f, \mu) = \frac{R(f, \mu)}{\mathbb{E}_P[R]},$$

---

[8] By this we mean that there exists a latent pair $(f^\star, \mu^\star)$ such that the true conditional distribution of the observed data is exactly $L(\cdot \mid f^\star, \mu^\star)$. In contrast, $\hat{L}$ is the surrogate likelihood induced by Eq. (11), and need not coincide with the true conditional distribution.

we have the exact identity

$$\chi^2(\hat{P}\|P) = \int \left(\frac{d\hat{P}}{dP} - 1\right)^2 dP = \mathbb{E}_P\left[\left(\frac{R}{\mathbb{E}_P[R]} - 1\right)^2\right] = \frac{\mathbb{E}_P[R^2]}{\mathbb{E}_P[R]^2} - 1.$$

Hence the assumption implies

$$\chi^2(\hat{P}\|P) \le \varepsilon.$$

Moreover, using the standard bound $\mathrm{TV}(\hat{P}, P) \le \frac{1}{2}\sqrt{\chi^2(\hat{P}\|P)}$, we obtain

$$\mathrm{TV}(\hat{P}, P) \le \tfrac{1}{2}\sqrt{\varepsilon}.$$

### A.4.3. PROPAGATION TO POSTERIOR OF CAUSAL EFFECT $\gamma(w, z)$

The above control on the posterior over latent processes propagates automatically to the posterior of the causal effect $\gamma(w, z)$. In particular, let

$$P_{w,z} := P \circ T_{w,z}^{-1}, \qquad \hat{P}_{w,z} := \hat{P} \circ T_{w,z}^{-1}$$

denote the induced posteriors of the scalar quantity $\gamma_{w,z} = \gamma(w, z)$ under $P$ and $\hat{P}$.

For total variation, we immediately get the contraction bound

$$\mathrm{TV}(P_{w,z}, \hat{P}_{w,z}) = \sup_{B \in \mathcal{B}(\mathbb{R})} |P(T_{w,z}^{-1}(B)) - \hat{P}(T_{w,z}^{-1}(B))| \le \sup_{A \in \mathcal{F}} |P(A) - \hat{P}(A)| = \mathrm{TV}(P, \hat{P}).$$

Since $\chi^2$ is an $f$-divergences, the data-processing inequality for the measurable map $T_{w,z}$ also gives

$$\chi^2(\hat{P}_{w,z}\|P_{w,z}) \le \chi^2(\hat{P}\|P).$$

Combining this with the previous bounds yields

$$\chi^2(\hat{P}_{w,z}\|P_{w,z}) \le \varepsilon, \qquad \mathrm{TV}(P_{w,z}, \hat{P}_{w,z}) \le \tfrac{1}{2}\sqrt{\varepsilon}. \tag{24}$$

**Remark on Usefulness of Bounds Under Small/Large $\varepsilon$.** The strongest global consequence of the integrated likelihood-perturbation assumption is the chi-squared bound

$$\chi^2(\hat{P}_{w,z}\|P_{w,z}) \le \varepsilon.$$

Unlike total variation, this remains nontrivial even outside the small-misspecification regime. By contrast, the corresponding total-variation bound

$$\mathrm{TV}(P_{w,z}, \hat{P}_{w,z}) \le \tfrac{1}{2}\sqrt{\varepsilon}$$

is useful only when $\varepsilon < 4$, since TV is always bounded by 1.

Let $\tilde{P}_{w,z}$ denote the Gaussian moment-matching approximation to the surrogate posterior $\hat{P}_{w,z}$ used in the main text to construct credible intervals from its posterior moments. Then, whenever the above total-variation bound is nontrivial, the triangle inequality gives the decomposition

$$\mathrm{TV}(P_{w,z}, \tilde{P}_{w,z}) \le \mathrm{TV}(P_{w,z}, \hat{P}_{w,z}) + \mathrm{TV}(\hat{P}_{w,z}, \tilde{P}_{w,z}) \le \tfrac{1}{2}\sqrt{\varepsilon} + \mathrm{TV}(\hat{P}_{w,z}, \tilde{P}_{w,z}).$$

Thus the total posterior error separates into two contributions: the first is the error induced by replacing the well-specified benchmark likelihood with the surrogate likelihood, while the second is the error introduced by the Gaussian approximation step itself.

### A.4.4. EFFECT ON POSTERIOR MOMENTS

Since the Gaussian approximation used in the main text is matched to the posterior mean and covariance of the surrogate posterior, it is natural to study how the above likelihood error can affect these moments. Below we derive bounds that are most useful when likelihood error is relatively small (e.g., $\epsilon < 1$).

**Bound based on $\chi^2$.** For brevity, define $\delta_\varepsilon := \sqrt{\varepsilon}$.

Now suppose $\gamma_{w,z} \in L^4(P_{w,z})$. Then, since $\int \left( \frac{d\hat{P}_{w,z}}{dP_{w,z}} - 1 \right) dP_{w,z} = 0$, for any measurable $h \in L^2(P_{w,z})$

$$
\mathbb{E}_{\hat{P}_{w,z}}[h(\gamma_{w,z})] - \mathbb{E}_{P_{w,z}}[h(\gamma_{w,z})] = \int h(x) \left( \frac{d\hat{P}_{w,z}}{dP_{w,z}}(x) - 1 \right) P_{w,z}(dx)
$$

$$
= \int \left( h(x) - \mathbb{E}_{P_{w,z}}[h(\gamma_{w,z})] \right) \left( \frac{d\hat{P}_{w,z}}{dP_{w,z}}(x) - 1 \right) P_{w,z}(dx),
$$

Now, by Cauchy–Schwarz,

$$
\left| \mathbb{E}_{\hat{P}_{w,z}}[h(\gamma_{w,z})] - \mathbb{E}_{P_{w,z}}[h(\gamma_{w,z})] \right| \leq \sqrt{\chi^2(\hat{P}_{w,z}\|P_{w,z})} \sqrt{\mathrm{Var}_{P_{w,z}}(h(\gamma_{w,z}))}.
$$

Using the bound on $\chi^2(\hat{P}_{w,z}\|P_{w,z})$, this gives

$$
\left| \mathbb{E}_{\hat{P}_{w,z}}[h(\gamma_{w,z})] - \mathbb{E}_{P_{w,z}}[h(\gamma_{w,z})] \right| \leq \delta_\varepsilon \sqrt{\mathrm{Var}_{P_{w,z}}(h(\gamma_{w,z}))}.
$$

Applying this with $h(x) = x$ and then $h(x) = x^2$ yields the first and second moment bounds

$$
\left| \mathbb{E}_{\hat{P}_{w,z}}[\gamma_{w,z}] - \mathbb{E}_{P_{w,z}}[\gamma_{w,z}] \right| \leq \delta_\varepsilon \sqrt{\mathrm{Var}_{P_{w,z}}(\gamma_{w,z})}. \tag{25}
$$

$$
\left| \mathbb{E}_{\hat{P}_{w,z}}[\gamma_{w,z}^2] - \mathbb{E}_{P_{w,z}}[\gamma_{w,z}^2] \right| \leq \delta_\varepsilon \sqrt{\mathrm{Var}_{P_{w,z}}(\gamma_{w,z}^2)}.
$$

Finally, writing

$$
\Delta_1 := \mathbb{E}_{\hat{P}_{w,z}}[\gamma_{w,z}] - \mathbb{E}_{P_{w,z}}[\gamma_{w,z}], \qquad \Delta_2 := \mathbb{E}_{\hat{P}_{w,z}}[\gamma_{w,z}^2] - \mathbb{E}_{P_{w,z}}[\gamma_{w,z}^2],
$$

we have

$$
\left| \mathrm{Var}_{\hat{P}_{w,z}}(\gamma_{w,z}) - \mathrm{Var}_{P_{w,z}}(\gamma_{w,z}) \right| = \left| \Delta_2 - \left( (\mathbb{E}_{\hat{P}_{w,z}}[\gamma_{w,z}])^2 - (\mathbb{E}_{P_{w,z}}[\gamma_{w,z}])^2 \right) \right|
$$

$$
\leq |\Delta_2| + |\Delta_1| \left( |\mathbb{E}_{\hat{P}_{w,z}}[\gamma_{w,z}]| + |\mathbb{E}_{P_{w,z}}[\gamma_{w,z}]| \right)
$$

$$
\leq |\Delta_2| + |\Delta_1| \left( 2|\mathbb{E}_{P_{w,z}}[\gamma_{w,z}]| + |\Delta_1| \right),
$$

since $|\mathbb{E}_{\hat{P}_{w,z}}[\gamma_{w,z}]| \leq |\mathbb{E}_{P_{w,z}}[\gamma_{w,z}]| + |\Delta_1|$. Substituting the bounds on $\Delta_1$ and $\Delta_2$ gives

$$
\left| \mathrm{Var}_{\hat{P}_{w,z}}(\gamma_{w,z}) - \mathrm{Var}_{P_{w,z}}(\gamma_{w,z}) \right| \leq \delta_\varepsilon \sqrt{\mathrm{Var}_{P_{w,z}}(\gamma_{w,z}^2)} + 2|\mathbb{E}_{P_{w,z}}[\gamma_{w,z}]| \, \delta_\varepsilon \sqrt{\mathrm{Var}_{P_{w,z}}(\gamma_{w,z})} + \delta_\varepsilon^2 \mathrm{Var}_{P_{w,z}}(\gamma_{w,z}).
$$
$$\tag{26}$$

## A.5. IMPSPEC Equivalence of Posterior Mean Functions and Kernel Ridge Regression

We now show that the posterior means of the Gaussian process (GP) priors used for $f$ and $\mu$ in IMPSPEC coincide exactly with the kernel ridge regression (KRR) estimators of the corresponding conditional expectations. This establishes that our GP formulation provides a fully Bayesian extension of the original kernel estimator introduced in Singh et al. (2024). We note this equivalence for the scalar-valued GP case (e.g., for $f$) has already been established in Kanagawa et al. (2018).

**Notation.** Throughout this section we use the following notation for Gram matrices and kernel evaluation vectors. Let $\{(Y_i, W_i, V_i, Z_i)\}_{i=1}^n \sim \mathbb{P}_{Y,W,V,Z}^{\otimes n}$ denote the relevant training samples.

- $K_W \in \mathbb{R}^{n \times n}$ and $K_V \in \mathbb{R}^{n \times n}$ are the Gram matrices of $k_W$ and $k_V$, respectively:

$$
[K_W]_{ij} := k_W(W_i, W_j), \qquad [K_V]_{ij} := k_V(V_i, V_j).
$$

- $K_{WV} \in \mathbb{R}^{n \times n}$ is the Gram matrix of the product kernel $k_W \otimes k_V$:

$$[K_{WV}]_{ij} := k_W(W_i, W_j) \, k_V(V_i, V_j) = \langle \psi_W(W_i) \otimes \psi_V(V_i), \, \psi_W(W_j) \otimes \psi_V(V_j) \rangle.$$

- $\boldsymbol{k}_W(w) \in \mathbb{R}^n$ and $\boldsymbol{k}_V(v) \in \mathbb{R}^n$ are the kernel evaluation vectors at test inputs $w \in \mathcal{W}$ and $v \in \mathcal{V}$:

$$[\boldsymbol{k}_W(w)]_i := k_W(w, W_i), \qquad [\boldsymbol{k}_V(v)]_i := k_V(v, V_i).$$

- $\boldsymbol{k}_Z(z) \in \mathbb{R}^n$ is the kernel evaluation vector for $k_Z$ at $z \in \mathcal{Z}$:

$$[\boldsymbol{k}_Z(z)]_i := k_Z(z, Z_i).$$

**Posterior mean for $f$**    Recall from (10) in the main text that $Y = f(\psi_W(W) \otimes \psi_V(V)) + U, \, U \sim \mathcal{N}(0, \sigma^2)$. As we show in the proof of Theorem 4.1 (i.e., by a simple application of Lemma C.3), under the prior $f \sim \mathcal{GP}(0, \langle \bullet, \bullet \rangle_{\mathcal{H}_W \otimes \mathcal{H}_V})$, the posterior mean of $f$ conditional on data $\mathcal{D}_n^{(1)} = \{(Y_i, \psi_W(W_i), \psi_V(V_i))\}_{i=1}^n$ is

$$\hat{f}(\cdot) = \Phi(\cdot)^\top (K_{WV} + \sigma^2 I)^{-1} \boldsymbol{Y}, \qquad \Phi(\cdot) := [\psi_W(W_i) \otimes \psi_V(V_i)]_{i=1}^n \in (\mathcal{H}_W \otimes \mathcal{H}_V)^{1 \times n}.$$

so that, by the reproducing property $k_X(x, x') = \langle \psi_X(x), \psi_X(x') \rangle$ for $x \in \{w, v\}$, we have

$$\hat{f}(\psi_W(w) \otimes \psi_V(v)) = (\boldsymbol{k}_W(w) \odot \boldsymbol{k}_V(v))^\top (K_{WV} + \sigma^2 I)^{-1} \boldsymbol{Y}$$

On the other hand, it is a standard fact that the kernel ridge regression (KRR) estimator for $f$ minimizing

$$\min_{f \in \mathcal{H}_W \otimes \mathcal{H}_V} \frac{1}{n} \sum_{i=1}^n (Y_i - \langle f, \psi_W(W_i) \otimes \psi_V(V_i) \rangle_{\mathcal{H}_W \otimes \mathcal{H}_V})^2 + \lambda \|f\|_{\mathcal{H}_W \otimes \mathcal{H}_V}^2$$

has closed form

$$\hat{f}_{\mathrm{KRR}}(\cdot) = \Phi(\cdot)^\top (K_{WV} + n\lambda I)^{-1} \boldsymbol{Y}.$$

where again by the reproducing property

$$\hat{f}_{\mathrm{KRR}}(\psi_W(w) \otimes \psi_V(v)) = (\boldsymbol{k}_W(w) \odot \boldsymbol{k}_V(v))^\top (K_{WV} + n\lambda I)^{-1} \boldsymbol{Y}$$

Setting $\lambda = \sigma^2/n$ shows the equivalence $\hat{f}_{\mathrm{KRR}} = \hat{f}$, i.e. the GP posterior mean exactly reproduces the KRR estimator for $f$, when evaluating both functions on features of $(w, v)$.

**Posterior mean for $\mu$**    Analogously, recall from (11) in the main text that

$$\phi_{V,i}(V) = \mu_i(Z) + \xi_i, \qquad \xi_i \sim \mathcal{N}(0, \eta^2),$$

where each $\mu_i \sim \mathcal{GP}(0, k_Z)$ and $\{\phi_{V,i}\}_{i \in I}$ are the Mercer eigenfunctions of $k_V$ with eigenvalues $\{\lambda_{V,i}\}_{i \in I}$. Recall also from (6) in the main text that vector-valued function $\mu = \mathbb{E}[\psi_V(V)|Z = \bullet]$ is parameterized by the coordinates $\{\mu_i\}_{i \in I}$ (where $I \subseteq \mathbb{N}$) via

$$\mu(z) = \sum_{i \in I} \lambda_{V,i} \, \mu_i(z) \, \phi_{V,i} = \sum_{i \in I} \sqrt{\lambda_{V,i}} \, \mu_i(z) \, e_{V,i}.$$

See Assumption A.4 for the full assumptions regarding Mercer's theorem in the present setting.

For each fixed spectral index $i$, define the observation vector $\phi_{V,i}(\boldsymbol{V}) := (\phi_{V,i}(V_1), \ldots, \phi_{V,i}(V_n))^\top$. With independent priors $\mu_i \sim \mathcal{GP}(0, k_Z)$ and noise variance $\eta^2$. Since the model (11) and prior on $\mu_i$ corresponds to a standard GP regression of $\phi_{V,i}(V)$ on $Z$ with Gaussian noise, and the noise variables and GP priors are independent across $i$, we get the standard form of the GP posterior mean in this setting,

$$\hat{\mu}_i(z) = \phi_{V,i}(\boldsymbol{V})^\top (K_Z + \eta^2 I)^{-1} \boldsymbol{k}_Z(z).$$

Therefore using the fact that $k_V(v, v') = \sum_{i \in I} \lambda_{V,i} \, \phi_{V,i}(v) \, \phi_{V,i}(v')$, the posterior mean of $\mu(z)$ is

$$\hat{\mu}(z) = \sum_{i \in I} \lambda_{V,i} \, \hat{\mu}_i(z) \, \phi_{V,i} = \sum_{j=1}^n \alpha_j(z) \, \psi_V(V_j) = \Phi_V \boldsymbol{\alpha}(z),$$

where $\boldsymbol{\alpha}(z) := (K_Z + \eta^2 I)^{-1} \boldsymbol{k}_Z(z)$ and $\Phi_V := (\psi_V(V_1), \ldots, \psi_V(V_n)) \in \mathcal{H}_V^{1 \times n}$.

Now, consider doing vector-valued KRR with separable operator-valued kernel $\Gamma(z, z') = k_Z(z, z') I_{\mathcal{H}_V}$:

$$\min_{\mu \in \mathcal{H}_Z \otimes \mathcal{H}_V} \frac{1}{n} \sum_{j=1}^{n} \|\psi_V(V_j) - \mu(Z_j)\|_{\mathcal{H}_V}^2 + \lambda \|\mu\|_{\mathcal{H}_Z \otimes \mathcal{H}_V}^2.$$

By the vector-valued representer theorem the estimator is given as follows (see (Grünewälder et al., 2012)),

$$\hat{\mu}_{\mathrm{KRR}}(z) = \Phi_V (K_Z + n\lambda I)^{-1} \boldsymbol{k}_Z(z).$$

Setting $\lambda = \eta^2/n$ yields $\hat{\mu}_{\mathrm{KRR}} = \hat{\mu}$.

As a result, the posterior mean of the overall causal function $\gamma(w, z) = f(\psi_W(w) \otimes \mu(z))$ reduces to the plug-in kernel estimator of Singh et al. (2024), as demonstrated in the main text. The GP formulation therefore preserves the same point estimator while additionally providing posterior uncertainty via the covariance terms derived in Theorem 4.1.

### A.6. Limitations of BayesIMP

Here we provide additional analysis on the limitations of the kernel constructions used in Chau et al. (2021b), supporting our claims in Section 6 and Section 7.

#### A.6.1. NONSTATIONARITY AND UNDERFITTING

In the Toy Example in Section 7 we observe that BayesIMP's posterior mean systematically underfits the true causal function $\mathbb{E}[Y|\mathrm{do}(A = a)]$. Here we demonstrate what we believe to be the root of the problem: the nonstationarity of the kernel constructions used. For $f : \mathcal{X} \to \mathbb{R}$ (here we let $\mathcal{X} = \mathcal{V} \times \mathcal{W}$ for convenience), they use the following kernel

$$r(x, x') = \int_{\mathcal{X}} k(x, t) k(t, x') \, d\nu(t) = \langle k(x, \bullet), k(x', \bullet) \rangle_{L^2(\nu)}.$$

where $\nu$ is a finite, non-degenerate measure on $\mathcal{X}$. Since $r$ satisfies a certain *'nuclear-dominance'* property w.r.t. $k$ (see Lukić & Beder (2001); Flaxman et al. (2016)), a GP $f \sim \mathcal{GP}(0, r)$ is known to lie in the RKHS $\mathcal{H}_X$ with feature map $\psi_X(x) = k(\bullet, x)$. When $\nu$ is a probability measure concentrated around a location $m$ (e.g. a Gaussian $\mathcal{N}(m, \varsigma^2 I)$, as used in Flaxman et al. (2016); Chau et al. (2021b)) and $k$ is radial, $k(x, t) = h(\|x - t\|)$, the kernel $r$ is *nonstationary*: it depends on both $x - x'$ and on the location of $x, x'$ relative to $m$. Unfortunately, the nature of this non-stationarity has undesirable consequences: the "basis function" $x' \mapsto r(x, x')$ shrinks to zero as $x$ moves away from $m$. Below we formalize this effect, and further quantify the rate of decay under specific choices of kernel $k$ and measure $\nu$.

**Proposition A.10** (Uniform tail limit for the nuclear-dominant kernel basis function). *Let $k : \mathbb{R}^d \times \mathbb{R}^d \to [0, 1]$ be a stationary, positive-definite kernel of the radial form $k(x, t) = h(\|x - t\|)$, where $h : [0, \infty) \to [0, 1]$ is nonincreasing and satisfies $\lim_{r \to \infty} h(r) = 0$. Let $\nu$ be a finite Borel measure on $\mathbb{R}^d$ and define*

$$r(x, x') = \int k(x, t) k(t, x') \, d\nu(t). \tag{27}$$

*Then, for every $x \in \mathbb{R}^d$,*

$$\sup_{x' \in \mathbb{R}^d} r(x, x') \leq \int k(x, t) \, d\nu(t) \tag{28}$$

*and consequently $\sup_{x' \in \mathbb{R}^d} r(x, x') \xrightarrow[\|x - m\| \to \infty]{} 0$, for any fixed $m \in \mathbb{R}^d$.*

**Proposition A.11** (Decay rate for log-convex, integrable $h$ and density of $\nu$). *Under the conditions of Proposition A.10, assume further that (i) $\log h$ is convex, (ii) $\int_{\mathbb{R}^d} h(x) dx < \infty$ and (iii) $\nu$ admits density $p$ w.r.t. Lebesgue measure*

$$p(t) = C_p \, h(\|t - m\|),$$

*where $m \in \mathbb{R}^d$ is the mode and $C_p > 0$ is the normalizing constant. Then, for all $x \in \mathbb{R}^d$,*

$$\sup_{x' \in \mathbb{R}^d} r(x, x') \leq h(\tfrac{1}{2}\|x - m\|). \tag{29}$$

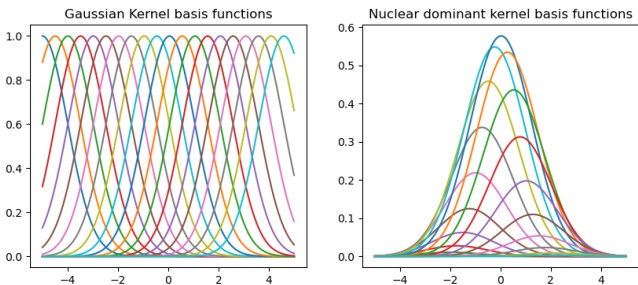

*Figure 7.* Basis functions of the Gaussian kernel $k(x, \bullet) = \exp(-\|x - \bullet\|)$ (left) and nuclear dominant kernel $r(x, \bullet) = \int k(x,t)k(\bullet,t)\mathcal{N}(dt|0,1)$ (right) for $x$ grid-spaced on $\{-5...,5\}$. $r(x, \bullet)$ collapses exponentially fast as $|x| \to \infty$.

*Remark* A.12. The log-convexity and integrability assumptions on $h$ is satisfied by popular stationary kernels such as the Gaussian kernel, Laplace kernel, and Cauchy kernel.

Proposition A.10 and Proposition A.11 show that the entire basis function $x' \mapsto r(x, x')$ collapses in magnitude as $\|x - m\| \to \infty$, and when $\nu$ is chosen of the same form as $k$ and $h$ satisfies the log-convexity assumption, the rate of decay is given by the rate of tail decay of the kernel itself. Figure 7 demonstrates the decay for the Gaussian kernel $k(x, x') = \exp(-\|x - x'\|)$. This may induce location-dependent shrinkage and systematic underfitting in KRR/GP fits, particularly when evaluating away from $m$.

Tuning the variance $\varsigma$ of the density of $\nu$ can mitigate but not remove the effect; it vanishes only in the limit of translation-invariant (improper) $\nu$, which is not finite on $\mathbb{R}^d$ (finiteness is required to enforce the nuclear dominance property). In experiments, we tried treating $\varsigma$ as an additional hyperparameter in the Toy Example and optimizing the marginal likelihood derived in BayesIMP. Whilst this improved performance (see Appendix B), it did not fully resolve the underfitting problem.

### A.6.2. FINITE DIMENSIONAL APPROXIMATION AND VARIANCE COLLAPSE

In the synthetic benchmark in Section 7, we also observed that BayesIMP's posterior variance collapsed out-of- distribution. Here we show that this occurs as a consequence of the finite dimensional GP approximations they use for tractability. Abstracting away from this particular experiment, the posterior variance of any causal effect $\gamma(w, z)$ of the form Equation (1) for BayesIMP is given by

$$\mathbb{V}ar[\gamma(w,z)|\mathcal{X}^n] := \mathbb{V}ar\langle f, \psi(w) \otimes \mu(z)\rangle_{\mathcal{H}_W \otimes \mathcal{H}_V}, \quad f \sim \mathcal{GP}(\hat{m}_f, \hat{r}_f), \quad \mu(z) \sim \mathcal{GP}(\hat{m}_z, \hat{r}_z) \tag{30}$$

where now $\hat{m}_f, \hat{r}_f$ are posterior functions trained on datasets $\mathcal{D}_1 = (Y_i^{(1)}, W_i^{(1)}, V_i^{(1)})_{i=1}^n$ and $\hat{m}_z, \hat{r}_z$ are posterior functions trained on $\mathcal{D}_2 = (W_i^{(2)}, V_i^{(2)}, Z_i^{(2)})_{i=1}^m$ respectively. Note, Chau et al. (2021b) focus on the setting where $\mathcal{D}_1$ and $\mathcal{D}_2$ are disjoint sets of samples (see Appendix A.3 for more details on this set-up), hence why we emphasize the distinction between these datasets. When all observations arise from a single shared dataset, we have $n = m$ and $(W_i^{(1)}, V_i^{(1)}, Z_i^{(1)}) = (W_i^{(2)}, V_i^{(2)}, Z_i^{(2)}) \, \forall i \in \{1, ..., n\}$. As this variance is not tractable in closed form given the bespoke nuclear-dominant kernels used for $r_f, r_z$, Chau et al. (2021b) replace $f$ and $\mu(z)$ with the finite dimensional approximations that retain the true posterior mean

$$\hat{f} = \sum_{i=1}^{n+m} a_i k_{W,V}((W_i, V_i), \bullet), \quad \boldsymbol{a} \sim \mathcal{N}(\boldsymbol{m}_a, C_a)$$

$$\hat{\mu}(z) = \sum_{i=1}^{n+m} b_i(z) k_V(V_i, \bullet), \quad \boldsymbol{b}(z) \sim \mathcal{N}(\boldsymbol{m}_b(z), C_b(z))$$

where now $(W_i, V_i)_{i=1}^{n+m}$ are all observations concatenated from $\mathcal{D}_1 \cup \mathcal{D}_2$ (in the single dataset case $m = 0$) and $\boldsymbol{m}_a = (K_W \odot K_V)^{-1}\hat{\boldsymbol{m}}_f, C_a = (K_W \odot K_V)^{-1}\hat{R}_f(K_W \odot K_V)^{-1}, \boldsymbol{m}_b(z) = K_V^{-1}\hat{\boldsymbol{m}}_z, C_b(z) = K_V^{-1}\hat{R}_z K_V^{-1}$. Here $\hat{\boldsymbol{m}}_f, \hat{\boldsymbol{m}}_z$, $\hat{R}_f, \hat{R}_z$ are the posterior means and covariances of the true $f, \mu(z)$ evaluated on $(W_i, V_i)_{i=n+1}^{n+m}$. With these approximations,

Equation (30) is then estimated as

$$\widehat{\mathbb{V}ar}\langle f, \psi(w) \otimes \mu(z)\rangle_{\mathcal{H}_W \otimes \mathcal{H}_V} = \mathbb{V}ar[\boldsymbol{a}^\top D(w) K_{VV} \boldsymbol{b}(z)]$$
$$= \boldsymbol{m}_a^\top D(w) K_V C_b(z) K_V D(w) \boldsymbol{m}_a + \boldsymbol{m}_b(z)^\top D(w) K_V C_a K_{VV} D(w) \boldsymbol{m}_b(z) + Tr[C_a D(w) K_{VV} C_b(z) K_V D(w)]$$
$$= \boldsymbol{k}_W(w)^\top \Theta \boldsymbol{k}_W(w)$$

where $D(w) = \mathrm{diag}(k_W(w, W_1), \ldots, k_W(w, W_{n+m}))$, $\boldsymbol{k}_W(w) = [k_W(w, W_1), \ldots, k_W(w, W_{n+m})]^\top$ and $\Theta \in \mathbb{R}^{(n+m)\times(n+m)}$. The last line follows from the general facts

$$u^\top D A D u = d^\top (A \odot uu^\top) d \quad \text{and} \quad \mathrm{Tr}(CDAD) = d^\top (A \odot C^\top) d,$$

with $D = \mathrm{diag}(d)$ and $d = \boldsymbol{k}_W(w)$, applied termwise to the three summands.

For any radial kernel $k_W$ (e.g. Gaussian, Cauchy, Exponential), the $i$th entry of $\boldsymbol{k}_W(w)$ decays to 0 as $\|w - W_i\| \to \infty$. Since each term in $\widehat{\mathbb{V}ar}\langle f, \psi(w) \otimes \mu(z)\rangle$ is quadratic in $\boldsymbol{k}_W(w)$, the overall variance decays roughly according to the *square of the kernel decay rate*. For example, with a Gaussian kernel $k_W(w, w') = \exp(-\|w - w'\|^2)$, the posterior variance vanishes at a Gaussian-squared rate $\sim \exp(-2\|w\|^2)$ as $w$ moves out of the support of the data. This leads to extreme model overconfidence in those regions. This behaviour can be avoided for certain causal estimands where $W = \emptyset$ (e.g., when estimating ATE in the causal data fusion setting discuss in Appendix A.3), but in most standard single-dataset and fusion settings $W \neq \emptyset$, so this pathology is unavoidable.

# B. Experiments

## B.1. Evaluation Metrics

We define the metrics used to evaluate methods in the synthetic experiments. Let $\mathcal{X}_r^n$, $r = 1, \ldots, R$, denote the dataset used in the $r$-th independent trial, drawn from the repeated-sampling distribution $\mathbb{P}_{\mathcal{X}^n}$. In the causal data fusion setting, $\mathcal{X}_r^n$ denotes the union of the datasets available in that trial, e.g. $\mathcal{X}_r^n = \mathcal{D}_{1,r} \cup \mathcal{D}_{2,r}$. Let $\Theta = \{\theta_j\}_{j=1}^J$ denote the grid of causal-effect evaluation points used in a given experiment, where $\theta_j$ may be an intervention value or an intervention–conditioning pair. For example, $\theta_j = a_j$ in the toy ATE experiment, while $\theta_j = (d_j, b)$ in the synthetic CATE benchmark.

For each $\theta_j$, let $\gamma(\theta_j)$ denote the true causal effect. In the synthetic experiments, $\gamma$ is known from the data-generating structural causal model and is evaluated independently of the training datasets. When a closed-form expression is not used, we approximate it by high-accuracy Monte Carlo simulation from the interventional distribution,

$$\gamma(\theta_j) \approx \frac{1}{L} \sum_{\ell=1}^L Y_\ell^{\mathrm{do}(\theta_j)},$$

with $L$ chosen large enough that this Monte Carlo error is negligible relative to the reported experimental variation. In the toy experiment, for instance, we estimate $\gamma(a) = \mathbb{E}[Y \mid \mathrm{do}(A = a)]$ by drawing $10^5$ Monte Carlo samples from the interventional distribution for each $a$ on the evaluation grid. These ground-truth values are used only for evaluation. They are distinct from the plug-in estimates $\hat{\gamma}$ used inside the calibration procedure in Algorithm 2.

Let $\hat{\mu}_r(\theta_j)$ be the posterior mean produced by a method on trial $r$, and let

$$C_{r,j,\alpha} = [L_{r,j,\alpha}, U_{r,j,\alpha}]$$

denote the central $\alpha$-credible interval, or confidence interval for frequentist baselines, for $\gamma(\theta_j)$ on trial $r$, where $\alpha \in (0, 1)$.

**Root mean squared error.** For each trial, we compute the root mean squared error of the posterior mean,

$$\mathrm{RMSE}_r = \left\{ \frac{1}{J} \sum_{j=1}^J \left(\hat{\mu}_r(\theta_j) - \gamma(\theta_j)\right)^2 \right\}^{1/2}.$$

Tables report the mean and standard deviation of $\mathrm{RMSE}_r$ across trials.

**Calibration error.** A central $\alpha$-credible interval is calibrated at $\theta_j$ if it covers the true causal effect with probability $\alpha$ over repeated draws of the dataset:

$$\mathbb{P}_{\mathcal{X}^n}\left(\gamma(\theta_j) \in C_{\theta_j, \alpha}(\mathcal{X}^n)\right) = \alpha.$$

Thus, for a fixed evaluation point $\theta_j$ and nominal level $\alpha$, the calibration error is

$$\Delta_{j,\alpha} = \left|\mathbb{P}_{\mathcal{X}^n}\left(\gamma(\theta_j) \in C_{\theta_j, \alpha}(\mathcal{X}^n)\right) - \alpha\right|.$$

Since $\mathbb{P}_{\mathcal{X}^n}$ is unknown, we estimate it using the empirical distribution over trials,

$$\widehat{\mathbb{P}}_{\mathcal{X}^n} = \frac{1}{R}\sum_{r=1}^{R}\delta_{\mathcal{X}_r^n}.$$

This gives the empirical coverage

$$\widehat{\mathrm{Cov}}_{j,\alpha} = \frac{1}{R}\sum_{r=1}^{R}\mathbb{1}\left\{\gamma(\theta_j) \in C_{r,j,\alpha}\right\},$$

and the empirical calibration error

$$\widehat{\Delta}_{j,\alpha} = \left|\widehat{\mathrm{Cov}}_{j,\alpha} - \alpha\right|.$$

For calibration-error profiles, such as those shown in the main text, we compute the average calibration error separately for each fixed nominal level $\alpha$:

$$\widehat{\mathrm{CalErr}}(\alpha) = \frac{1}{J}\sum_{j=1}^{J}\widehat{\Delta}_{j,\alpha}.$$

In our experiments, these profiles are computed over a grid of 100 nominal levels $\alpha \in (0, 1)$. For scalar summaries in tables, we average over a grid $\mathcal{A}$ of nominal levels:

$$\widehat{\mathrm{CalErr}} = \frac{1}{|\mathcal{A}|}\sum_{\alpha \in \mathcal{A}}\widehat{\mathrm{CalErr}}(\alpha) = \frac{1}{J|\mathcal{A}|}\sum_{j=1}^{J}\sum_{\alpha \in \mathcal{A}}\left|\widehat{\mathrm{Cov}}_{j,\alpha} - \alpha\right|.$$

This is the normalized version of the calibration objective in Equation (23); the normalization does not affect comparisons between methods.

We also report marginal calibration curves, which plot

$$\widehat{\mathrm{MCov}}(\alpha) = \frac{1}{RJ}\sum_{r=1}^{R}\sum_{j=1}^{J}\mathbb{1}\left\{\gamma(\theta_j) \in C_{r,j,\alpha}\right\}$$

against the nominal level $\alpha$. Marginal calibration curves are standard in the calibration literature and summarize average empirical coverage. However, unlike the calibration-error profile $\widehat{\mathrm{CalErr}}(\alpha)$, they can appear close to the diagonal even when a method is miscalibrated at individual evaluation points $\theta_j$, because over-coverage and under-coverage can cancel after averaging over $\theta_j$.

**Bootstrap uncertainty for calibration summaries.** For each method, the trials yield a single empirical calibration profile and a single marginal calibration curve. To construct approximate confidence intervals around these curves, we use the empirical bootstrap over trials. Specifically, each bootstrap replication samples $R$ datasets with replacement from

$$\widehat{\mathbb{P}}_{\mathcal{X}^n} = \frac{1}{R}\sum_{r=1}^{R}\delta_{\mathcal{X}_r^n},$$

and recomputes the calibration-error profile and marginal calibration curve on the resampled trials. In the reported experiments we use 100 bootstrap replications, each consisting of $R = 50$ sampled trials.

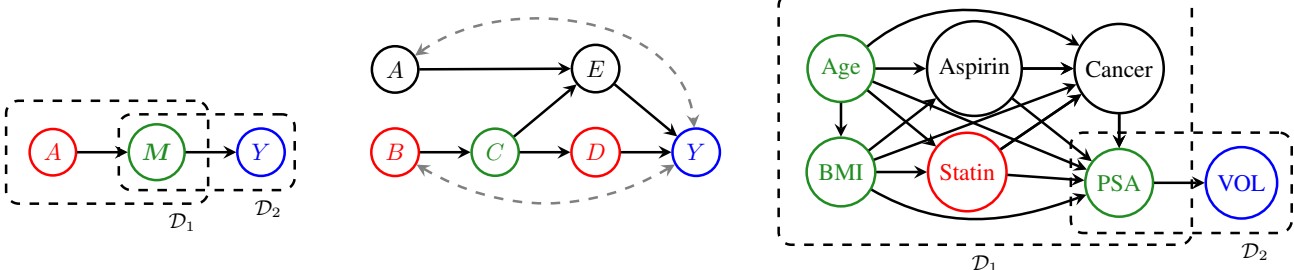

*Figure 8.* Left: Causal graph used in Toy Example where we aim to identify the average effect of $A$ on $Y$. Middle: Synthetic benchmark from Aglietti et al. (2020) where we aim to identify different causal effects of $B$ and $D$ on $Y$ with unobserved confounding between $A \leftrightarrow Y$ and $B \leftrightarrow Y$. Right: Healthcare Example from Chau et al. (2021b) where we aim to find the optimal level of statin dose to reduce cancer volume. $\mathcal{D}_1$ and $\mathcal{D}_2$ indicate that we only have access to datasets of observations in those subsets. Blue = outcomes, red = interventions, green = required observables for identifiability.

**Interval score.** Calibration alone does not penalize unnecessarily wide intervals. We therefore also report interval scores, which reward narrow intervals while penalizing intervals that fail to cover the true causal effect. For an interval $C_{r,j,\alpha} = [L_{r,j,\alpha}, U_{r,j,\alpha}]$, the interval score is

$$\text{IS}_{r,j,\alpha} = U_{r,j,\alpha} - L_{r,j,\alpha} + \frac{2}{1-\alpha}\big(L_{r,j,\alpha} - \gamma(\theta_j)\big)\mathbb{1}\{\gamma(\theta_j) < L_{r,j,\alpha}\} + \frac{2}{1-\alpha}\big(\gamma(\theta_j) - U_{r,j,\alpha}\big)\mathbb{1}\{\gamma(\theta_j) > U_{r,j,\alpha}\}.$$

We report the average interval score

$$\widehat{\text{IS}}_\alpha = \frac{1}{RJ}\sum_{r=1}^{R}\sum_{j=1}^{J}\text{IS}_{r,j,\alpha}.$$

In particular, $\text{IS}_{90}$ and $\text{IS}_{95}$ denote interval scores computed with $\alpha = 0.90$ and $\alpha = 0.95$, respectively. As with calibration error, uncertainty estimates for interval scores are obtained by bootstrapping over trials.

## B.2. Toy Example

In this experiment we aim to estimate the ATE $\mathbb{E}[Y|\text{do}(A = a)]$ for the left causal graph in Figure 8.

### B.2.1. SIMULATION DETAILS

The data generating process is given by

$$A \sim \mathcal{U}(0,1) \tag{31}$$

$$M_d|A \sim \mathcal{N}(\sin(\alpha_d A), \sigma_d^2), \quad d \in \{1,...,5\} \tag{32}$$

$$Y|\boldsymbol{M} \sim \mathcal{N}(\boldsymbol{\beta}^\top \sin(\boldsymbol{M}), \sigma_y^2) \tag{33}$$

Where $\boldsymbol{\alpha} = 10 \times [1, 1.75, 2.5, 3.25, 4]$, $\boldsymbol{\beta} = 1/[1, 2, 3, 4, 5]$ and $(\sigma_d^2)_{d=1}^5$, $\sigma_y$ are set so that the signal to noise ratios are 2:1.

For each simulation we draw two i.i.d. datasets $(Y_i^{(1)}, \boldsymbol{M}_i^{(1)}, A_i^{(1)})_{i=1}^n$, $(Y_j^{(2)}, \boldsymbol{M}_j^{(2)}, A_j^{(2)})_{j=1}^n$ of size $n = 100$, and assume we only have access to the sets of observation pairs $\mathcal{D}_1 = (Y_i^{(1)}, \boldsymbol{M}_i^{(1)})_{i=1}^n$ and $\mathcal{D}_2 = (\boldsymbol{M}_j^{(2)}, A_j^{(2)})_{j=1}^n$.

### B.2.2. CAUSAL EFFECT IDENTIFICATION

Using the rules of do-calculus (Pearl, 2009b), we have

$$\mathbb{E}[Y|\text{do}(A = a)] = \int_{\mathcal{M}} \mathbb{E}[Y|\text{do}(A = a), M = m]\mathbb{P}_{M|\text{do}(A)}(dm|a) \tag{34}$$

$$= \int_{\mathcal{M}} \mathbb{E}[Y|\text{do}(A = a), M = m]\mathbb{P}_{M|A}(dm|a) \quad (Y \perp\!\!\!\perp A \in \mathcal{G}_{\underline{A}}) \tag{35}$$

$$= \int_{\mathcal{M}} \mathbb{E}[Y|M = m]\mathbb{P}_{M|A}(dm|a) \quad (Y \perp\!\!\!\perp A|M \in \mathcal{G}_{\overline{A}}) \tag{36}$$

### B.2.3. IMPLEMENTATION DETAILS

**True causal function:** The true causal function is estimated by drawing $10^5$ Monte Carlo samples from $\mathbb{P}_{Y|A}(\bullet|A = a)$ and averaging. We do this for $a \in \hat{\mathcal{A}}$, where $\hat{\mathcal{A}}$ is a uniformly spaced grid on $[0, 1]$ of size 100.

**IMPspec:** Our estimation target is $\gamma(w, z)$ as defined in Equation (1) in the main text, where $(W, V, Z) := (\emptyset, \boldsymbol{M}, A)$ (i.e., so that $\gamma(w, a) = \gamma(a)$). Since we are in the causal data fusion setting (i.e., we only observe observation pairs $(Y, \boldsymbol{M})$ and $(M, A)$) and $W = \emptyset$), we estimate the posterior mean and variance of $\gamma(z)$ using the modification of Theorem 4.1 for this case (see Appendix A.3 for details). The posterior credible intervals are constructed using the procedure outlined in Section 4.2. The exact algorithm used is the modification of Algorithm 3 (for $W = \emptyset$) discussed in Remark A.9 in Appendix A.3.

All kernels are set as Gaussian kernels with per-dimension lengthscales. The kernel parameters and noise variances are optimized using the marginal likelihoods in Section 5.1. We use 1000 iterations of ADAM with a base learning rate of 0.1. We implement our method both with and without our spectral calibration procedure Algorithm 2, to do an ablation of this effect on calibration performance. When not using this procedure, we fix $\nu = \mathcal{N}(\bar{\boldsymbol{M}}, \hat{S})$, where $\hat{S} = diag(\hat{\mathbb{V}}ar(M_1)^{\frac{1}{2}}, ..., \hat{\mathbb{V}}ar(M_5)^{\frac{1}{2}})$ and $\bar{\boldsymbol{M}} \approx 0$ is the empirical mean. When using this procedure, we set $\nu = \mathcal{N}(\bar{\boldsymbol{M}}, \omega\hat{S})$ and choose the optimal $\omega \in 2^{\{-4,-2,0,2,4\}}$ that minimises the calibration error Equation (23) for $\alpha \in \Gamma$, and $a \in \hat{\mathcal{A}}$, where $\Gamma$ is a uniformly spaced grid on $[0, 1]$ of size 101. We estimate Equation (23) using a 50:50 sample split (i.e. $n = 50$ observations from each dataset are used to estimate $\mathbb{P}_{\mathcal{X}^{n/2}}$ via $\mathbb{P}_{\hat{\mathcal{X}}^{n/2}|\mathcal{X}^{n/2}}$ and the remaining $n = 50$ observations from each dataset are used to estimate $\hat{\gamma}(a)$ via our posterior mean (recall IMPspec's posterior mean is equivalent to the original estimator in Singh et al. (2024) up to hyperparameter equivalence. We use 20 bootstrap replications to estimate $\mathbb{P}_{\hat{\mathcal{X}}^{n/2}|\mathcal{X}^{n/2}}$.

**BayesIMP:** We implement BayesIMP using the implementation in Proposition 5 in Chau et al. (2021b)[9]. The kernel $k_A$ are set as Gaussian kernels with per-dimension lengthscales, and the (nuclear dominant) kernel $r_M$ for the GP prior on $f$ is parameterized using their implementation:

$$r_M = \int k_M(\bullet, m)k_M(\bullet, m)\mathcal{N}(0, \hat{S})[dt] \tag{37}$$

where recall that this construction is used to ensure samples of $f$ can be realized as elements of the RKHS $\mathcal{H}_M$ associated with base kernel $k_M$. We estimate the base kernel parameters and noise hyperparameters using (i) the marginal likelihood $p(\boldsymbol{y}|\boldsymbol{m}_1, ..., \boldsymbol{m}_d) = \mathcal{N}(\boldsymbol{0}, R_M + \sigma^2 I)$ and (ii) the marginal likelihood for $\langle\mathbb{E}[\psi(\boldsymbol{M})|A], \psi(\boldsymbol{m})\rangle_{\mathcal{H}_M}$ specified in their Proposition 3. We use 1000 iterations of ADAM with a base learning rate of 0.1. Since they do not provide details on how to set $\hat{S}$, we set $\hat{S}$ identically to IMPspec.

**Sampling GP:** The baseline sampling GP we implement is essentially the method in Witty et al. (2020) in the case where there is no unobserved confounding. In particular, in this case their approach specifies $Y = f(M) + U, M_d = g_d(A) + V_d$ where $U \sim \mathcal{N}(0, \sigma^2), V_d \sim \mathcal{N}(0, \eta_d^2)$, $f \sim \mathcal{GP}(0, k_M)$ and $g_d \sim \mathcal{GP}(0, k_{A,d})$. The kernel parameters and noise variances are trained by maximising the usual marginal likelihoods for each respective dataset: $p(\boldsymbol{y}|\boldsymbol{m}_1, ..., \boldsymbol{m}_d) = \mathcal{N}(\boldsymbol{y}|\boldsymbol{0}, K_M + \sigma^2 I)$ and $\prod_{j=1}^d p(\boldsymbol{m}_j|\boldsymbol{a}) = \prod_j \mathcal{N}(\boldsymbol{m}_j|\boldsymbol{0}, K_{A,d} + \eta_d^2 I)$. We use 1000 iterations of ADAM with a base learning rate of 0.1. The causal effect is then estimated by sampling from the posterior predictive distributions $\prod_{j=1}^d p(\boldsymbol{m}_j|\boldsymbol{a}, \mathcal{D}_2), p(\boldsymbol{y}|\boldsymbol{m}_1, ..., \boldsymbol{m}_d, \mathcal{D}_1)$, which are again given by the usual formulae (Williams & Rasmussen, 2006). We use Gaussian kernels for $k_M$ and each $k_{A,d}$, with per-dimension lengthscales.

### B.2.4. ADDITIONAL ABLATIONS

We report additional ablations of IMPspec and BayesIMP on the toy example. Unless stated otherwise, all results are averaged over 50 trials and use the same hyperparameter and training settings as reported above.

**Effect of sample splitting.** We first analyze the effect of sample splitting in IMPspec's spectral calibration procedure. In the main experiments, we use separate data splits to estimate $\hat{\gamma}$ and to estimate the empirical distribution $\hat{\mathbb{P}}_{\mathcal{X}^n}$ appearing in the calibration objective (23). Table 5 shows that calibration performance worsens when this sample split is removed. One possible explanation is that using the same data for both terms induces dependence between $\hat{\gamma}$ and $\hat{\mathbb{P}}_{\mathcal{X}^n}$. Since the

---

[9]We found two typos in Proposition 5, which when removed significantly improved the performance of their method. All presented results for BayesIMP are for the case where these typos are fixed.

uncalibrated uncertainty estimates are slightly overconfident in this experiment, this dependence may cause the calibration procedure to select a spectral representation with posterior variance that remains too small.

**BayesIMP ablations.** We also run two ablations for BayesIMP. The first quantifies the effect of optimizing the marginal likelihood $\log p(\boldsymbol{y}|\boldsymbol{m}_1, \ldots, \boldsymbol{m}_d)$ over the variance parameter $\varsigma$ of the integrating measure $\mathcal{N}(0, \varsigma\hat{S})$ used to construct the nuclear-dominating kernel $r_M$; see (37). This parameter controls the nonstationarity of $r_M$, which we argued in Appendix A.6.1 can lead to systematic underfitting. The second ablation quantifies the effect of Monte Carlo approximating $r_M$ during training and inference. Except for special choices of base kernel $k_M$, the kernel $r_M$ is not available in closed form, so such an approximation is required in general. During training, we use $m = \alpha n$ Monte Carlo samples with $\alpha = 1$, which ensures that the resulting Gram matrix $R_M$ is invertible. At test time, where these matrices are computed only once, we use $10^5$ Monte Carlo samples.

Table 5 shows that optimizing the integrating-measure variance improves BayesIMP, but does not fully resolve the underfitting problem relative to IMPspec. Monte Carlo approximating $r_M$ further degrades performance, highlighting an additional practical limitation of BayesIMP. In contrast, IMPspec maintains lower RMSE and calibration error, with sample splitting providing a further improvement in calibration.

*Table 5.* Toy-example ablations. Results are mean $\pm$ standard deviation over 50 trials. BayesIMP-approx uses a Monte Carlo approximation to the nuclear-dominating kernel $r_M$; BayesIMP-opt optimizes the variance of the integrating measure used to construct $r_M$; IMPspec-nosplit removes sample splitting from the calibration procedure.

| Method | RMSE | Calibration error |
|---|---|---|
| BayesIMP-approx | $0.366 \pm 0.156$ | $0.136 \pm 0.015$ |
| BayesIMP | $0.314 \pm 0.082$ | $0.143 \pm 0.009$ |
| BayesIMP-opt | $0.269 \pm 0.086$ | $0.094 \pm 0.010$ |
| IMPspec-nosplit (ours) | $\mathbf{0.223} \pm 0.046$ | $0.087 \pm 0.009$ |
| IMPspec (ours) | $\mathbf{0.223} \pm 0.046$ | $\mathbf{0.065} \pm 0.008$ |

**Effect of number of bootstrap replications.** We next study sensitivity to the number of bootstrap replications $B$ used by the calibration procedure. Table 6 shows that RMSE, calibration error, and wall-clock time are relatively stable across the range considered. This suggests that, in this experiment, only a modest number of bootstrap replications is sufficient to obtain a useful calibration signal.

*Table 6.* Sensitivity to the number of bootstrap replications used in calibration. Results are averaged over 50 trials on NVIDIA Quadro P5000 and RTX A4500 GPUs.

| | Bootstrap replications | | | |
|---|---|---|---|---|
| Metric | $B = 10$ | $B = 20$ | $B = 50$ | $B = 100$ |
| RMSE | $0.23 \pm 0.01$ | $0.23 \pm 0.01$ | $0.23 \pm 0.01$ | $0.23 \pm 0.01$ |
| Calibration error | $0.07 \pm 0.01$ | $0.08 \pm 0.01$ | $0.08 \pm 0.01$ | $0.07 \pm 0.01$ |
| Wall-clock time (s) | $19.11 \pm 1.72$ | $16.10 \pm 1.27$ | $20.10 \pm 1.94$ | $19.03 \pm 1.65$ |

**Effect of kernel choice.** Finally, we test whether IMPspec's performance is specific to the Gaussian kernel used in the main experiments. Table 7 compares the Gaussian kernel to a more flexible $\gamma$-exponential kernel family. The two choices give similar calibration error after calibration, with only a small increase in RMSE for the $\gamma$-exponential kernel. This suggests that the performance of IMPspec is not especially sensitive to this kernel choice in the toy example.

*Table 7.* Kernel ablation in the toy example. We compare the Gaussian kernel used in the main experiments with a $\gamma$-exponential kernel.

| Method | Kernel | RMSE | Calibration error |
|---|---|---|---|
| IMPspec-nocal | Gaussian | $0.223 \pm 0.046$ | $0.098 \pm 0.010$ |
| IMPspec-cal | Gaussian | $0.223 \pm 0.046$ | $0.065 \pm 0.008$ |
| IMPspec-nocal | $\gamma$-Exp | $0.236 \pm 0.006$ | $0.088 \pm 0.010$ |
| IMPspec-cal | $\gamma$-Exp | $0.238 \pm 0.006$ | $0.067 \pm 0.008$ |

**Robustness to noise misspecification and limited support.** We also evaluate all methods under two more challenging variants of the toy example. First, to test robustness to noise misspecification, we replace the Gaussian outcome noise with heavy-tailed Student-$t$ noise. Second, to test performance under limited support coverage, we draw treatments from a skewed $\text{Beta}(3, 1/2)$ distribution, which concentrates mass near one end of the treatment domain and makes extrapolation across the full range more difficult. Table 8 reports RMSE, calibration error, and 90%/95% interval scores. IMPspec remains best calibrated in both settings and achieves the best interval scores after calibration, although under limited support coverage Sampling-GP obtains slightly lower RMSE.

*Table 8.* Robustness to noise misspecification and limited support coverage in the toy example. Results are mean $\pm$ standard deviation over 50 trials. The heavy-tailed design replaces Gaussian outcome noise with Student-$t_{\nu=3}$ noise, while the limited-support design draws treatments from a skewed $\text{Beta}(3, 1/2)$ distribution. IMPspec-cal achieves the best calibration error and interval scores in both settings.

| Design | Method | RMSE | Cal. error | IS 90% | IS 95% |
|---|---|---|---|---|---|
| Heavy-tailed noise | Sampling-GP | $0.37 \pm 0.01$ | $0.16 \pm 0.09$ | $2.41 \pm 0.20$ | $3.73 \pm 0.42$ |
| | BayesIMP | $0.58 \pm 0.01$ | $0.35 \pm 0.17$ | $6.43 \pm 0.34$ | $11.41 \pm 0.69$ |
| | IMPspec-nocal | $0.35 \pm 0.02$ | $0.15 \pm 0.09$ | $2.31 \pm 0.26$ | $3.47 \pm 0.50$ |
| | IMPspec-cal | $\mathbf{0.35} \pm 0.02$ | $\mathbf{0.10} \pm 0.04$ | $\mathbf{2.02} \pm 0.22$ | $\mathbf{2.84} \pm 0.41$ |
| Limited support | Sampling-GP | $\mathbf{0.32} \pm 0.01$ | $0.10 \pm 0.05$ | $1.40 \pm 0.06$ | $1.74 \pm 0.10$ |
| | BayesIMP | $0.45 \pm 0.02$ | $0.19 \pm 0.12$ | $2.92 \pm 0.31$ | $4.47 \pm 0.61$ |
| | IMPspec-nocal | $0.35 \pm 0.01$ | $0.08 \pm 0.05$ | $1.87 \pm 0.12$ | $2.61 \pm 0.21$ |
| | IMPspec-cal | $0.36 \pm 0.01$ | $\mathbf{0.06} \pm 0.04$ | $\mathbf{1.43} \pm 0.07$ | $\mathbf{1.70} \pm 0.10$ |

## B.3. Synthetic Benchmark

In this experiment we aim to estimate the CATE $\mathbb{E}[Y|\text{do}(D = d), B = b]$ for the middle causal graph in Figure 8.

### B.3.1. SIMULATION DETAILS

We use the same data generating process as in Aglietti et al. (2020), which is:

$$U_1 = \epsilon_1 \tag{38}$$
$$U_2 = \epsilon_2 \tag{39}$$
$$F = \epsilon_3 \tag{40}$$
$$A = F^2 + U_1 + \epsilon_A \tag{41}$$
$$B = U_2 + \epsilon_B \tag{42}$$
$$C = \exp(-B) + \epsilon_C \tag{43}$$
$$D = \exp(-C)/10 + \epsilon_D \tag{44}$$
$$E = \cos(A) + C/10\epsilon_E \tag{45}$$
$$Y = \cos(D) + \sin(E) + U_1 + U_2\epsilon_Y \tag{46}$$

where all noise variables $\epsilon_\bullet \sim \mathcal{N}(0, 1)$.

For each simulation we draw a single dataset $(A_i, B_i, C_i, D_i, E_i, Y_i)_{i=1}^n$ of size $n = 100$.

### B.3.2. CAUSAL EFFECT IDENTIFICATION

Using the rule of do-calculus, we have

$$\mathbb{E}[Y|\mathrm{do}(D=d), B=b] = \int_{\mathcal{C}} \mathbb{E}[Y|\mathrm{do}(D=d), B=b, C=c]\mathbb{P}_{C|\mathrm{do}(D),B}(dc|d,b) \tag{47}$$

$$= \int_{\mathcal{C}} \mathbb{E}[Y|\mathrm{do}(D=d), B=b, C=c]\mathbb{P}_{C|B}(dc|b) \qquad (C \perp\!\!\!\perp D|B \in \mathcal{G}_{\overline{D}}) \tag{48}$$

$$= \int_{\mathcal{C}} \mathbb{E}[Y|D=d, B=b, C=c]\mathbb{P}_{C|B}(dc|b) \qquad (Y \perp\!\!\!\perp D|B, C \in \mathcal{G}_{\underline{D}}) \tag{49}$$

### B.3.3. IMPLEMENTATION DETAILS

**True causal function:** Since $\mathbb{E}[Y|D=d, B=b, C=c] = \cos(d) + \mathbb{E}[\sin(E)|C=c] = \mathbb{E}[Y|D=d, C=c]$, we can estimate $\mathbb{E}[Y|\mathrm{do}(D=d), B=b]$ for each pair $(d,b)$ as follows. We first sample $10^5$ Monte Carlo samples from $\mathbb{P}_{C|B}(\bullet|b)$ and $P(A)$. This gives samples $(A_i, C_i)_{i=1}^{10^5}$ from $\mathbb{P}_{A,C|\mathrm{do}(D=d),B=b}$. We then use each sample to sample from $\mathbb{P}_{E|C,A}$. This gives samples $(E_i)_{i=1}^{10^5}$ from $\mathbb{P}_{E|\mathrm{do}(D=d),B=b}$. Averaging $\cos(d) + \sin(E)$ w.r.t. these samples therefore estimates $\hat{\mathbb{E}}[Y|\mathrm{do}(D=d), B=b]$.

**IMPspec:** Our estimation target is $\gamma(w, z)$ as defined in Equation (1) in the main text, where now $(W, V, Z) := ((D, B), C, B)$. We therefore estimate (i) posterior moments of $\gamma$ using Theorem 4.1 in the main text, credible intervals of $\gamma$ using Algorithm 1 in the main text, and (iii) optimize posterior calibration using Algorithm 2 in the main text. The kernels used, training, and calibration settings are all identical to the Toy Example.

**BayesIMP:** We implement BayesIMP (Chau et al., 2021b) using their equations in Proposition 5 extended for the case where $W \neq \emptyset$ in Equation (1) in the main text. This extension is derived in Appendix A. Note that their derivations are for the causal data fusion setting discussed in Appendix A.3. However, their method can be used in the single dataset setting $\mathcal{D}_1 = \mathcal{D}_2$ (in this case the concatenation of the datasets that they use becomes $\mathcal{D}_1 \cup \mathcal{D}_2 = \mathcal{D}_1 = \mathcal{D}_2$). The choice of kernels as well as the training and inference details are the same as in the Toy Example.

**Sampling GP:** The sampling GP from Witty et al. (2020) specifies $Y = f(B, C, D) + U$ and $C = g(B) + V$, with $f, g, U, V$ specified as in the Toy Example. The choice of kernels as well as the training and inference details are the same as in the Toy Example.

**Continuous DR:** We implement the continuous-treatment doubly robust estimator of Kennedy et al. (2017). Since this estimator targets marginal continuous-treatment dose-response curves, rather than conditional dose-response functions, we give it a favorable adaptation to the fixed-$B = 0$ benchmark. Specifically, for this baseline we train on observations from the fixed-$B = 0$ subproblem, i.e. samples from $P(\mathrm{Data} \mid B = 0)$, so that the target reduces to the marginal dose-response curve

$$d \mapsto \mathbb{E}[Y \mid \mathrm{do}(D=d), B = 0].$$

This removes the need for Continuous DR to learn the conditioning on $B$, and therefore gives it an advantage relative to methods that directly estimate the full conditional causal function.

Using the notation in Kennedy et al. (2017), we set the continuous treatment to $T = D$, the adjustment covariate to $L = C$, and the outcome to $Y$. The nuisance components are estimated using flexible regression models: a stacked learner with linear regression, decision trees, random forests, gradient boosting, ridge regression, and lasso for the treatment regression and conditional treatment variance, and a random forest for the outcome regression $\hat{\mu}(l, t) \approx \mathbb{E}[Y \mid L = l, T = t]$. Conditional treatment densities are estimated from standardized treatment residuals using a Gaussian kernel density estimator, with bandwidth selected by five-fold cross-validation.

Given these nuisance estimates, we form the doubly robust pseudo-outcome

$$\hat{\xi}_i = \frac{Y_i - \hat{\mu}(L_i, T_i)}{\hat{\pi}(T_i \mid L_i)/\hat{\varpi}(T_i)} + \hat{m}(T_i),$$

where $\hat{\pi}$ and $\hat{\varpi}$ denote the estimated conditional and marginal treatment densities, and $\hat{m}$ is the marginal outcome-regression curve. The final dose-response estimate is obtained by local-linear kernel regression of $\hat{\xi}_i$ on $T_i$, with bandwidth chosen by

leave-one-out cross-validation. Pointwise confidence intervals use the corresponding sandwich-style standard error estimate and Gaussian critical values. All reported results use 50 independent trials with $n = 100$ observations per trial and 100 intervention grid points.

**Orthogonal RF:** We implement orthogonal random forests using the `DMLOrthoForest` estimator from `EconML`. ORF is based on a partially linear treatment-effect model of the form

$$Y = g(X, W) + \theta(X)T + \epsilon,$$

where $T$ is the treatment, $X$ contains the heterogeneity features, and $W$ contains additional controls. Thus ORF estimates the heterogeneous linear treatment-effect slope $\theta(X)$, rather than an unrestricted nonlinear dose-response curve.

For the back-door synthetic benchmark, we set $T = D$, $X = B$, $W = C$, and the outcome to $Y$. Under the ORF model, the implied fixed-$B = 0$ response curve has the form

$$d \mapsto \mathbb{E}[g(0, C) \mid B = 0] + \theta(0)d,$$

so ORF provides the slope $\hat{\theta}(0)$ but not the intercept needed to put the estimate on the same response-curve scale as the other methods. We therefore reconstruct the ORF curve as

$$\hat{m}_{\mathrm{ORF}}(d) = \hat{c} + d\,\hat{\theta}(0),$$

where the intercept is chosen as the MSE-optimal vertical offset on the evaluation grid,

$$\hat{c} = \frac{1}{J} \sum_{j=1}^{J} \left\{ \gamma(d_j, 0) - d_j \hat{\theta}(0) \right\}.$$

This oracle offset is favorable to ORF, since it removes vertical bias in the reconstructed curve; however, the resulting estimate remains restricted to the linear-in-$d$ form implied by the ORF model.

The treatment and outcome nuisance regressions in the first stage are random forests with 200 trees and minimum leaf size 10. The final local nuisance models are fit using `LassoCV` with three-fold cross-validation. For the orthogonal forest itself, we use 500 trees, minimum leaf size 10, maximum depth 10, and subsampling ratio 0.7. We use `inference="auto"` to obtain pointwise intervals for $\theta(0)$. Pointwise intervals for the reconstructed response curve are obtained by transforming the ORF slope intervals through the same affine map $d \mapsto \hat{c} + d\theta$, swapping endpoints when $d < 0$. Calibration and interval-score metrics are then computed from these transformed intervals as described in Appendix B.1.

### B.4. Causal Bayesian Optimization Tasks

**Primer on Causal Bayesian optimization:** In this experiment we use our method to construct a GP prior for Bayesian optimization (BO) of several causal effects of interest. In general, in causal BO (CBO) (Aglietti et al., 2020), one aims to find $\mathrm{argmin}_{a \in \mathcal{A}} \mathbb{E}[Y|\mathrm{do}(A = a)]$ or $\mathrm{argmax}_{a \in \mathcal{A}} \mathbb{E}[Y|\mathrm{do}(A = a)]$ by querying as few values of the treatment as possible. At the start, one typically has no interventional data, and so has to construct a GP prior for $\mathbb{E}[Y|\mathrm{do}(A = \bullet)] \sim \mathcal{GP}(m, k)$ (either naively or by using observational data). Each iteration recovers an observation $(\hat{x}, \mathbb{E}[Y|\mathrm{do}(X = \hat{x})])$, which is then used to condition the GP prior on. Note that in practice to recover this observation one would have to run a large scale experiment under the intervention $\mathrm{do}(A = \hat{a})$, and return an estimated $\mathbb{E}[Y|\mathrm{do}(A = \hat{a})]$ using a very large empirical average. At each iteration the GP is used to construct an acquisition function. This function determines the next optimal treatment value $a$ by using the posterior uncertainty estimates to optimally trade off exploration and exploitation. GPs are popular surrogates for BO tasks as they are flexible, non-parametric models with good generalisation and uncertainty quantification properties (Williams & Rasmussen, 2006); enable acquisition functions to be computed in closed form; and are efficient with their use of the data (Shahriari et al., 2015).

In our case, we aim to find the optimal intervention that (i) minimises the CATE $\mathbb{E}[Y|\mathrm{do}(D = d), B = b]$ in the synthetic benchmark (middle causal graph in Figure 8), (ii) maximises the ATT $\mathbb{E}[Y|\mathrm{do}(B = b), B = b']$ in the synthetic benchmark (middle causal graph in Figure 8), and (iii) minimises the ATE $\mathbb{E}[\mathrm{VOL}|\mathrm{do}(\mathrm{Statin})]$ in a real Healthcare example with access to partial datasets (see right causal graph in 8). We search for the optimal intervention level, whilst fixing $B = 0$ for the CATE and ATT in the synthetic benchmark. We note that despite the ATT being considered a counterfactual quantity, it is a 'single-world' counterfactual quantity, and can be estimated experimentally (Richardson & Robins, 2013).

B.4.1. SIMULATION DETAILS

**Synthetic benchmark:** The data generating process for the synthetic benchmark is already described above. For the task of minimising the CATE $\mathbb{E}[Y|\text{do}(D=d), B=b]$ w.r.t. $D$ we draw datasets of size $n=100$, and for the task of maximising the ATT $\mathbb{E}[Y|\text{do}(B=b), B=b']$ we draw dataset sizes of $n=500$.

**Healthcare example:** For the first dataset $\mathcal{D}_1$ (see Figure 8), we use the same data generating process as in Chau et al. (2021b):

$$\text{age} = \mathcal{U}[15, 75] \tag{50}$$
$$\text{bmi} = \mathcal{N}(27 - 0.01 \cdot \text{age}, 0.7) \tag{51}$$
$$\text{aspirin} = \sigma(-8.0 + 0.1 \cdot \text{age} + 0.03 \cdot \text{bmi}) \tag{52}$$
$$\text{statin} = \sigma(-13 + 0.1 \cdot \text{age} + 0.2 \cdot \text{bmi}) \tag{53}$$
$$\text{cancer} = \sigma(2.2 - 0.05 \cdot \text{age} + 0.01 \cdot \text{bmi} - 0.04\text{statin} + 0.02 \cdot \text{aspirin}) \tag{54}$$
$$\text{PSA} = \mathcal{N}(6.8 + 0.04 \cdot \text{age} - 0.15 \cdot \text{bmi} - 0.6 \cdot \text{statin} + 0.55 \cdot \text{aspirin} + \text{cancer}, 0.4) \tag{55}$$

and for each simulation we draw $n=100$ observations. For the second dataset $\mathcal{D}_2$ (see Figure 8), we did not have access to the original data used in (Chau et al., 2021b), but we still were able to construct a simulator based on real data. In particular, we use the estimated linear model (VOL $= \hat{\beta}\text{PSA}$) of the effect of prostate specific antigen (PSA) on cancer volume (VOL) in Carvalhal et al. (2010) as a simulator, by setting VOL $= \hat{\beta}\text{PSA} + U$, where $U \sim \mathcal{N}(0, \sigma^2)$ with $\sigma^2$ set to match the estimated $R^2 = 0.13$ in their study. For each simulation we draw $n=100$ observations for $\mathcal{D}_2$ using this model (note we use the data generating process above to first draw observations for PSA).

B.4.2. CAUSAL EFFECT IDENTIFICATION

We have already derived the identification formulae for the CATE $\mathbb{E}[Y|\text{do}(D=d), B=b]$ in the synthetic benchmark using the back-door criterion. The ATT $\mathbb{E}[Y|\text{do}(B=b), B=b']$ in the synthetic benchmark can be identified using the do calculus on a counterfactual graph which has an added node $B'$ with the same parents as $B$ but all outgoing edges removed (see Theorem 1 in (Shpitser & Pearl, 2009)). We call this graph $\mathcal{G}'$.

$$\mathbb{E}[Y|\text{do}(B=b), B=b'] = \int_{\mathcal{C}} \mathbb{E}[Y|\text{do}(B=b), B'=b', C=c]\mathbb{P}_{C|\text{do}(B), B'}(dc|b, b') \tag{56}$$
$$= \int_{\mathcal{C}} \mathbb{E}[Y|\text{do}(B=b), B'=b', C=c]\mathbb{P}_{C|\text{do}(B)}(dc|b) \qquad (C \perp\!\!\!\perp B' | B \in \mathcal{G}'_{\overline{B}}) \tag{57}$$
$$= \int_{\mathcal{C}} \mathbb{E}[Y|\text{do}(B=b), B'=b', C=c]\mathbb{P}_{C|B}(dc|b) \qquad (C \perp\!\!\!\perp B \in \mathcal{G}'_{\underline{B}}) \tag{58}$$
$$= \int_{\mathcal{C}} \mathbb{E}[Y|B'=b', C=c]\mathbb{P}_{C|B}(dc|b) \qquad (Y \perp\!\!\!\perp B | B', C \in \mathcal{G}'_{\overline{B}}) \tag{59}$$
$$= \int_{\mathcal{C}} \mathbb{E}[Y|B=b', C=c]\mathbb{P}_{C|B}(dc|b) \tag{60}$$

Lastly, the ATE $\mathbb{E}[\text{Vol}|\text{do}(\text{Statin})]$ for the healthcare example can be identified as follows

$$\mathbb{E}[\text{Vol}|\text{do}(\text{Statin})] = \int \mathbb{E}[\text{Vol}|\text{do}(\text{Statin}), \text{PSA}]\mathbb{P}(d\text{PSA}|\text{do}(\text{Statin})) \tag{61}$$
$$= \int \mathbb{E}[\text{Vol}|\text{PSA}]\mathbb{P}(d\text{PSA}|\text{do}(\text{Statin})) \qquad (\text{Vol} \perp\!\!\!\perp \text{Statin}|\text{PSA} \in \mathcal{G}_{\overline{\text{Statin}}}) \tag{62}$$
$$= \int \int \mathbb{E}[\text{Vol}|\text{PSA}]\mathbb{P}(d\text{PSA}|\text{Statin}, \text{Age}, \text{BMI})\mathbb{P}(d(\text{Age} \times \text{BMI})) \tag{63}$$

Where the last-line follows from the fact that (Age, BMI) satisfies the back-door criterion w.r.t. (Statin,PSA).

B.4.3. IMPLEMENTATION DETAILS

**True causal function:** (Synthetic benchmark) Since $\mathbb{E}[Y|B=b, C=c] = \mathbb{E}[\cos(D)|C=c] + \mathbb{E}[\sin(E)|C=c] = \mathbb{E}[Y|C=c]$, we can estimate $\mathbb{E}[Y|\text{do}(B=b), B=b']$ for each pair $(b, b')$ as follows. We sample $10^5$ Monte Carlo samples

from $\mathbb{P}_{C|B}(\bullet|b)$ and $P(A)$ which gives samples $(A_i, C_i)_{i=1}^{10^5}$ from $\mathbb{P}_{A,C|\text{do}(B=b),B=b'}$. We then use each sample to sample once from $\mathbb{P}_{E|A,C}$ and $\mathbb{P}_{D|C}$. This gives samples $(D_i, E_i)_{i=1}^{10^5}$ from $\mathbb{P}_{D,E|\text{do}(B=b),B=b'}$. Averaging $\cos(D) + \sin(E)$ w.r.t. these samples therefore estimates $\mathbb{E}[Y|\text{do}(B=b), B=b']$. See Appendix B.2 for estimation strategy for $\mathbb{E}[Y|\text{do}(D=d), B=b]$. (Healthcare example) Since $\mathbb{E}[\text{Vol}|\text{do}(\text{Statin})]$ is a marginal interventional quantity, one can estimate it by simply modifying the data generating process so that $\text{Statin} = s$ in the case $\text{do}(\text{Statin} = s)$.

The following settings are constant across all BO experiments, and so in what follows we assume we are optimizing an abstract causal function $\gamma : \mathcal{A} \to \mathbb{R}$, and $a$ is the intervention value. Note that we target the CATE and ATT above while fixing the conditioning variable $B$ to zero, so this also covers those cases.

**General BO settings:** For all methods and causal effects we use the expected improvement (EI) acquisition function, and run 10 BO iterations using Algorithm 1 in Aglietti et al. (2020). The kernel hyperparameters of each method are updated every $K = 3$ iterations. We assess the performance of each method by the cumulative regret scores $\sum_{i=1}^{10} |\gamma(a^*) - \gamma(a_i^{best})|$ and also through analysing the convergence profiles of $\gamma(a_i^{best})_{i=1}^{10}$ (here $a_i^{best} = \arg\min_{a \in \{a_1,..,a_i\}} |\gamma(a^*) - \gamma(a)|$ is the current best intervention). Cumulative regret rankings of methods can be sensitive to the number of iterations run. We focus on performance after 10 iterations (rather than a larger number) because in most real-life situations it is challenging to run more than 10 full scale experimental studies sequentially. Moreover, in healthcare situations this can be unethical: an exploratory intervention for a cancer drug dosage could cost livelihoods. We emphasise that finding the optimum (or achieving the same performance) in just two or three fewer iterations can have substantial practical impact.

**IMPspec:** The implementation used for the ATT in the synthetic example $\mathbb{E}[Y|\text{do}(B = b), B = b']$ is the same as for the CATE $\mathbb{E}[Y|\text{do}(D = d), B = b]$ (see Appendix B.2). The ATE in the healthcare example is of the form $\gamma(s) = \mathbb{E}_{\text{Age,BMI}}[\gamma(\text{Statin} = s, \text{Age}, \text{BMI})]$, where $\gamma(\text{Statin}, \text{Age}, \text{BMI})$ is derived from the two-stage model $(\text{Statin}, \text{Age}, \text{BMI}) \to \text{PSA}, \text{PSA} \to \text{Vol}$. The posterior on $\gamma(\text{Statin}, \text{Age}, \text{BMI})$ can therefore be estimated using the modification of Theorem 4.1 for the causal data fusion setting presented in Appendix A.3 and where $W = \emptyset$ (see Algorithm 3 and Remark A.9). The posterior on $\gamma(s)$ then follows by applying the formulae in Corollary C.6 in Appendix C. Note, for the outer integrals we use an average over the empirical samples of Age and BMI. For all CBO tasks, we use IMPspec to construct a GP prior for the causal effects using the posterior mean and covariance derived for each causal function: $f \sim \mathcal{GP}(\mathbb{E}[\gamma(\bullet)|\mathcal{X}^n], \mathbb{C}ov[\gamma(\bullet), \gamma(\bullet)|\mathcal{X}^n] + k_{RBF})$. Here $k_{RBF}$ is a Gaussian kernel. The choice of IMPspec's kernels as well as the training and inference details are the same as in the Toy Example.

**BayesIMP:** The implementation used for the ATT in the synthetic example $\mathbb{E}[Y|\text{do}(B = b), B = b']$ is the same as for the CATE $\mathbb{E}[Y|\text{do}(D = d), B = b]$ (see Appendix B.2). For the ATE in the Healthcare example we use the exact implementation in Proposition 5 in Chau et al. (2021b). The choice of BayesIMP's kernels as well as the training and inference details are the same as in the Toy Example. For the CBO tasks the implementation is the same as IMPspec.

**CBO:** Aglietti et al. (2020) construct the CBO prior $f \sim \mathcal{GP}(\hat{\mathbb{E}}[Y|\text{do}(\bullet)], \sqrt{\hat{\mathbb{V}}ar(Y|\text{do}(\bullet))} + k_{RBF})$, where $k_{RBF}$ is a Gaussian (radial basis function) kernel and $\hat{\mathbb{E}}[Y|\text{do}(X)], \hat{\mathbb{V}}ar(Y|\text{do}(X))$ are estimated from observational data. We estimate these functions using the approach in Singh et al. (2024), using the same kernels and hyperparameters as our method. Note, to use this approach to estimate the variance, we estimate $\hat{\mathbb{E}}[Y^2|\text{do}(X)]$ and then compute $\hat{\mathbb{V}}ar(Y|\text{do}(X)) = \hat{\mathbb{E}}[Y^2|\text{do}(X)] - \hat{\mathbb{E}}[Y|\text{do}(X)]^2$.

**BO:** The standard BO approach does not make use of observational data, and so simply uses $f \sim \mathcal{GP}(0, k_{RBF})$ as the prior (see Aglietti et al. (2020) for further details).

### B.5. 401(k) Dataset

**Dataset background.** We use the well-known 401(k) dataset, previously studied in Chernozhukov & Hansen (2004) and commonly used as a benchmark for treatment-effect estimation. The dataset records household-level financial and demographic information relevant to the effect of 401(k) pension-plan eligibility on financial wealth. We access the preprocessed version through the `fetch_401K` loader in the `DoubleML` package (Bach et al., 2022). In this dataset, the variable `e401` indicates whether an employer offers access to a 401(k) plan, while `net_tfa` measures net total financial assets. We therefore interpret eligibility as the treatment and net financial assets as the outcome. Since there is no ground-truth causal effect available for this observational dataset, we use it as a real-data case study for assessing whether IMPSPEC produces plausible uncertainty-aware effect estimates at a realistic sample size.

**Causal estimand.**    Let $Y = \texttt{net\_tfa}$ denote net total financial assets, and let $A = \texttt{e401}$ denote the binary indicator for 401(k) eligibility. We study how the effect of eligibility varies with household income, $I = \texttt{inc}$. The adjustment variables used in our implementation are

$$V = (\texttt{age}, \texttt{marr}, \texttt{twoearn}, \texttt{pira}, \texttt{hown}),$$

corresponding to age, marital status, two-earner household status, IRA participation, and home ownership. Thus, in the notation of Equation (1), we set

$$W = (A, I), \qquad Z = I, \qquad V = (\texttt{age}, \texttt{marr}, \texttt{twoearn}, \texttt{pira}, \texttt{hown}).$$

For $a \in \{0, 1\}$ and income level $i$, the corresponding conditional interventional mean is

$$\gamma((a, i), i) = \int \mathbb{E}[Y \mid A = a, I = i, V = v] \, P_{V|I}(dv \mid i).$$

The conditional average treatment effect of interest is then

$$\tau(i) = \gamma((1, i), i) - \gamma((0, i), i).$$

**Preprocessing.**    We set the random seed to zero and shuffle the observations before subsampling. We use the full sample size of $n = 9915$. All variables used by IMPSPEC are standardized column-wise using

$$x \mapsto \frac{x - \bar{x}}{s_x},$$

where $\bar{x}$ and $s_x$ are the empirical mean and standard deviation computed on the retained data. Standardization is applied to the outcome $Y$, income $I$, the intervention-regression input $W = (A, I)$, and the adjustment variables $V$. Predictions are transformed back to the original outcome scale before plotting.

**Model fitting.**    We use Gaussian kernels for $k_W$, $k_V$, and $k_Z$, with per-dimension lengthscales. The spectral term in the posterior variance is approximated using $10^5$ Monte Carlo samples from the spectral measure. The final model is trained on the full standardized dataset for 1000 iterations with learning rate 0.2, ridge regularization $10^{-3}$, and minibatches of size 512. The median heuristic for kernel initialization is computed using 512 subsamples.

**Spectral calibration.**    Since the 401(k) dataset has no ground-truth causal effect, we calibrate the spectral-measure scale using the plug-in bootstrap procedure in Algorithm 2. We search over the candidate scales

$$\beta \in \{0.01, 0.1, 1\},$$

using 20 bootstrap replications and nominal levels $\alpha \in \{0.01, \ldots, 0.99\}$. Calibration is performed with a 50/50 train-calibration split. For computational efficiency, the hyperparameter training steps inside calibration use the same minibatch kernel objectives as final training, with minibatches of size 512. Conditional on these fitted hyperparameters, each bootstrap calibration interval is computed using the full calibration bootstrap sample. To ensure that calibration only selects the spectral scale and does not alter the final fitted hyperparameters, we run the calibration procedure on a temporary model, retain only the selected scale $\hat{\beta}$, discard the temporary model, and then train the final IMPSPEC model from scratch on the full dataset. The selected scale $\hat{\beta}$ is then used in the posterior variance and covariance computations for the final CATE intervals.

**CATE evaluation grid.**    We evaluate the treatment effect over a grid of 50 income values. To avoid extrapolating far outside the empirical support, the grid is restricted to the central empirical income range,

$$i \in \left[ \widehat{Q}_{0.05}(I), \widehat{Q}_{0.95}(I) \right].$$

For each income value $i$, we construct the two test inputs $w_0(i) = (0, i)$, $w_1(i) = (1, i)$, with common conditioning value $z = i$. These test inputs are standardized using the training-set means and standard deviations before posterior evaluation.

**Posterior mean and variance for the treatment effect.** Define

$$m_a(i) := \gamma((a, i), i), \qquad a \in \{0, 1\}.$$

We compute the posterior means

$$\hat{m}_0(i) = \mathbb{E}[m_0(i) \mid \mathcal{X}^n], \qquad \hat{m}_1(i) = \mathbb{E}[m_1(i) \mid \mathcal{X}^n],$$

using the paired posterior-mean computation. The posterior mean of the CATE is then

$$\hat{\tau}(i) = \hat{m}_1(i) - \hat{m}_0(i).$$

To account for posterior dependence between the two intervention levels, we compute

$$\widehat{\mathrm{Var}}(m_0(i) \mid \mathcal{X}^n), \qquad \widehat{\mathrm{Var}}(m_1(i) \mid \mathcal{X}^n), \qquad \widehat{\mathrm{Cov}}(m_0(i), m_1(i) \mid \mathcal{X}^n),$$

rather than treating the two posterior curves as independent. The posterior variance of the CATE is therefore computed as

$$\widehat{\mathrm{Var}}(\tau(i) \mid \mathcal{X}^n) = \widehat{\mathrm{Var}}(m_1(i) \mid \mathcal{X}^n) + \widehat{\mathrm{Var}}(m_0(i) \mid \mathcal{X}^n) - 2\widehat{\mathrm{Cov}}(m_0(i), m_1(i) \mid \mathcal{X}^n).$$

In the implementation, this quantity is clamped below at zero for numerical stability before taking the square root. The posterior standard deviation is then transformed back to the original outcome scale.

For a nominal level $\alpha$, the central credible interval is

$$\widehat{\mathrm{CI}}_\alpha(i) = \left[ \hat{\tau}(i) - \Phi^{-1}\left(\frac{1+\alpha}{2}\right) \hat{\sigma}_\tau(i), \ \hat{\tau}(i) + \Phi^{-1}\left(\frac{1+\alpha}{2}\right) \hat{\sigma}_\tau(i) \right],$$

where

$$\hat{\sigma}_\tau(i) = \sqrt{\widehat{\mathrm{Var}}(\tau(i) \mid \mathcal{X}^n)}.$$

In Figure 6, we plot the posterior mean together with central 50%, 90%, and 95% credible intervals, using the calibrated spectral scale $\hat{\beta}$ in the posterior variance and covariance terms.

**Runtime and memory.** The runtime and memory measurements reported in Table 4 are recorded separately for calibration, final training, and final inference. Calibration is run first on a temporary model and is used only to select the spectral scale $\hat{\beta}$. Final training is then performed from scratch on the full dataset using minibatches of size 512. Final posterior inference uses the full dataset and computes the posterior means, variances, and covariance terms needed for the CATE credible intervals. Peak GPU memory is reported as the maximum allocated memory during each stage.

## C. Mathematical Results and Proofs

### C.1. Auxiliary Results

**Lemma C.1.** *Let $\mathcal{V}$ satisfy Assumption A.1 and $k_V : \mathcal{V}^2 \to \mathbb{R}$ be a positive definite kernel that satisfies Assumption A.3 and Assumption A.4. Then, the feature map $\phi_V : \mathcal{V} \to \mathbb{R}^I, v \mapsto (\phi_{V,i}(v))_{i \in I}$ is injective, where $(\phi_{V,i})_{i \in I}$ are the eigenfunctions of the operator $\mathcal{T}_{k,\nu}$ defined in Assumption A.4.*

*Proof.* To start, note by Mercer's theorem that the canonical feature map can be written as

$$\psi_V(v) = \sum_{i \in I} a_i \, e_i \quad \in \mathcal{H}_V,$$

where $(a_i)_{i \in I} := (\sqrt{\lambda_{V,i}} \, \phi_{V,i}(v))_{i \in I} \in \ell_2(I)$ and $(e_i)_{i \in I} := (\sqrt{\lambda_{V,i}} \, \phi_{V,i}(\bullet))_{i \in I}$ is an orthonormal basis of $\mathcal{H}_V$. Hence each $\psi_V(V_i)$ has coordinates $\langle \psi_V(V_i), e_j \rangle = \sqrt{\lambda_{V,j}} \phi_{V,j}(V_i)$. Since all eigenvalues $\lambda_{V,i}$ are strictly positive, the map

$$g : \mathbb{R}^I \longrightarrow \mathcal{H}_V, \quad (b_i)_{i \in I} \longmapsto \sum_{i \in I} b_i \lambda_{V,i}^{\frac{1}{2}} e_i$$

is injective. This holds because the map $(b_i)_{i \in I} \mapsto (\lambda_{V,i}^{\frac{1}{2}} b_i)_{i \in I}$ from $\mathbb{R}^I \to \ell_2$ is injective, and each element of $\mathcal{H}_V$ has a unique set of co-ordinates $(a_i)_{i \in I} \in \ell_2(I)$. Combining the injectivity of $g$ with the injectivity of the composition $\psi_V := g \circ \phi_V$ (which holds by characteristic-ness of $k_V$), we establish that $\phi_V : \mathcal{V} \to \mathbb{R}^I, v \mapsto (\phi_{V,i}(v))_{i \in I}$ is injective. $\square$

**Lemma C.2** (Series representation of an isonormal Gaussian process). *Let $(\Omega, \mathcal{F}, \mathbb{P})$ be a probability space, and let $H$ be a separable real Hilbert space with orthonormal basis $(e_i)_{i \in I}$, where $I$ is countable. Let $X = (X(h))_{h \in H}$ be an isonormal Gaussian process on $H$, that is, a centered Gaussian family satisfying*

$$\mathbb{E}[X(h)X(g)] = \langle h, g \rangle_H, \qquad \forall h, g \in H.$$

*Then there is a sequence $(Z_i)_{i \in I}$ of independent standard Normal variables $(Z_i)_{i \in I}$ such that for every $h \in H$,*

$$X(h) = \sum_{i \in I} \langle h, e_i \rangle Z_i, \tag{64}$$

*where the series converges in $L^2(\Omega)$.*

*Proof.* For each $i \in I$, define $Z_i := X(e_i)$. Since $X$ is centered Gaussian and

$$\mathbb{E}[Z_i Z_j] = \mathbb{E}[X(e_i)X(e_j)] = \langle e_i, e_j \rangle_H = \delta_{ij},$$

it follows that $(Z_i)_{i \in I}$ is a sequence of independent $\mathcal{N}(0,1)$ random variables.

Fix $h \in H$ and set

$$S_n(h) := \sum_{i=1}^{n} \langle h, e_i \rangle Z_i, \qquad n \in I.$$

We show that $S_n(h) \to X(h)$ in $L^2(\Omega)$ directly by expanding the mean-square error:

$$\mathbb{E}\left[\left|X(h) - S_n(h)\right|^2\right] = \mathbb{E}[X(h)^2] + \mathbb{E}[S_n(h)^2] - 2\,\mathbb{E}[X(h)S_n(h)].$$

Using the defining covariance structure of the isonormal process,

$$\mathbb{E}[X(h)^2] = \|h\|_H^2.$$

For the second term, note that $S_n(h)$ is a finite linear combination of Gaussian variables:

$$\begin{aligned}
\mathbb{E}[S_n(h)^2] &= \sum_{i,j=1}^{n} \langle h, e_i \rangle \langle h, e_j \rangle \, \mathbb{E}[X(e_i)X(e_j)] \\
&= \sum_{i=1}^{n} \langle h, e_i \rangle^2 \\
&= \|P_n h\|_H^2,
\end{aligned}$$

where $P_n h := \sum_{i=1}^{n} \langle h, e_i \rangle e_i$ is the orthogonal projection of $h$ onto $\text{span}\{e_1, \ldots, e_n\}$.

For the cross term,

$$\begin{aligned}
\mathbb{E}[X(h)S_n(h)] &= \sum_{i=1}^{n} \langle h, e_i \rangle \, \mathbb{E}[X(h)X(e_i)] \\
&= \sum_{i=1}^{n} \langle h, e_i \rangle^2 \\
&= \|P_n h\|_H^2.
\end{aligned}$$

Substituting these expressions gives

$$\begin{aligned}
\mathbb{E}\left[\left|X(h) - S_n(h)\right|^2\right] &= \|h\|_H^2 + \|P_n h\|_H^2 - 2\|P_n h\|_H^2 \\
&= \|h - P_n h\|_H^2.
\end{aligned}$$

Since $P_n h \to h$ in $H$ as $n \to \infty$, the right-hand side tends to 0, proving that $S_n(h) \to X(h)$ in $L^2(\Omega)$. This establishes the series representation (64). $\qquad\square$

**Lemma C.3** (GP Regression Posterior on General Input Space). *Let $(\mathcal{X}, d)$ be any metric space. Let $k : \mathcal{X} \times \mathcal{X} \to \mathbb{R}$ be a symmetric positive definite kernel and let $f \sim \mathcal{GP}(m, k)$ be a Gaussian process on $\mathcal{X}$ with mean function $m : \mathcal{X} \to \mathbb{R}$. Fix distinct design points $X_{1:n} = (x_1, \ldots, x_n) \in \mathcal{X}^n$ and observe*

$$Y_i \;=\; f(x_i) + \xi_i, \qquad \xi_i \overset{i.i.d.}{\sim} \mathcal{N}(0, \sigma^2), \quad \xi \perp f.$$

*Write $\mathbf{Y} = (Y_1, \ldots, Y_n)^\top$, $m(X_{1:n}) = (m(x_1), \ldots, m(x_n))^\top$, $K(X_{1:n}, X_{1:n}) = [k(x_i, x_j)]_{i,j=1}^n$, and for $x \in \mathcal{X}$ define $k(x, X_{1:n}) = (k(x, x_1), \ldots, k(x, x_n))^\top$. Then the posterior of $f$ given $\mathbf{Y}$ is a Gaussian process*

$$f \mid \mathbf{Y} \;\sim\; \mathcal{GP}(m_*, k_*),$$

*with mean and covariance*

$$m_*(x) \;=\; m(x) + k(x, X_{1:n})^\top \big(K(X_{1:n}, X_{1:n}) + \sigma^2 I_n\big)^{-1}\big(\mathbf{Y} - m(X_{1:n})\big), \tag{65}$$

$$k_*(x, x') \;=\; k(x, x') - k(x, X_{1:n})^\top \big(K(X_{1:n}, X_{1:n}) + \sigma^2 I_n\big)^{-1} k(X_{1:n}, x'). \tag{66}$$

*Equivalently, for any finite test set $X_* = (x_1^*, \ldots, x_m^*)$, the vector $f(X_*) \mid \mathbf{Y}$ is multivariate normal with mean and covariance obtained by evaluating (65)–(66) on $X_*$.*

*Proof.* By the GP prior, the joint vector

$$\big(f(X_*), \, \mathbf{Y}\big) \;=\; \big(f(X_*), \, f(X_{1:n}) + \xi\big)$$

is multivariate Gaussian. Its mean is $\big(m(X_*), \, m(X_{1:n})\big)$, and its covariance is

$$\begin{pmatrix} K(X_*, X_*) & K(X_*, X_{1:n}) \\ K(X_{1:n}, X_*) & K(X_{1:n}, X_{1:n}) + \sigma^2 I_n \end{pmatrix},$$

since $\xi$ is independent of $f$ and has covariance $\sigma^2 I_n$. Conditioning a jointly Gaussian vector on its second block yields that $f(X_*) \mid \mathbf{Y}$ is Gaussian with mean and covariance given by the standard block-conditioning formulas, which are exactly (65)–(66) evaluated on $X_*$. As this holds for every finite $X_*$, the posterior is the GP with mean $m_*$ and covariance $k_*$ stated above. $\qquad\square$

*Remark* C.4. Lemma C.3 is simply the standard GP regression posterior in the case where $\mathcal{X}$ is a general metric space. Since this case is not regularly considered (we note Koepernik & Pfaff (2021) recently analyzed GP posteriors in this setting), we present it for completeness.

**Lemma C.5** (Expectation of Isonormal GP). *Let $(\Omega_X, \mathcal{F}_X, \mathbb{P}_X)$ and $(\Omega_H, \mathcal{F}_H, \mathbb{P}_H)$ be probability spaces and write $(\Omega, \mathcal{F}, \mathbb{P}) := (\Omega_X \times \Omega_H, \, \mathcal{F}_X \otimes \mathcal{F}_H, \, \mathbb{P}_X \otimes \mathbb{P}_H)$. Let $\mathcal{H}$ be a separable real Hilbert space, and let $X : \mathcal{H} \to L^2(\mathbb{P}_X)$ be an isonormal Gaussian process; i.e., $X$ is a centered Gaussian linear isometry with $\mathbb{E}[X(h)X(g)] = \langle h, g \rangle_{\mathcal{H}}$ for all $h, g \in \mathcal{H}$. Let $H : \Omega_H \to \mathcal{H}$ be square-integrable, $H \in L^2(\mathbb{P}_H; \mathcal{H})$. Define the random variable on $\Omega$ by $X(H)(\omega_X, \omega_H) := X\big(H(\omega_H)\big)(\omega_X)$ and the conditional expectation $\mathbb{E}_H[\cdot] := \mathbb{E}[\cdot \mid \mathcal{F}_X]$, i.e. integration over $\Omega_H$ only. Then,*

$$\mathbb{E}_H\big[X(H)\big] \;=\; X\big(\mathbb{E}[H]\big) \quad \text{in } L^2(\mathbb{P}_X).$$

*Proof.* Fix an orthonormal basis $(e_i)_{i \in I}$ of $\mathcal{H}$ and set $Z_i := X(e_i) \in L^2(\mathbb{P}_X)$. Since $X$ is an isonormal GP, it admits the expansion

$$X(h) \;=\; \sum_{i \in I} \langle h, e_i \rangle Z_i \quad \text{in } L^2(\mathbb{P}_X),$$

with $(Z_i)$ i.i.d. $N(0, 1)$ (for a proof see Lemma C.2).

Let now $H \in L^2(\mathbb{P}_H; \mathcal{H})$ be a random element and write $P_n H := \sum_{i=1}^n \langle H, e_i \rangle e_i$. In this case, we can extend the above $L^2(\mathbb{P}_X)$ convergence to convergence in $L^2(\mathbb{P}_X \otimes \mathbb{P}_H)$. In particular, by Fubini's Theorem,

$$\mathbb{E}_{X,H}\left[\left|X(H) - \sum_{i=1}^n \langle H, e_i \rangle Z_i\right|^2\right] = \mathbb{E}_H \, \mathbb{E}_X\left[\left|X(H) - \sum_{i=1}^n \langle H, e_i \rangle Z_i\right|^2 \,\Big|\, H\right]$$

$$= \mathbb{E}_H\big[\|H - P_n H\|_{\mathcal{H}}^2\big],$$

where we use the convention that $\mathbb{E}_X[\bullet]$ is the integral w.r.t $\mathbb{P}_X$ and the same for $\mathbb{E}_H[\bullet]$.

Since $P_n$ is an orthogonal projection, $\|H - P_n H\|_{\mathcal{H}} \leq \|H\|_{\mathcal{H}}$ and $P_n H(\omega_H) \to H(\omega_H)$ in $\mathcal{H}$ for each $\omega_H$. Thus, by dominated convergence (with dominator $\|H\|_{\mathcal{H}}^2 \in L^1(\Omega_H)$),

$$\lim_{n \to \infty} \mathbb{E}_H\big[\|H - P_n H\|_{\mathcal{H}}^2\big] = \mathbb{E}_H\big[\lim_{n \to \infty} \|H - P_n H\|_{\mathcal{H}}^2\big] = 0.$$

Hence, for random $H \in L^2(\mathbb{P}_H; \mathcal{H})$, we have the jointly convergent representation

$$X(H) = \sum_{i \in I} \langle H, e_i \rangle Z_i \quad \text{in } L^2(\mathbb{P}_X \otimes \mathbb{P}_H) \tag{67}$$

Now, for each $n$, let $S_n := \sum_{i=1}^n \langle H, e_i \rangle Z_i$. Because $\mathbb{E}_H[\cdot]$, by Jensen's Inequality is the orthogonal projection onto $L^2(\mathbb{P}_X)$, it is a contraction on $L^2(\mathbb{P}_X \otimes \mathbb{P}_H)$.[10] This implies that

$$\big\|\mathbb{E}_H[S_n] - \mathbb{E}_H[X(H)]\big\|_{L^2(\mathbb{P}_X)} \leq \|S_n - X(H)\|_{L^2(\mathbb{P}_X \otimes \mathbb{P}_H)} \xrightarrow[n \to \infty]{} 0.$$

and so,

$$\mathbb{E}_H[S_n] \xrightarrow[n \to \infty]{L^2(\mathbb{P}_X)} \mathbb{E}_H[X(H)].$$

Now, note for finite sums we can pull $\mathbb{E}_H$ inside:

$$\mathbb{E}_H[S_n] = \sum_{i=1}^n \big(\mathbb{E}\langle H, e_i \rangle\big) Z_i,$$

since each $Z_i$ is $\mathcal{F}_X$–measurable (hence constant w.r.t. $\mathbb{P}_H$). Therefore,

$$\mathbb{E}_H[X(H)] = \sum_{i \in I} \big(\mathbb{E}\langle H, e_i \rangle\big) Z_i \quad \text{in } L^2(\mathbb{P}_X). \tag{68}$$

Finally, we identify the element of $\mathcal{H}$ whose coordinates are the means of the coordinates of $H$. By Cauchy–Schwarz and Parseval,

$$\sum_{i \in I} \big(\mathbb{E}\langle H, e_i \rangle\big)^2 \leq \sum_{i \in I} \mathbb{E}\langle H, e_i \rangle^2 = \mathbb{E}\|H\|_{\mathcal{H}}^2 < \infty,$$

so $m := \sum_{i \in I} \big(\mathbb{E}\langle H, e_i \rangle\big) e_i \in \mathcal{H}$. For any $u = \sum_{i \in I} u_i e_i \in \mathcal{H}$ with $(u_i) \in \ell^2$,

$$\langle m, u \rangle_{\mathcal{H}} = \sum_{i \in I} \big(\mathbb{E}\langle H, e_i \rangle\big) u_i = \mathbb{E} \sum_{i \in I} \langle H, e_i \rangle u_i = \mathbb{E}\langle H, u \rangle_{\mathcal{H}},$$

where exchanging sum and expectation is justified by Cauchy–Schwarz: $\mathbb{E}|\sum_{i \in I} \langle H, e_i \rangle u_i| \leq (\mathbb{E} \sum_{i \in I} \langle H, e_i \rangle^2)^{1/2}(\sum_{i \in I} u_i^2)^{1/2} < \infty$. Thus $m$ represents the Bochner mean of $H$; i.e. $m = \mathbb{E}[H] \in \mathcal{H}$. Using the series representation for deterministic arguments once more,

$$X\big(\mathbb{E}[H]\big) = X(m) = \sum_{i \in I} \big(\mathbb{E}\langle H, e_i \rangle\big) Z_i \quad \text{in } L^2(\mathbb{P}_X).$$

Comparing with (68) yields $\mathbb{E}_H[X(H)] = X(\mathbb{E}[H])$ in $L^2(\mathbb{P}_X)$, as claimed. $\qquad\square$

## C.2. Proofs of Main Results and Omitted Results

### C.2.1. REPRESENTATION OF IMPSPEC

Here we show that the processes defined in (8) and (9) converge in mean square. The key step is to justify that the expectation can pass through the (possibly unbounded) linear functional $f$, which follows directly from Lemma C.5. For clarity, we define the Gaussian process $f$ and the random variables $(W, V, Z)$ on independent probability spaces.

---

[10]Note for any $U \in L^2(\mathbb{P}_X \otimes \mathbb{P}_H)$ we have $\|\mathbb{E}_H[U]\|_{L^2(\mathbb{P}_X)}^2 = \mathbb{E}_X\big[(\mathbb{E}_H[U])^2\big] \leq \mathbb{E}_X\big[\mathbb{E}_H[U^2]\big] = \|U\|_{L^2(\mathbb{P}_X \otimes \mathbb{P}_H)}^2$.

**Proposition C.6** ($L^2$-Convergent Representation of IMPSPEC)**.** *Let $(\mathcal{W}, \mathcal{V}, \mathcal{Z})$ be standard Borel spaces and $\mathcal{H}_W, \mathcal{H}_V$ be separable real Hilbert spaces with feature maps $\psi_W : \mathcal{W} \to \mathcal{H}_W$ and $\psi_V : \mathcal{V} \to \mathcal{H}_V$. Let $\mathcal{H} := \mathcal{H}_W \otimes \mathcal{H}_V$ denote the Hilbert tensor product, and let $f : \mathcal{H} \to L^2(\mathbb{P}_f)$ be an isonormal Gaussian process on a probability space $(\Omega_f, \mathcal{F}_f, \mathbb{P}_f)$; that is, $f$ is a centered Gaussian linear isometry satisfying $\mathbb{E}_f[f(h)f(g)] = \langle h, g \rangle_{\mathcal{H}}$ for all $h, g \in \mathcal{H}$.*

*Let $(\Omega_h, \mathcal{F}_h, \mathbb{P}_h)$ be another standard Borel probability space, independent of $(\Omega_f, \mathcal{F}_f, \mathbb{P}_f)$, and let $V : \Omega_h \to \mathcal{V}$ and $Z : \Omega_h \to \mathcal{Z}$ be random elements. Assume $\psi_V(V) \in L^2(\mathbb{P}_h; \mathcal{H}_V)$.*

*For each $w \in \mathcal{W}$ and $z \in \mathcal{Z}$, define*

$$\gamma(w, z) := \mathbb{E}_V \left[ f\big(\psi_W(w) \otimes \psi_V(V)\big) \,\big|\, Z = z \right] \quad \in L^2(\mathbb{P}_f),$$

*where the conditional expectation is understood as a measurable version of the regular conditional expectation of the $L^2(\mathbb{P}_f \otimes \mathbb{P}_h)$ random variable $f(\psi_W(w) \otimes \psi_V(V))$ given $Z$.*

*Then, for $\mathbb{P}_Z$–almost every $z \in \mathcal{Z}$,*

$$\gamma(w, z) \;=\; f\Big(\psi_W(w) \otimes \mathbb{E}[\psi_V(V) \mid Z = z]\Big) \quad \text{in } L^2(\mathbb{P}_f).$$

*Proof.* Fix $w \in \mathcal{W}$ and set $\mathcal{H} := \mathcal{H}_W \otimes \mathcal{H}_V$. Define

$$H(\omega_h) := \psi_W(w) \otimes \psi_V(V(\omega_h)) \in \mathcal{H}.$$

Since $\psi_V(V) \in L^2(\mathbb{P}_h; \mathcal{H}_V)$ and $\psi_W(w)$ is fixed, we have $H \in L^2(\mathbb{P}_h; \mathcal{H})$ with $\|H(\omega_h)\|_{\mathcal{H}} = \|\psi_W(w)\|_{\mathcal{H}_W} \|\psi_V(V(\omega_h))\|_{\mathcal{H}_V}$.

Because $\mathcal{Z}$ is standard Borel and $\mathcal{H}$ is separable, there exists a regular conditional law $\mathbb{P}_{H|Z=z}$ of $H$ given $Z$. Since $H \in L^2(\mathbb{P}_h; \mathcal{H})$,

$$\int_{\mathcal{H}} \|h\|_{\mathcal{H}}^2 \, \mathbb{P}_{H|Z=z}(dh) < \infty \qquad \text{for } \mathbb{P}_Z\text{-a.e. } z,$$

so the conditional second moment is finite.

For such a $z$, define the conditional space $(\Omega_h^z, \mathcal{F}_h^z, \mathbb{P}_{H|Z=z}) := (\mathcal{H}, \mathcal{B}(\mathcal{H}), \mathbb{P}_{H|Z=z})$ and let $H^z : \Omega_h^z \to \mathcal{H}$ be the identity map, $H^z(\omega) = \omega$. Then $H^z \in L^2(\mathbb{P}_{H|Z=z}; \mathcal{H})$ and $H^z \sim \mathbb{P}_{H|Z=z}$.

Now consider the product space

$$(\Omega, \mathcal{F}, \mathbb{P}) := (\Omega_f \times \Omega_h^z, \; \mathcal{F}_f \otimes \mathcal{F}_h^z, \; \mathbb{P}_f \otimes \mathbb{P}_{H|Z=z}).$$

By independence of $(\Omega_f, \mathcal{F}_f, \mathbb{P}_f)$ and $(\Omega_h, \mathcal{F}_h, \mathbb{P}_h)$, conditioning on $Z = z$ does not affect the law of $f$, so $f$ and $H^z$ are independent on $\Omega$. Define

$$f(H^z)(\omega_f, \omega_h^z) := f(H^z(\omega_h^z))(\omega_f), \quad \text{so } f(H^z) \in L^2(\mathbb{P}_f \otimes \mathbb{P}_{H|Z=z}).$$

Applying Lemma C.5 (Expectation of Isonormal GP) to $X = f$ and the random element $H^z$ yields

$$\mathbb{E}_{H^z}[f(H^z)] = f(\mathbb{E}[H^z]) \quad \text{in } L^2(\mathbb{P}_f).$$

By the definition of the conditional law,

$$\mathbb{E}_{H^z}[f(H^z)] = \mathbb{E}[f(H) \mid Z = z], \qquad \mathbb{E}[H^z] = \mathbb{E}[H \mid Z = z].$$

Therefore, for $\mathbb{P}_Z$–a.e. $z \in \mathcal{Z}$,

$$\mathbb{E}[f(H) \mid Z = z] = f(\mathbb{E}[H \mid Z = z]) \quad \text{in } L^2(\mathbb{P}_f).$$

Finally, since $H = \psi_W(w) \otimes \psi_V(V)$ and $T : \mathcal{H}_V \to \mathcal{H}$, $u \mapsto \psi_W(w) \otimes u$, is bounded linear, the Bochner conditional expectation gives

$$\mathbb{E}[H \mid Z = z] = T\big(\mathbb{E}[\psi_V(V) \mid Z = z]\big) = \psi_W(w) \otimes \mathbb{E}[\psi_V(V) \mid Z = z].$$

Substituting in these definitons, we get

$$\gamma(w, z) = \mathbb{E}[f(\psi_W(w) \otimes \psi_V(V)) \mid Z = z] = f\Big(\psi_W(w) \otimes \mathbb{E}[\psi_V(V) \mid Z = z]\Big) \quad \text{in } L^2(\mathbb{P}_f),$$

for $\mathbb{P}_Z$–a.e. $z$, as claimed. $\qquad\square$

C.2.2. PROOF OF THEOREM 4.1

**Theorem C.1** (Posterior Moments of $\gamma$ (Full Statement)). *Let Assumption A.1, Assumption A.3 and Assumption A.4 hold. Under* (5)–(11) *in the main text, the posterior mean and variance of $\gamma(w, z) \mid \mathcal{X}^n$ are given by*

$$\mathbb{E}[\gamma(w, z)|\mathcal{X}^n] = \boldsymbol{\beta}(z)^\top K_V \boldsymbol{\alpha}(w)$$
$$\mathbb{V}ar[\gamma(w, z)|\mathcal{X}^n] = S_1 + S_2 + S_3$$

*where*

$$S_1 = \boldsymbol{\beta}(z)^\top K_V (Ik_W(w, w) - A(w)K_V)\boldsymbol{\beta}(z)$$
$$S_2 = \hat{k}_Z(z, z)(Tr[\tilde{K}_V(\boldsymbol{\alpha}(w)\boldsymbol{\alpha}(w)^\top - A(w))])$$
$$S_3 = \tau(k(z, z) - \boldsymbol{k}_Z(z)^\top \boldsymbol{\beta}(z))k_W(w, w)$$
$$\boldsymbol{\alpha}(w) = D(w)(K_W \odot K_V + \sigma^2 I)^{-1}\boldsymbol{Y}$$
$$A(w) = D(w)(K_W \odot K_V + \sigma^2 I)^{-1}D(w)$$
$$\boldsymbol{\beta}(z) = (K_Z + \eta^2 I)^{-1}\boldsymbol{k}_Z(z)$$

*and, for $x \in \{w, v, z\}$ we use the definitions*

$$D(w) := \mathrm{diag}(\boldsymbol{k}_W(w)) \quad \boldsymbol{k}_X(x) := [k_X(x, X_i)]_{i=1}^n,$$
$$\tilde{K}_V := \int \boldsymbol{k}_V(v)\boldsymbol{k}_V(v)^\top d\nu(v) \quad \tau := \sum_{i \in I} \lambda_i,$$

*Proof.* We first derive the factorization structure of the posterior on $f, (\mu_i)_{i \in I} \mid \mathcal{X}^n$, where recall $\mathcal{X}^n := \{(Y_i, V_i, W_i, Z_i)\}_{i=1}^n$. We write $\boldsymbol{Y} := (Y_i)_{i=1}^n$ and similarly for $\boldsymbol{W}, \boldsymbol{V}, \boldsymbol{Z}$, and note that $I \subseteq \mathbb{N}$ is countable by Assumption A.4. For any random variables $X_1, X_2$ taking values in standard Borel spaces, we denote by $\mathbb{P}_{X_1|X_2}(\bullet \mid \bullet)$ or equivalently $\mathbb{P}(X_1 \in \bullet \mid \bullet)$ the corresponding regular conditional distribution (probability kernel).

**Deriving the Posterior Factorization.** Let

$$\boldsymbol{f} := \big[f(\psi_W(W_1) \otimes \psi_V(V_1)), \dots, f(\psi_W(W_n) \otimes \psi_V(V_n))\big], \quad \boldsymbol{\mu}_i := [\mu_i(Z_1), \dots, \mu_i(Z_n)] \; \forall i \in I,$$

be the evaluations at the training points, and let

$$\boldsymbol{f}^* := \big[f(\psi_W(w_1) \otimes \psi_V(v_1)), \dots, f(\psi_W(w_m) \otimes \psi_V(v_m))\big], \quad \boldsymbol{\mu}_i^* := [\mu_i(z_1), \dots, \mu_i(z_m)], \; \forall i \in I,$$

be the evaluations at $m$ fixed test points. Under the GP priors on $f$ and $(\mu_i)_{i \in I}$, these are Gaussian random vectors:

$$(\boldsymbol{f}, \boldsymbol{f}^*) \sim \mathcal{N}(0, \Sigma_f), \qquad ((\boldsymbol{\mu}_i, \boldsymbol{\mu}_i^*)_{i \in I}) \sim \bigotimes_{i \in I} \mathcal{N}(0, \Sigma_\mu),$$

where the block covariances $\Sigma_f$ and $\Sigma_\mu$ are induced by $k_W \otimes k_V$ and $k_Z$, respectively. Hence

$$\boldsymbol{f}, \boldsymbol{\mu}_i \in (\mathbb{R}^n, \mathcal{B}(\mathbb{R}^n)), \qquad \boldsymbol{f}^*, \boldsymbol{\mu}_i^* \in (\mathbb{R}^m, \mathcal{B}(\mathbb{R}^m)),$$

where $\mathcal{B}(\mathbb{R}^m)$ denotes the Borel $\sigma$-algebra on $\mathbb{R}^m$. By Assumption A.1, the observation spaces $\mathcal{Y}, \mathcal{W}, \mathcal{V}, \mathcal{Z}$ are also standard Borel. Since a countable product of standard Borel spaces is again standard Borel under the product $\sigma$-algebra (see, e.g., Parthasarathy (2005, Thm. 2.3)), the joint law

$$\mathbb{P}_{\mathcal{X}^n, \boldsymbol{f}, \boldsymbol{f}^*, (\boldsymbol{\mu}_i, \boldsymbol{\mu}_i^*)_{i \in I}}$$

is a Borel probability measure, and regular conditional distributions exist on all components (Kallenberg, 1997). Under the Gaussian observation models (10)–(11) and independence of the priors, this joint law admits the following disintegration:

$$\mathbb{P}_{\mathcal{X}^n, \boldsymbol{f}, (\boldsymbol{\mu}_i)_{i \in I}, \boldsymbol{f}^*, (\boldsymbol{\mu}_i^*)_{i \in I}} = \mathbb{P}_{\boldsymbol{Y}|\boldsymbol{V}, \boldsymbol{W}, \boldsymbol{f}} \otimes \mathbb{P}_{\boldsymbol{W}|\boldsymbol{V}, \boldsymbol{Z}} \otimes \mathbb{P}_{\boldsymbol{V}|\boldsymbol{Z}, (\boldsymbol{\mu}_i)_{i \in I}} \otimes \mathbb{P}_{\boldsymbol{Z}} \otimes \mathbb{P}_{\boldsymbol{f}^*|\boldsymbol{f}} \otimes \mathbb{P}_{\boldsymbol{f}} \otimes \bigotimes_{i \in I} \big(\mathbb{P}_{\boldsymbol{\mu}_i^*|\boldsymbol{\mu}_i} \otimes \mathbb{P}_{\boldsymbol{\mu}_i}\big).$$

where we have used the fact that (i) $\boldsymbol{Y} \perp\!\!\!\perp (\boldsymbol{Z}, \boldsymbol{f}^*, (\boldsymbol{\mu}_i, \boldsymbol{\mu}_i^*)_{i \in I}) \mid \boldsymbol{V}, \boldsymbol{W}, \boldsymbol{f}$ by (10), (ii) $\boldsymbol{V} \perp\!\!\!\perp (\boldsymbol{f}, \boldsymbol{f}^*, (\boldsymbol{\mu}_i^*)_{i \in I}) \mid \boldsymbol{Z}, (\boldsymbol{\mu}_i)_{i \in I}$ by[11] (11), as well as the fact that $\boldsymbol{W} \perp\!\!\!\perp (\boldsymbol{f}, \boldsymbol{f}^*, (\boldsymbol{\mu}_i, \boldsymbol{\mu}_i^*)_{i \in I}) \mid \boldsymbol{V}, \boldsymbol{Z}$ and $\boldsymbol{Z} \perp\!\!\!\perp (\boldsymbol{f}, \boldsymbol{f}^*, (\boldsymbol{\mu}_i, \boldsymbol{\mu}_i^*)_{i \in I})$ which holds because $f, (\mu_i)_{i \in I}$ only affect the conditional factors for $\boldsymbol{Y}, \boldsymbol{V}$ respectively, by construction.

Hence, letting $p(\boldsymbol{Y} | \boldsymbol{f}, \boldsymbol{W}, \boldsymbol{V})$ and $p(\boldsymbol{V} \mid (\boldsymbol{\mu}_i)_{i \in I}, \boldsymbol{Z})$ be (conditional) densities w.r.t. Lebesgue measure (which exist under the Gaussian noise models[12] (10)-(11)), by Bayes' rule and the independence of the prior,

$$\mathbb{P}_{\boldsymbol{f}^*, (\boldsymbol{\mu}_i^*)_{i \in I} | \mathcal{X}^n}(A, B) \;\propto$$
$$\int_B \int_{\mathbb{R}^{m \times I}} \int_A \int_{\mathbb{R}^m} p(\boldsymbol{Y} \mid \boldsymbol{f}, \boldsymbol{W}, \boldsymbol{V}) p(\boldsymbol{V} \mid (\boldsymbol{\mu}_i)_{i \in I}, \boldsymbol{Z}) \mathbb{P}(d\boldsymbol{f} \mid \boldsymbol{f}^*) \mathbb{P}(d\boldsymbol{f}^*) \mathbb{P}(d(\boldsymbol{\mu}_i)_{i \in I} \mid (\boldsymbol{\mu}_i^*)_{i \in I}) \mathbb{P}(d(\boldsymbol{\mu}_i^*)_{i \in I})$$
$$\propto \int_A \int_{\mathbb{R}^m} p(\boldsymbol{Y} \mid \boldsymbol{f}, \boldsymbol{W}, \boldsymbol{V}) \mathbb{P}(d\boldsymbol{f} \mid \boldsymbol{f}^*) \mathbb{P}(d\boldsymbol{f}^*) \int_B \int_{\mathbb{R}^{m \times I}} p(\boldsymbol{V} \mid (\boldsymbol{\mu}_i)_{i \in I}, \boldsymbol{Z}) \mathbb{P}(d(\boldsymbol{\mu}_i)_{i \in I} \mid (\boldsymbol{\mu}_i^*)_{i \in I}) \mathbb{P}(d(\boldsymbol{\mu}_i^*)_{i \in I})$$
$$= \mathbb{P}_{\boldsymbol{f}^* | \boldsymbol{V}, \boldsymbol{W}, \boldsymbol{Y}}(A) \otimes \mathbb{P}_{(\boldsymbol{\mu}_i^*)_{i \in I} | \boldsymbol{V}, \boldsymbol{Z}}(B)$$

for any measurable sets $(A, B) \in \mathcal{B}(\mathbb{R}^m) \otimes \mathcal{B}(\mathbb{R}^{I \times m})$. Here $\mathcal{B}(\mathbb{R}^{I \times m}) := \bigotimes_{i \in I} \mathcal{B}(\mathbb{R}^m)$ is the product $\sigma$-algebra on $(\mathbb{R}^m)^I$.

We next further simplify each posterior component. For the posterior on $\boldsymbol{f}^*$, since $k_W, k_V$ are assumed characteristic, their canonical feature maps $\psi_W, \psi_V$ are injective. Defining $\Phi := \left( \psi_W(W_i) \otimes \psi_V(V_i) \right)_{i=1}^n$, the map $(\boldsymbol{W}, \boldsymbol{V}) \mapsto \Phi$ is therefore also injective (coordinate-wise). The $\sigma$-algebras $\sigma(\boldsymbol{W}, \boldsymbol{V})$ and $\sigma(\Phi)$ therefore generate the same information and so conditioning on $\Phi$ is equivalent to conditioning on $(\boldsymbol{W}, \boldsymbol{V})$. Thus, for $\boldsymbol{f}^*$, it suffices to recover the posterior on $\boldsymbol{f}^* \mid \boldsymbol{Y}, \Phi$.

Meanwhile, the posterior on $(\boldsymbol{\mu}^*)_{i \in I}$ factorizes over the co-ordinates $i \in I$. This is easy to see since for any measurable $B \in \mathcal{B}(\mathbb{R}^{I \times m})$,

$$\mathbb{P}\left((\boldsymbol{\mu}_j^*)_{j \geq 1} \in B \,\middle|\, \boldsymbol{V}, \boldsymbol{Z}\right) = \mathbb{P}\left((\boldsymbol{\mu}_j^*)_{j \geq 1} \in B \,\middle|\, (\phi_{V,j}(\boldsymbol{V}))_{j \geq 1}, \boldsymbol{Z}\right)$$
$$= \bigotimes_{j \in I} \mathbb{P}\left(\boldsymbol{\mu}_j^* \in \pi_j[B] \,\middle|\, \phi_{V,j}(\boldsymbol{V}), \boldsymbol{Z}\right),$$

where the first equality holds by the injectivity of $\phi_V$, the second equality holds by the independent noise model for each $\phi_{V,i}(V)$ (11), and $\pi_j$ denotes the projection onto the $j$th coordinate. Thus, it suffices to analyze the posterior distributions $\mu_j \mid \phi_{V,j}(\boldsymbol{V}), \boldsymbol{Z}$ for $j \in I$.

Altogether, this implies that the posterior factorizes as

$$\mathbb{P}_{\boldsymbol{f}^*, (\boldsymbol{\mu}_i^*)_{i \in I} | \mathcal{X}^n}(A, B) = \mathbb{P}_{\boldsymbol{f}^* | \Phi, \boldsymbol{Y}}(A) \otimes \left( \bigotimes_{j \in I} \mathbb{P}_{\boldsymbol{\mu}_j^* | \phi_{V,j}(\boldsymbol{V}), \boldsymbol{z}} \right)$$

Since $m \in \mathbb{N}$ and the test points $(w_1, \ldots, w_m), (v_1, \ldots, v_m)$, and $(z_1, \ldots, z_m)$ were arbitrary, the above finite-dimensional posteriors jointly characterize the posterior distributions $f \mid \Phi, \boldsymbol{Y}$ and $(\mu_i)_{i \in I} \mid \boldsymbol{V}, \boldsymbol{Z}$ of each process, which together define the joint posterior $(f, (\mu_i)_{i \in I}) \mid \mathcal{X}^n$.

**Deriving the Posteriors on $f$ and $\mu$** Note $f$ is an isonormal GP on $\mathcal{H} = \mathcal{H}_W \otimes \mathcal{H}_V$ with covariance $\langle \cdot, \cdot \rangle_{\mathcal{H}}$. Define the $n \times n$ Gram matrix $K_W \odot K_V := K_W \odot K_V$ with $(K_W)_{ij} = k_W(W_i, W_j)$, $(K_V)_{ij} = k_V(V_i, V_j)$, and write for any $h \in \mathcal{H}$

$$h^\top \Phi := [\langle h, \psi_W(W_i) \otimes \psi_V(V_1) \rangle_{\mathcal{H}}, \ldots, \langle h, \psi_W(W_i) \otimes \psi_V(V_n) \rangle_{\mathcal{H}}] \in \mathbb{R}^{1 \times n}.$$

---

[11]By Lemma C.1 the feature map $\phi_V := v \mapsto (\phi_{V,i}(v))_{i \in I}$ is injective. Thus, the conditional independence between $\boldsymbol{V}$ and $\boldsymbol{g} := (\boldsymbol{f}, \boldsymbol{f}^*, (\boldsymbol{\mu}_i^*)_{i \in I})$ is equivalent to the conditional independence between $\phi_V(\boldsymbol{V})$ and $\boldsymbol{g}$. To see this, note by injectivity $\mathbb{P}(\boldsymbol{V} \in A \mid \boldsymbol{g}, (\boldsymbol{\mu}_i)_{i \in I}, \boldsymbol{Z}) = \mathbb{P}(\phi_V(\boldsymbol{V}) \in \phi_V(A) \mid \boldsymbol{g}, (\boldsymbol{\mu}_i)_{i \in I}, \boldsymbol{Z})$. By (11), we have $\phi_{V,i}(V) = \mu_i(Z) + \xi_i$, and so the conditional independence $\phi_V(\boldsymbol{V}) \perp\!\!\!\perp \boldsymbol{g} \mid (\boldsymbol{\mu}_i)_{i \in I}, \boldsymbol{Z}$ holds. This means $\mathbb{P}(\boldsymbol{V} \in A \mid \boldsymbol{g}, (\boldsymbol{\mu}_i)_{i \in I}, \boldsymbol{Z}) = \mathbb{P}(\phi_V(\boldsymbol{V}) \in \phi_V(A) \mid (\boldsymbol{\mu}_i)_{i \in I}, \boldsymbol{Z}) = \mathbb{P}(\boldsymbol{V} \in \phi_V^{-1}\{\phi_V(A)\} \mid (\boldsymbol{\mu}_i)_{i \in I}, \boldsymbol{Z}) = \mathbb{P}(\boldsymbol{V} \in A \mid (\boldsymbol{\mu}_i)_{i \in I}, \boldsymbol{Z})$.

[12]Note that since $\phi_{V,i}(V) \mid Z, (\mu_i)_{i \in I}$ is continuous, so is $V \mid Z, (\mu_i)_{i \in I}$. Indeed, assume for contradiction that $V \mid Z, (\mu_i)_{i \in I}$ has an atom at $v_0$, i.e. $\mathbb{P}(V = v_0 \mid Z, (\mu_i)_{i \in I}) = p > 0$. Since $\phi_{V,i}$ is deterministic, $\{V = v_0\} \subseteq \{\phi_{V,i}(V) = \phi_{V,i}(v_0)\} \Rightarrow \mathbb{P}(\phi_{V,i}(V) = \phi_{V,i}(v_0) \mid Z, (\mu_i)_{i \in I}) \geq p > 0$, so $\phi_{V,i}(V) \mid Z, (\mu_i)_{i \in I}$ has an atom at $\phi_{V,i}(v_0)$, contradicting the assumption that it is continuous. Hence, $V \mid Z, (\mu_i)_{i \in I}$ must also be atomless.

where we define $(\Phi^\top h) = (h^\top \Phi)^\top$. By Lemma C.3, under the observation model (10) with noise variance $\sigma^2$, the GP posterior is

$$f \mid \boldsymbol{Y}, \Phi \; \sim \; \mathcal{GP}(m, v),$$

with

$$m(h) \;=\; h^\top \Phi \left( K_W \odot K_V + \sigma^2 I_n \right)^{-1} \boldsymbol{Y},$$
$$v(h, h') \;=\; \langle h, h' \rangle_{\mathcal{H}} \; - \; h^\top \Phi \left( K_W \odot K_V + \sigma^2 I_n \right)^{-1} (\Phi^\top h').$$

Similarly, by the noise model (11) and GP prior on each $\mu_i$, the posterior on each $\mu_i$ is also given by

$$\mu_j \mid \phi_{V,j}(\boldsymbol{V}), \boldsymbol{Z} \; \sim \; \mathcal{GP}(m_j, v_j),$$

with

$$m_j(z) = \boldsymbol{k}_Z(z)^\top \left( K_Z + \eta^2 I_n \right)^{-1} \phi_{V,j}(\boldsymbol{V}),$$
$$v_j(z, z') = k_Z(z, z') - \boldsymbol{k}_Z(z)^\top \left( K_Z + \eta^2 I_n \right)^{-1} \boldsymbol{k}_Z(z).$$

where $\boldsymbol{k}_Z(z) = [k_Z(z, Z_1), \ldots, k_Z(z, Z_n)]^\top$.

**Deriving Posterior Moments of $\gamma$**   Now that we have the required posteriors, we combine them to derive the posterior moments of $\gamma(w, z)$ using the $L^2$–convergent representation of the model

$$\gamma(w, z) = f\big(\psi_W(w) \otimes \mu(z)\big),$$

where recall that[13] $\mu(z) := \sum_{i \in I} \lambda_{V,i}^{\frac{1}{2}} \mu_i(z) e_i$ and $e_i := \lambda_{V,i}^{\frac{1}{2}} \phi_{V,i}$ satisfies $\langle e_i, e_j \rangle_{\mathcal{H}_V} = \delta_{i=j}$. In particular, since $f \perp\!\!\!\perp \mu \mid \mathcal{X}^n$, we will use the fact that

$$\gamma(w, z) \mid \mathcal{X}^n, \mu(z) =_d f\big(\psi_W(w) \otimes \mu(z)\big) \mid \mathcal{X}^n$$

where above we treat $\mu(z)$ as fixed temporarily. Thus, by the law of total expectation and variance, the posterior moments of $\gamma(w, z)$ are given as

$$\mathbb{E}[\gamma(w, z) \mid \mathcal{X}^n] = \mathbb{E}[m(\phi_W(w) \otimes \mu(z))] \tag{69}$$
$$\mathbb{V}ar[\gamma(w, z) \mid \mathcal{X}^n] = \mathbb{V}ar[m(\phi_W(w) \otimes \mu(z))] + \mathbb{E}[v(\phi_W(w) \otimes \mu(z), \phi_W(w) \otimes \mu(z))] \tag{70}$$

All that remains is to calculate each of the terms using the posterior moments of $f$ and $\mu(z)$. For the mean note we can rewrite the posterior mean of $f$ evaluated at this input as

$$m(\phi_W(w) \otimes \mu(z)) = \mu(z)^\top \Phi_V \, \boldsymbol{\alpha}(w) \tag{71}$$

where $\boldsymbol{\alpha}(w) = D(w)(K_W \odot K_V + \sigma^2 I_n)^{-1} \boldsymbol{Y}$, $D(w) = \mathrm{diag}(k_W(w, W_1), \ldots, k_W(w, W_n)) \in \mathbb{R}^{n \times n}$ and $\mu(z)^\top \Phi_V = [\langle \mu(z), \psi_V(V_1) \rangle_{\mathcal{H}_V}, \ldots, \langle \mu(z), \psi_V(V_n) \rangle_{\mathcal{H}_V}]^\top \in \mathbb{R}^{1 \times n}$. Using this definition, the posterior mean is then

$$\mathbb{E}[\gamma(w, z) \mid \mathcal{X}^n] = \mathbb{E}[\mu(z)^\top \Phi_V \, \boldsymbol{\alpha}(w) \mid \mathcal{X}^n]$$
$$= \mathbb{E}[\mu(z)^\top \Phi_V \mid \mathcal{X}^n] \, \boldsymbol{\alpha}(w) \tag{72}$$

Now, we can straightforwardly recover the required expectation from the posterior mean of each $\mu_i$. In particular, for any

---

[13]By standard results on Gaussian measures in separable Hilbert spaces, $\mu(z)$ is a well-defined Gaussian random element in $\mathcal{H}_V$; see, e.g., Chapter 5 of Kukush (2020).

$v \in \mathcal{V}$, the mean evaluation of $\mu(z)$ on $\psi_V(v)$ is

$$
\begin{aligned}
\mathbb{E}[\langle \psi_V(v), \mu(z)\rangle_{\mathcal{H}_V} \mid \mathcal{X}^n] &= \mathbb{E}\left[\left\langle \sum_{i \in I} \lambda_{V,i}^{\frac{1}{2}} \phi_{V,i}(v) e_i, \sum_{i \in I} \lambda_{V,i}^{\frac{1}{2}} \mu_i(z) e_i \right\rangle_{\mathcal{H}_V} \;\middle|\; \mathcal{X}^n \right] \\
&= \mathbb{E}\left[\sum_{i \in I} \lambda_{V,i} \phi_{V,i}(v) \mu_i(z) \;\middle|\; \mathcal{X}^n\right] \\
&= \sum_{i \in I} \lambda_{V,i} \phi_{V,i}(v) \mathbb{E}[\mu_i(z) \mid \mathcal{X}^n] \\
&= \sum_{i \in I} \lambda_{V,i} \phi_{V,i}(v) \phi_{V,i}(\boldsymbol{V})(K_Z + \eta^2 I_n)^{-1} \boldsymbol{k}_Z(z) \\
&= \boldsymbol{k}_V(v)(K_Z + \eta^2 I_n)^{-1} \boldsymbol{k}_Z(z)
\end{aligned}
$$

where the second line follows from the orthonormality of $(e_i)_{i \in I}$, and the third line follows from Fubini's theorem.[14] Substituting this into (72) we get

$$
\mathbb{E}[\gamma(w, z)|\mathcal{X}^n] = \boldsymbol{k}_Z(z)^\top \left(K_Z + I\eta^2\right)^{-1} K_V \boldsymbol{\alpha}(w) \tag{73}
$$

which is the form of the expectation in the theorem.

Now we turn to the variance. For the first term, we have

$$
Var[m(\phi_W(w) \otimes \mu(z)) \mid \mathcal{X}^n] = \underbrace{\mathbb{E}[m(\phi_W(w) \otimes \mu(z))^2|\mathcal{X}^n]}_{I_1} - \underbrace{\mathbb{E}[m(\phi_W(w) \otimes \mu(z)) \mid \mathcal{X}^n]^2}_{I_2} \tag{74}
$$

Using the definition of $m(\phi_W(w) \otimes \mu(z))$ in (71) and basic properties of tensor-product RKHS's, we can expand $I_1$ as follows

$$
\begin{aligned}
I_1 &= \mathbb{E}[\mu(z)^\top \Phi_V \boldsymbol{\alpha}(w)\boldsymbol{\alpha}(w)^\top \Phi_V^\top \mu(z)|\mathcal{X}^n] \\
&= \sum_{l,m=1}^{n} \alpha_{n,m}(w)\mathbb{E}[\langle \mu(z), \psi_V(V_l)\rangle_{\mathcal{H}_V}\langle \mu(z), \psi_V(V_m)\rangle_{\mathcal{H}_V}|\mathcal{X}^n] \\
&= \sum_{l,m=1}^{n} \alpha_{n,m}(w)\mathbb{E}[\langle \mu(z) \otimes \mu(z), \psi_V(V_l) \otimes \psi_V(V_m)\rangle_{HS(\mathcal{H}_V, \mathcal{H}_V)}|\mathcal{X}^n] \\
&= \sum_{l,m=1}^{n} \alpha_{n,m}(w)\langle \mathbb{E}[\mu(z) \otimes \mu(z)|\mathcal{X}^n], \psi_V(V_l) \otimes \psi_V(V_m)\rangle_{HS(\mathcal{H}_V, \mathcal{H}_V)} \\
&= \sum_{l,m=1}^{n} \alpha_{n,m}(w)\langle \mathbb{E}[\mu(z)|\mathcal{X}^n] \otimes \mathbb{E}[\mu(z)|\mathcal{X}^n] + \mathbb{C}ov[\mu(z)|\mathcal{X}^n], \psi_V(V_l) \otimes \psi_V(V_m)\rangle_{HS(\mathcal{H}_V, \mathcal{H}_V)} \\
&= \sum_{l,m=1}^{n} \alpha_{n,m}(w)\left(\langle \mathbb{E}[\mu(z)|\mathcal{X}^n], \psi_V(V_l)\rangle_{\mathcal{H}_V}\langle \mathbb{E}[\mu(z)|\mathcal{X}^n], \psi_V(V_m)\rangle_{\mathcal{H}_V} + \langle \psi_V(V_l), \mathbb{C}ov[\mu(z)|\mathcal{X}^n]\psi_V(V_m)\rangle_{\mathcal{H}_V}\right) \\
&= \underbrace{\mathbb{E}[\mu(z)|\mathcal{X}^n]^\top \Phi_V \boldsymbol{\alpha}(w)\boldsymbol{\alpha}(w)^\top \Phi_V^\top \mathbb{E}[\mu(z)|\mathcal{X}^n, Z]}_{I_{11}} + \underbrace{Tr[\boldsymbol{\alpha}(w)\boldsymbol{\alpha}(w)^\top \Phi_V^\top \mathbb{C}ov[\mu(z)|\mathcal{X}^n]\Phi_V]}_{I_{12}}
\end{aligned} \tag{75}
$$

where $HS(\mathcal{H}_V, \mathcal{H}_V)$ is the space of Hilbert-Schmidt operators $\mathcal{H}_V \to \mathcal{H}_V$, and $\mathbb{C}ov[\mu(z) \mid \mathcal{X}^n] : \mathcal{H}_V \to \mathcal{H}_V$ is the covariance operator

$$
\mathbb{C}ov[\mu(z) \mid \mathcal{X}^n] = \mathbb{E}\left[\left(\mu(z) - \mathbb{E}[\mu(z) \mid \mathcal{X}^n]\right) \otimes \left(\mu(z) - \mathbb{E}[\mu(z) \mid \mathcal{X}^n]\right) \mid \mathcal{X}^n\right]
$$

---

[14]Fubini's theorem lets us swap the infinite sum with the expectation if $\sum_{i \in I} |\lambda_{V,i}\phi_{V,i}(v)\mathbb{E}[\mu_i(z) \mid \mathcal{X}^n]| < \infty$. By Cauchy Schwartz, this is bounded by $\left(\sum_{i \in I} \lambda_{V,i}\phi_{V,i}(v)^2\right)^{\frac{1}{2}} \left(\sum_{i \in I} \lambda_{V,i}\left(m_i(z)^2 + v_i(z, z)\right)\right)^{\frac{1}{2}}$. By Mercer's Theorem, the first term is $k_V(v, v)$, which is bounded by definition. Since $v_i(z, z) \leq k_Z(z, z)$ and $\sum_j \lambda_{V,j} m_j(z)^2 = \sum_{i,l=1}^{n} \sum_{j \in I} \lambda_{V,i}\phi_{V,i}(V_i)\phi_{V,i}(V_l)B_{i,l}(z) \leq n^2 \sup_{z,z'} k_Z(z, z')$, the second term is also finite.

Note, we are able to pass the expectation inside the inner product in (75) because the operator $A \mapsto \mathbb{E}[\langle \mu(z) \otimes \mu(z), A \rangle_{HS(\mathcal{H}_V, \mathcal{H}_V)} | \mathcal{X}^n]$ is bounded. In particular by Cauchy-Schwartz and Jensen's inequality,

$$\mathbb{E}[\langle \mu(z) \otimes \mu(z), A \rangle_{HS(\mathcal{H}_V, \mathcal{H}_V)} | \mathcal{X}^n] \leq \sqrt{\mathbb{E}[\|\mu(z) \otimes \mu(z)\|^2_{HS(\mathcal{H}_V, \mathcal{H}_V)} | \mathcal{X}^n]} \|A\|_{HS(\mathcal{H}_V, \mathcal{H}_V)}$$

(Hilbert-Schmidt norm)
$$= \sqrt{\mathbb{E}\left[\sum_{i \in I} \langle e_i, \mu(z) \rangle^2_{\mathcal{H}^2_V} \bigg| \mathcal{X}^n\right]} \|A\|_{HS(\mathcal{H}_V, \mathcal{H}_V)}$$

(Expanding $\mu(z)$ into co-ordinates)
$$= \sqrt{\mathbb{E}\left[\sum_{i \in I} \lambda_{V,i} \mu_i(z)^2 \bigg| \mathcal{X}^n\right]} \|A\|_{HS(\mathcal{H}_V, \mathcal{H}_V)}$$

$(\sum_{i \in I} \lambda_{V,i} < \infty)$
$$= \sqrt{\sum_{i \in I} \lambda_{V,i} \mathbb{E}[\mu_i(z)^2 | \mathcal{X}^n]} \|A\|_{HS(\mathcal{H}_V, \mathcal{H}_V)}$$

(Expanding $[\mu_i(z)^2]$)
$$\leq \sqrt{\sum_{i \in I} (\lambda_{V,i} m_i(z)^2 + \lambda_{V,i} \mathbb{V}ar[\mu_i(z) | \mathcal{X}^n]} \|A\|_{HS(\mathcal{H}_V, \mathcal{H}_V)}$$

(Prior variance > Posterior variance)
$$\leq \sqrt{\sum_{i \in I} (\lambda_{V,i} m_i(z)^2 + \lambda_{V,i} k_Z(z,z))} \|A\|_{HS(\mathcal{H}_V, \mathcal{H}_V)}$$

(Definition of $m_i(z)$)
$$= \sqrt{\boldsymbol{\beta}(z)^\top K_V \boldsymbol{\beta}(z) + \left(\sum_{i \in I} \lambda_{V,i}\right) k_Z(z,z)} \|A\|_{HS(\mathcal{H}_V, \mathcal{H}_V)}$$

$$\leq C \|A\|_{HS(\mathcal{H}_V, \mathcal{H}_V)}$$

where $C < \infty$ is implied by Assumption A.3 and Assumption A.4.

Now, since $I_{11} = I_2$, we have $\mathbb{V}ar[m(\phi_W(w) \otimes \mu(z)) | \mathcal{X}^n] = I_{12}$. To calculate this term, we require $\mathbb{C}ov[\mu(z) | \mathcal{X}^n]$, which we derive below. Recall the definitions

$$\mu(z) = \sum_{i \in I} \lambda_{V,i}^{1/2} \mu_i(z) e_i, \qquad \mu_i(z) := \lambda_{V,i}^{-1/2} \langle \mu(z), e_i \rangle_{\mathcal{H}_V}.$$

By independence of coordinates,

$$\mathbb{C}ov(\langle \mu(z), h \rangle, \langle \mu(z'), h' \rangle | \mathcal{X}^n) = \mathbb{C}ov\left(\sum_{i \in I} h_i \lambda_{V,i}^{\frac{1}{2}} \mu_i(z), \sum_{i \in I} h_i \lambda_{V,i}^{\frac{1}{2}} \mu_i(z)\right)$$

To show we can exchange the covariance operation with the infinite sums, we define the truncated sums $X_N := \sum_{i=1}^N \lambda_{V,i}^{1/2} \tilde{\mu}_i(z) h_i$ and $Y_M := \sum_{j=1}^M \lambda_{V,j}^{1/2} \tilde{\mu}_j(z') h'_j$, where $\tilde{\mu}_i(z) = \mu_i(z) - m_i(z)$ is a centered version of $\mu_i(z)$. For fixed $N, M$,

$$\mathbb{C}ov(X_N, Y_M | \mathcal{X}^n) = \sum_{i=1}^N \sum_{j=1}^M \lambda_{V,i}^{1/2} \lambda_{V,j}^{1/2} h_i h'_j \mathbb{C}ov(\mu_i(z), \mu_j(z') | \mathcal{X}^n).$$

where we use the fact that $\mathbb{C}ov(\tilde{\mu}_i(z), \tilde{\mu}_j(z') | \mathcal{X}^n) = \mathbb{C}ov(\mu_i(z), \mu_j(z') | \mathcal{X}^n)$. By conditional independence of coordinates, $\mathbb{C}ov(\mu_i(z), \mu_j(z') | \mathcal{X}^n) = \delta_{ij} v_i(z, z')$, so

$$\mathbb{C}ov(X_N, Y_M | \mathcal{X}^n) = \sum_{i=1}^{\min\{N,M\}} \lambda_{V,i} v_i(z, z') h_i h'_i.$$

For $M > N$, the $L^2(\mathbb{P}(\bullet | \mathcal{X}^n)$ norm of $X_M - X_N$ can be written as

$$\mathbb{E}((X_M - X_N)^2 | \mathcal{X}^n) = \mathbb{V}ar(X_M - X_N | \mathcal{X}^n) = \mathbb{V}ar\left(\sum_{i=N+1}^M \lambda_{V,i}^{1/2} \tilde{\mu}_i(z) h_i \bigg| \mathcal{X}^n\right) = \sum_{i=N+1}^M \lambda_{V,i} v_i(z, z) h_i^2,$$

where we used finite-sum bilinearity and $\mathbb{C}ov(\tilde{\mu}_i, \tilde{\mu}_j \mid \mathcal{X}^n) = 0$ for $i \neq j$. Since $h_i = \langle h, e_i \rangle_{\mathcal{H}} \leq \|h\|_{\mathcal{H}}$ and $v_i(z, z) \leq k_Z(z, z) \leq \tau$ for some $\tau \in \mathbb{R}_+$, $\sum_{i \in I} \lambda_{V,i} v_i(z, z) h_i^2 < \infty$, so the tail sum tends to 0 and $(X_N)_{N \geq 1}$ is Cauchy in $L^2(\mathbb{P}(\bullet \mid \mathcal{X}^n))$. Since this is a Hilbert space, the Cauchy criterion suffices for $X_N \to X$ in $L^2(\mathbb{P}(\bullet \mid \mathcal{X}^n))$. The same argument gives $Y_M \to Y$ in $L^2(\mathbb{P}(\bullet \mid \mathcal{X}^n))$.

Using the fact that the covariance is a continuous bilinear form on $L^2$ and the triangle inequality, we have the general bound,

$$
\begin{aligned}
|\mathbb{C}ov(U, V) - \mathbb{C}ov(U', V')| &= |\mathbb{C}ov(U - U', V) + \mathbb{C}ov(U', V - V')| \\
&\leq \|U - U'\|_{L^2} \|V\|_{L^2} + \|V - V'\|_{L^2} \|U'\|_{L^2},
\end{aligned}
$$

for any $U, U', V, V' \in L^2(\mathbb{P})$. Thus, defining the limits $X := \sum_{i \in I} h_i \lambda_{V,i}^{\frac{1}{2}} \mu_i(z)$, $Y := \sum_{i \in I} h_i \lambda_{V,i}^{\frac{1}{2}} \mu_i(z')$, we have $\mathbb{C}ov(X_N, Y_M \mid \mathcal{X}^n) \to \mathbb{C}ov(X, Y \mid \mathcal{X}^n)$ as $N, M \to \infty$, yielding

$$
\mathbb{C}ov\big(\langle \mu(z), h \rangle, \langle \mu(z'), h' \rangle \mid \mathcal{X}^n\big) = \sum_{i \in I} \lambda_{V,i} \, v_i(z, z') \, h_i h_i',
$$

By Riesz's representation theorem, this identifies the operator-valued posterior covariance as

$$
\mathbb{C}ov\big(\mu(z), \mu(z') \mid \mathcal{X}^n\big) = \sum_{i \in I} \lambda_{V,i} \, v_i(z, z') \, e_i \otimes e_i.
$$

To use this definition of the operator to compute the term $\Phi_V^\top \mathbb{C}ov(\mu(z) \mid \mathcal{X}^n) \Phi_V$ in $I_{12}$, we will use the fact that $v_i(z, z') := \hat{k}_Z(z, z') := k_Z(z, z') - \boldsymbol{k}_Z(z)^T (K_Z + \eta^2 I_n)^{-1} \boldsymbol{k}_Z(z')$ and $\langle e_i, \psi_V(v) \rangle_{\mathcal{H}} = \lambda_{V,i}^{\frac{1}{2}} \phi_{V,i}(v)$. In particular,

$$
\begin{aligned}
[\mathbb{C}ov(\mu(z) \mid \mathcal{X}^n) \Phi_V]_j &= \left( \sum_{i \in I} \lambda_{V,i} \, v_i(z, z') \, e_i \otimes e_i \right) \psi_V(V_j) = \sum_{i \in I} \lambda_{V,i} \, v_i(z, z') \, e_i \langle e_i, \psi_V(V_j) \rangle_{\mathcal{H}} \\
&= \sum_{i \in I} \lambda_{V,i}^{\frac{3}{2}} \, v_i(z, z') \phi_{V,i}(V_j) e_i \\
&= \hat{k}_Z(z, z) \sum_{i \in I} \lambda_{V,i}^{\frac{3}{2}} \phi_{V,i}(V_j) e_i
\end{aligned}
$$

which implies that

$$
\begin{aligned}
[\Phi_V^\top \mathbb{C}ov(\mu(z) \mid \mathcal{X}^n) \Phi_V]_{l,j} &= \psi_V(V_l)^\top \left( \sum_{i \in I} \lambda_{V,i} \, v_i(z, z') \, e_i \otimes e_i \right) \psi_V(V_j) \\
&= \sum_{i \in I} \lambda_{V,i} \, v_i(z, z') \, \langle e_i, \psi_V(V_l) \rangle_{\mathcal{H}_V} \langle e_i, \psi_V(V_j) \rangle_{\mathcal{H}_V} \\
&= \hat{k}_Z(z, z) \sum_{i \in I} \lambda_{V,i}^2 \, \phi_{V,i}(v') \phi_{V,i}(v)
\end{aligned}
$$

Now, recall that by Mercer's theorem, $k_V(v, v') = \sum_{i \in \mathbb{N}} \lambda_{V,i} \phi_{V,i}(v) \phi_{V,i}(v')$. Substituting this into $k_V(V_i, t) k_V(t, V_j)$ and using orthonormality of $\{\phi_{V,i}\}_{i \in I}$ in $L^2(\nu)$ w.r.t. spectral measure $\nu \in \mathcal{P}(\mathcal{V})$ yields

$$
\int_{\mathcal{V}} k_V(V_i, t) \, k_V(t, V_j) \, d\nu(t) = \sum_{l \in \mathbb{N}} \lambda_l^2 \, \phi_l(V_i) \phi_l(V_j),
$$

From here, it becomes clear that

$$
\begin{aligned}
\mathbb{V}ar[m(\phi_W(w) \otimes \mu(z)) \mid \mathcal{X}^n] = I_{12} &= Tr[\boldsymbol{\alpha}(w) \boldsymbol{\alpha}(w)^\top \Phi_V^\top \mathbb{C}ov[\mu(z) \mid \mathcal{X}^n] \Phi_V] \\
&= Tr[\boldsymbol{\alpha}(w) \boldsymbol{\alpha}(w)^\top \tilde{K}_V] \hat{k}_Z(z, z)
\end{aligned}
$$

Where $[\tilde{K}_V]_{i,j} = \sum_{l \in I} \lambda_l^2 \phi_l(V_i)\phi_l(V_j) = \int_{\mathcal{V}} k(V_i, t)k(t, V_j)d\nu(t)$. This is the first part of $S_2$. Now, using the definition of the posterior variance of $f$, we can write the second term in the law of total variance decomposition (70) as

$$\mathbb{E}[v(\phi_W(w) \otimes \mu(z), \phi_W(w) \otimes \mu(z))|\mathcal{X}^n]$$
$$= \mathbb{E}\left[\langle\phi_W(w) \otimes \mu(z), \phi_W(w) \otimes \mu(z)\rangle_{\mathcal{H}} - (\phi_W(w) \otimes \mu(z))^\top \Phi_V \left(K_W \odot K_V + \sigma^2 I_n\right)^{-1} (\Phi_V^\top(\phi_W(w) \otimes \mu(z))) \mid \mathcal{X}^n\right]$$
$$= \mathbb{E}\left[k_W(w,w)\langle\mu(z), \mu(z)\rangle_{\mathcal{H}_V} - \langle\mu(z), \left(\Phi_V D(w)\left(K_W \odot K_V + \sigma^2 I_n\right)^{-1} D(w)\Phi_V^\top\right)\mu(z)\rangle_{\mathcal{H}_V} \mid \mathcal{X}^n\right]$$
$$= k_W(w,w)\mathbb{E}\left[\langle\mu(z), B\mu(z)\rangle_{\mathcal{H}_V} \mid \mathcal{X}^n\right]$$
$$= \mathbb{E}[Tr[B\mu(z) \otimes \mu(z)^\top]|\mathcal{X}^n]$$

where $B = Ik_W(w,w) - \Phi_V A(w)\Phi_V^\top$, $A(w) = D(w)\left(K_W \odot K_V + I\sigma^2\right)^{-1} D(w)$, and $D(w)$ is defined in the Theorem.

Now, recall that for any random element $X \in \mathcal{H}_V$ and bounded linear $A : \mathcal{H} \to \mathcal{H}$ we have $\mathbb{E}[AX] = A\mathbb{E}[X]$ whenever $\mathbb{E}\|X\|_{\mathcal{H}_V} < \infty$. Therefore, in our case we can exchange the expectation with $Tr \circ B$ as long as (i) $A \mapsto Tr[BA]$ is bounded and (ii) $\mathbb{E}[\|\mu(z) \otimes \mu(z)\|_{HS(\mathcal{H}_V, \mathcal{H}_V)}|\mathcal{X}^n] < \infty$. For (i), first note that $A \mapsto Tr[A]$ is bounded on the space $\mathcal{B}_1(\mathcal{H}_V)$ of trace class operators $\mathcal{H}_V \to \mathcal{H}_V$. Moreover, from the structure of $B$ it is obvious that $Tr[BA] \leq k_W(w,w)Tr[A] \leq cTr[A]$ for any trace class $A$, where $c < \infty$ by Assumption A.3. Combining these two facts, it suffices to show that $\mu(z) \otimes \mu(z) \in \mathcal{B}_1(\mathcal{H}_V)$ almost surely for (i) to hold. This is true because

$$Tr[\mu(z) \otimes \mu(z)] = \sum_{i \in I}\langle e_i, (\mu(z) \otimes \mu(z)])e_i\rangle_{\mathcal{H}_V}$$
$$= \sum_{i \in I}\langle e_i, \mu(z)\rangle_{\mathcal{H}_V}^2$$
$$= \sum_{i \in I}\lambda_{V,i}\mu_i(z)^2 < \infty$$

with posterior probability 1, since we already showed earlier that $\mathbb{E}[\sum_{i \in I}\lambda_{V,i}\mu_i(z)^2|\mathcal{X}^n] < \infty$. We also already showed that (ii) holds earlier in the proof. Therefore, we can exchange the expectation with $Tr \circ B$, which gives us

$$\mathbb{E}[\mathbb{V}ar[\gamma(w,z)|\mathcal{X}^n, L]|\mathcal{X}^n] = Tr[B\mathbb{E}[\mu(z) \otimes \mu(z)^\top]|\mathcal{X}^n]$$
$$= Tr[B(\mathbb{E}[\mu(z)|\mathcal{X}^n] \otimes \mathbb{E}[\mu(z)|\mathcal{X}^n]^\top + \mathbb{C}ov[\mu(z)|\mathcal{X}^n]))]$$
$$= Tr[B(\Phi_V^\top \boldsymbol{\beta}(z)\boldsymbol{\beta}(z)^\top \Phi_V + \mathbb{C}ov[\mu(z)|\mathcal{X}^n])]$$
$$= \boldsymbol{\beta}(z)^\top \Phi_V B\Phi_V^\top \boldsymbol{\beta}(z) + Tr[B\mathbb{C}ov[\mu(z)|\mathcal{X}^n]]$$

where $\boldsymbol{\beta}(z) := (K_Z + \eta^2 I_n)^{-1}\boldsymbol{k}_Z(z)$ Substituting in the definition of $B$ and $\mathbb{C}ov[\mu(z)|\mathcal{X}^n]$ gives

$$\boldsymbol{\beta}(z)^\top \Phi_V^\top B\Phi_V \boldsymbol{\beta}(z) = \boldsymbol{\beta}(z)^\top \left(K_V k_W(w,w) - K_V A(w)K_V\right)\boldsymbol{\beta}(z)$$
$$Tr[B\mathbb{C}ov[\mu(z)|\mathcal{X}^n]] = k_W(w,w)\sum_{i \in I}\lambda_i \hat{k}_Z(z,z) - Tr[A(w)\tilde{K}_V]\hat{k}_Z(z,z)$$

Re-arranging the terms we finally get $\mathbb{V}ar[\gamma(w,z)|\mathcal{X}^n] = S_1 + S_2 + S_3$ where

$$S_1 = \boldsymbol{\beta}(z)^\top K_V(Ik_W(w,w) - A(w)K_V)\boldsymbol{\beta}(z)$$
$$S_2 = \hat{k}_Z(z,z)(Tr[\tilde{K}_V(\boldsymbol{\alpha}(w)\boldsymbol{\alpha}(w)^\top - A(w))])$$
$$S_3 = \left(\sum_{i \in I}\lambda_i\right)k_W(w,w)\hat{k}_Z(z,z))$$

$\square$

*Remark* C.2. For the case $W = \emptyset$, the posterior moments in Theorem 4.1 and its modification (Algorithm 3) for the the causal data fusion setting discussed in Appendix A.3) are identical except where now $K_W = I$, $k_W(w,w) = 1$ and $\boldsymbol{k}_W(w) = \mathbf{1}$.

C.2.3. THEOREM 4.1 FOR INCREMENTAL EFFECTS

**Theorem C.5** (Posterior moments of incremental effects). *Under the conditions of Theorem 4.1, we have that*

$$\mathbb{E}[\gamma(w,z) - \gamma(w',z')|\mathcal{X}^n] = \mathbb{E}[\gamma(w,z)|\mathcal{X}^n] - \mathbb{E}[\gamma(w',z')|\mathcal{X}^n]$$
$$\mathbb{V}ar[\gamma(w,z) - \gamma(w',z')] = \mathbb{V}ar[\gamma(w,z)|\mathcal{X}^n] + \mathbb{V}ar[\gamma(w',z')|\mathcal{X}^n] - 2\mathbb{C}ov[\gamma(w,z),\gamma(w',z')|\mathcal{X}^n]$$

$\mathbb{E}[\gamma(w,z)|\mathcal{X}^n]$, $\mathbb{V}ar[\gamma(w,z)|\mathcal{X}^n]$ *are defined as in Theorem 4.1 and*

$$Cov[\gamma(w,z),\gamma(w',z')|\mathcal{X}^n] = C_1 + C_2 + C_3$$

*where*

$$C_1 = \boldsymbol{\beta}(z)^\top K_V(Ik_W(w,w') - A(w,w')K_V)\boldsymbol{\beta}(z')$$
$$C_2 = \hat{k}_Z(z,z')(Tr[\tilde{K}_V(\boldsymbol{\alpha}(w)\boldsymbol{\alpha}(w')^\top - A(w,w'))])$$
$$C_3 = \left(\sum_{i \in I} \lambda_i\right) k_W(w,w')\hat{k}_Z(z,z'))$$

*and all quantities except* $A(w,w') := D(w)(K_W \odot K_V + \sigma^2 I)^{-1}D(w')$ *are defined as in Theorem 4.1.*

*Proof.* To prove the result, it suffices to derive the form of the covariance, which we do using the law of total covariance,

$$\mathbb{C}ov[\gamma(w,z),\gamma(w',z') \mid \mathcal{X}^n] = \mathbb{C}ov[m(\psi_W(w) \otimes \mu(z)), m(\psi_W(w') \otimes \mu(z')) \mid \mathcal{X}^n]$$
$$+ \mathbb{E}[v(\psi_W(w) \otimes \mu(z), \psi_W(w') \otimes \mu(z')) \mid \mathcal{X}^n]$$

We compute these terms using near-identical steps to the proof of the posterior variance expression in Theorem 4.1. For the first term, we have

$$\mathbb{C}ov[m(\psi_W(w) \otimes \mu(z)), m(\psi_W(w') \otimes \mu(z')) \mid \mathcal{X}^n] = \underbrace{\mathbb{E}[m(\psi_W(w) \otimes \mu(z))m(\psi_W(w') \otimes \mu(z'))|\mathcal{X}^n]}_{I_1}$$
$$- \underbrace{\mathbb{E}[m(\psi_W(w) \otimes \mu(z)) \mid \mathcal{X}^n]\mathbb{E}[m(\psi_W(w') \otimes \mu(z')) \mid \mathcal{X}^n]}_{I_2}$$

As before, we can expand $I_1$ as

$$I_1 = \mathbb{E}[\mu(z)^\top \Phi_V \boldsymbol{\alpha}(w)\boldsymbol{\alpha}(w')^\top \Phi_V^\top \mu(z')|\mathcal{X}^n, Z]$$
$$= \underbrace{\mathbb{E}[\mu(z)|\mathcal{X}^n]\Phi_V^\top \boldsymbol{\alpha}(w)\boldsymbol{\alpha}(w')^\top \Phi_V^\top \mathbb{E}[\mu(z')|\mathcal{X}^n, Z]}_{I_{11}} + \underbrace{Tr[\boldsymbol{\alpha}(w)\boldsymbol{\alpha}(w')^\top \Phi_V^\top \mathbb{C}ov[\mu(z'),\mu(z)|\mathcal{X}^n]\Phi_V]}_{I_{12}}$$

Since $I_{11} = I_2$, we have $\mathbb{C}ov[m(\psi_W(w) \otimes \mu(z)), m(\psi_W(w') \otimes \mu(z'))|\mathcal{X}^n] = I_{12}$. We can compute this term using the definition of the covariance operator derived in the proof of Theorem 4.1,

$$\mathbb{C}ov[\mu(z),\mu(z') \mid \mathcal{X}^n] = \hat{k}_Z(z,z') \sum_{i \in I} \lambda_{V,i} e_i \otimes e_i$$

where recall from Theorem 4.1 that $\hat{k}_Z(z,z') = k_Z(z,z) - \boldsymbol{k}_Z(z)^T \boldsymbol{\beta}(z)$ is the posterior covariance between $(\mu_i(z),\mu_i(z'))$ and $e_i := \lambda_{V,i}^{\frac{1}{2}}\phi_{V,i}$ for each $i \in I$. Using the same steps to compute $\Phi_V^\top \mathbb{C}ov[\mu(z),\mu(z') \mid \mathcal{X}^n]\Phi_V$ as in the proof of Theorem 4.1, we get

$$\mathbb{C}ov[m(\psi_W(w) \otimes \mu(z)), m(\psi_W(w') \otimes \mu(z'))|\mathcal{X}^n] = Tr[\boldsymbol{\alpha}(w)\boldsymbol{\alpha}(w')^\top \Phi_V^\top \mathbb{C}ov[\mu(z'),\mu(z) \mid \mathcal{X}^n]\Phi_V]\hat{k}_Z(z,z') \quad (76)$$
$$= Tr[\boldsymbol{\alpha}(w)\boldsymbol{\alpha}(w')^\top \tilde{K}_V] \quad (77)$$

where $[\tilde{K}_V]_{i,j} = \sum_{l \in I} \lambda_l^2 \phi_l(V_i)\phi_l(V_j) = \int_{\mathcal{V}} k(V_i,t)k(t,V_j)d\nu(t)$. This is the first part of $S_2$. Now for the second term, we have

$$\mathbb{E}[\mathbb{C}ov[\gamma(w,z),\gamma(w',z')|\mathcal{X}^n, L]|\mathcal{X}^n] = Tr[B\mathbb{E}[\mu(z') \otimes \mu(z)^\top|\mathcal{X}^n]] \quad (78)$$
$$= Tr[B(\mathbb{E}[\mu(z')|\mathcal{X}^n] \otimes \mathbb{E}[\mu(z)|\mathcal{X}^n]^\top + \mathbb{C}ov[\mu(z'),\mu(z)|\mathcal{X}^n])] \quad (79)$$
$$= Tr[B(\Phi_V \boldsymbol{\beta}(z')\boldsymbol{\beta}(z)^\top \Phi_V^\top + \mathbb{C}ov[\mu(z'),\mu(z)|\mathcal{X}^n])] \quad (80)$$
$$= \boldsymbol{\beta}(z)^\top \Phi_V B \Phi_V^\top \boldsymbol{\beta}(z') + Tr[B\mathbb{C}ov[\mu(z'),\mu(z)|\mathcal{X}^n]] \quad (81)$$

where now $B = Ik_W(w,w') - \Phi_V^\top A(w,w')\Phi_V$, $A(w,w') = D(w)\left(K_W \odot K_V + I\sigma^2\right)^{-1} D(w')$ and $D(w) = \text{diag}(k_W(w,W_1),\ldots,k_W(w,W_N))$ as before. Note we can exchange the expectation with $Tr \circ B$ using the same reasoning as Theorem 4.1. Substituting in the definition of $B$ gives

$$\boldsymbol{\beta}(z)^\top \Phi_V B \Phi_V^\top \boldsymbol{\beta}(z') = \boldsymbol{\beta}(z)^\top \left(K_V k_W(w,w') - K_V A(w,w') K_V\right) \boldsymbol{\beta}(z') \tag{82}$$

$$Tr[B\mathbb{C}ov[\mu(z'),\mu(z)|\mathcal{X}^n]] = k_W(w,w') \sum_{i\in I} \lambda_i \hat{k}_Z(z,z') - Tr[A(w,w')\tilde{K}_V]\hat{k}_Z(z,z') \tag{83}$$

Re-arranging the terms we finally get $\mathbb{C}ov[\gamma(w,z),\gamma(w',z')|\mathcal{X}^n] = C_1 + C_2 + C_3$ where

$$C_1 = \boldsymbol{\beta}(z)^\top K_V (Ik_W(w,w') - A(w,w')K_V)\boldsymbol{\beta}(z') \tag{84}$$

$$C_2 = \hat{k}_Z(z,z')(Tr[\tilde{K}_V(\boldsymbol{\alpha}(w)\boldsymbol{\alpha}(w')^\top - A(w,w'))] \tag{85}$$

$$C_3 = \left(\sum\nolimits_{i\in I}\lambda_i\right) k_W(w,w')\hat{k}_Z(z,z')) \tag{86}$$

$\square$

### C.2.4. THEOREM 4.1 FOR AVERAGE TREATMENT EFFECTS

The following corollary shows how to use our derived posteriors for $\gamma$ to construct the posterior distribution on the ATE, which corresponds to marginal expectations of $\gamma$. Note, the following assumes we use the GP approximation of the posterior on the causal function $\hat{\gamma} \sim \mathcal{GP}(\mathbb{E}[\gamma(\bullet)\mid\mathcal{X}^n], \mathbb{C}ov[\gamma(\bullet),\gamma(\bullet)\mid\mathcal{X}^n])$.

**Corollary C.6** (Posteriors on average causal effects). *Under the approximated posterior distribution for the causal function, $\mathbb{P}_{\hat{\gamma}|\mathcal{X}^n} = \mathcal{GP}(\mathbb{E}[\gamma(\bullet)|\mathcal{X}^n], \mathbb{C}ov[\gamma(\bullet),\gamma(\bullet)|\mathcal{X}^n])$, if the absolute moments $\int |\mathbb{E}[\gamma(w,z)|\mathcal{X}^n]|\mathbb{P}_Z(dz)$ and $\int\int |\mathbb{C}ov[\gamma(w,z),\gamma(w,z')|\mathcal{X}^n]|\mathbb{P}_Z(dz)\mathbb{P}_Z(dz')$ are finite for all $z,z' \in \mathcal{Z}$, we have the following distribution of $\mathbb{E}[\hat{\gamma}(\bullet,Z)]$,*

$$\mathbb{E}[\hat{\gamma}(\bullet,Z)] \sim \mathcal{GP}(\hat{m},\hat{\kappa})$$

*where*

$$\hat{m}(z) = \int_{\mathcal{Z}} \mathbb{E}[\gamma(w,z)|\mathcal{X}^n]\mathbb{P}_Z(dz)$$

$$\hat{\kappa}(z,z') = \int_{\mathcal{Z}}\int_{\mathcal{Z}} \mathbb{E}[\mathbb{C}ov[\gamma(w,z),\gamma(w,z')|\mathcal{X}^n]\mathbb{P}_Z(dz)\mathbb{P}_Z(dz')$$

*Proof.* Follows immediately from Proposition 3.2 in Chau et al. (2021a). $\square$

*Remark* C.7. In practice, we compute the integrals in Corollary C.6 using averages respect to the empirical distribution $\hat{\mathbb{P}}_Z = \frac{1}{n}\sum_{i=1}^n \delta_{Z_i}$.
*Remark* C.8. The result holds analogously for $\mathbb{E}[\hat{\gamma}(W,Z)]$ and $\mathbb{E}[\gamma(W,\bullet)]$.

**Examples** The above result can be directly applied to obtain posteriors on population-level causal estimands such as the average treatment effect (ATE) in both back-door and front-door settings introduced earlier. For instance, in the back-door example of Figure 1 (left), the posterior on the conditional effect $\text{CATE}(a,c) = \gamma((a,c),c)$ derived from Theorem 4.1 can be averaged over the empirical distribution of $C$ to yield a posterior GP for the ATE,

$$\text{ATE}(a) = \int_{\mathcal{C}} \gamma((a,c),c)\,\mathbb{P}_C(dc).$$

Similarly, in the front-door example of Figure 1 (right), the posterior on $\text{ATT}(a,a') = \gamma(a',a)$ can be integrated over $\mathbb{P}_A$ to obtain the posterior on the corresponding ATE.

$$\text{ATE}(a) = \int_{\mathcal{A}} \gamma(a',a)\,\mathbb{P}_A(da').$$

Thus, Corollary C.6 provides a unified way to propagate uncertainty from posteriors on causal quantities of the form $\gamma(w,z)$ (often CATE, ATT) to averages of these causal quantities (such as the ATE).

C.2.5. CONSISTENCY OF CALIBRATION ESTIMATOR

Although bootstrap estimators are consistent under stability and smoothness conditions (Austern & Syrgkanis, 2020; Tang & Westling, 2024), the consistency of the estimator we use for the calibration error in Section 5 is unclear given the fact that we also use a non-parametric plug-in estimator to estimate $\hat{\gamma}$. However, below we verify that under appropriate smoothness conditions, this estimator is consistent whenever the empirical bootstrap estimator of $\mathbb{P}_{\mathcal{X}^n}$ is consistent. For simplicity, we prove the result for cumulative rather than highest-posterior-density regions, but the result extends under analogous conditions. In what follows, we denote by $\hat{\mathcal{X}}^n \mid \mathcal{X}^n$ the bootstrap resample of the observed dataset $\mathcal{X}^n = \{X_1, \ldots, X_n\}$ (note $X_i = \{Y_i, V_i, W_i, Z_i\}$), obtained by drawing $n$ samples with replacement from its rows. Formally, each $\hat{X}_i$ is drawn i.i.d. from the empirical distribution

$$\hat{\mathbb{P}}_{\mathcal{X}^n} = \frac{1}{n} \sum_{i=1}^{n} \delta_{X_i},$$

and the resulting joint law of the bootstrap sample is

$$\hat{\mathbb{P}}_{\hat{\mathcal{X}}^n \mid \mathcal{X}^n} = \hat{\mathbb{P}}_{\mathcal{X}^n}^{\otimes n}.$$

This distribution defines the conditional bootstrap measure used in the calibration estimator $\hat{\mathbb{P}}_{\hat{\mathcal{X}}^n \mid \mathcal{X}^n}(\hat{\gamma}(w, z) \in C_{w,z,\alpha,\mu}(\mathcal{X}^n))$.

**Theorem C.9.** *Under the conditions of Theorem 4.1, let $\{\mathcal{X}^n, \mathcal{X}^m\}$ be a partition of the dataset $\{(Y_i, V_i, W_i, Z_i)\}_{i=1}^{n+m} \sim \mathbb{P}_{Y,V,W,Z}^{\otimes n+m}$ for $n, m > 0$. Let $\mu_{w,z} = \mathbb{E}[\gamma(w, z) | \mathcal{X}^n]$, $\sigma_{w,z} = \mathbb{V}ar[\gamma(w, z) | \mathcal{X}^n]$, $F_\alpha(t) = \mathbb{P}_{\mathcal{X}^n}(\mu_{w,z} + t_\alpha \sigma_{w,z} \le t)$ and $t_\alpha$ be the $(1 - \alpha)$ quantile of $\mathcal{N}(0, 1)$. Let $\hat{\gamma}_m$ be the estimator for $\gamma$ in Singh et al. (2024) using $\mathcal{X}^m$. Then, under the assumptions of Theorem 6.1 in Singh et al. (2024) and $(L, \|\bullet\|_1)$-Lipschitzness of $F_\alpha$*

$$
\begin{aligned}
&\|\mathbb{P}_{\mathcal{X}^n}(\gamma(w, z) \le \mu_{w,z} + t_\alpha \sigma_{w,z}) \\
&- \hat{\mathbb{P}}_{\hat{\mathcal{X}}^n \mid \mathcal{X}^n}(\hat{\gamma}_m(w, z) \le \mu_{w,z} + t_\alpha \sigma_{w,z})\|_{L_1(\mathbb{P}_{\mathcal{X}^m})} \\
&\le \|D(\mathbb{P}_{\mathcal{X}^n}, \hat{\mathbb{P}}_{\hat{\mathcal{X}}^n \mid \mathcal{X}^n})\|_{L_1(\mathbb{P}_{\mathcal{X}^n})} + \mathcal{O}\left(m^{-\frac{1}{2}\frac{c-1}{c+\frac{1}{b}}}\right)
\end{aligned}
\tag{87}
$$

*where $D$ is the Kolmogorov metric, $c \in (1, 2]$, $b \in [1, \infty)$.*

*Proof.* To start note that by the triangle inequality we can split the deviation into $\mathcal{D}(\mathbb{P}_{\mathcal{X}^n}, \hat{\mathbb{P}}_{\hat{\mathcal{X}}^n \mid \mathcal{X}^n})$ and $|F_\alpha(\gamma(w, z)) - F_\alpha(\hat{\gamma}_m(w, z))|$.

$$|\mathbb{P}_{\mathcal{X}^n}(\gamma(w, z) \le \mu_{w,z} + t_\alpha \sigma_{w,z}) - \hat{\mathbb{P}}_{\hat{\mathcal{X}}^n \mid \mathcal{X}^n}(\hat{\gamma}_m(w, z) \le \mu_{w,z} + t_\alpha \sigma_{w,z})| \tag{88}$$

$$\le |\mathbb{P}_{\mathcal{X}^n}(\hat{\gamma}_m(w, z) \le \mu_{w,z} + t_\alpha \sigma_{w,z}) - \hat{\mathbb{P}}_{\hat{\mathcal{X}}^n \mid \mathcal{X}^n}(\hat{\gamma}_m(w, z) \le \mu_{w,z} + t_\alpha \sigma_{w,z})|$$

$$+ |\mathbb{P}_{\mathcal{X}^n}(\gamma(w, z) \le \mu_{w,z} + t_\alpha \sigma_{w,z}) - \mathbb{P}_{\mathcal{X}^n}(\hat{\gamma}_m(w, z) \le \mu_{w,z} + t_\alpha \sigma_{w,z})| \tag{89}$$

$$\le \sup_{t \in \mathbb{R}} |\mathbb{P}_{\mathcal{X}^n}(t \le \mu_{w,z} + t_\alpha \sigma_{w,z}) - \hat{\mathbb{P}}_{\hat{\mathcal{X}}^n \mid \mathcal{X}^n}(t \le \mu_{w,z} + t_\alpha \sigma_{w,z})|$$

$$+ |\mathbb{P}_{\mathcal{X}^n}(\gamma(w, z) \le \mu_{w,z} + t_\alpha \sigma_{w,z}) - \mathbb{P}_{\mathcal{X}^n}(\hat{\gamma}_m(w, z) \le \mu_{w,z} + t_\alpha \sigma_{w,z})| \tag{90}$$

$$= D(\mathbb{P}_{\mathcal{X}^n}, \hat{\mathbb{P}}_{\hat{\mathcal{X}}^n \mid \mathcal{X}^n}) + |\mathbb{P}_{\mathcal{X}^n}(\gamma(w, z) \le \mu_{w,z} + t_\alpha \sigma_{w,z}) - \mathbb{P}_{\mathcal{X}^n}(\hat{\gamma}_m(w, z) \le \mu_{w,z} + t_\alpha \sigma_{w,z})| \tag{91}$$

$$= D(\mathbb{P}_{\mathcal{X}^n}, \hat{\mathbb{P}}_{\hat{\mathcal{X}}^n \mid \mathcal{X}^n}) + |F_\alpha(\gamma(w, z)) - F_\alpha(\hat{\gamma}_m(w, z))| \tag{92}$$

Using the Lipschitz continuity of $F_\alpha$, we get

$$
\begin{aligned}
&|\mathbb{P}_{\mathcal{X}^n}(\gamma(w, z) \le \mu_{w,z} + t_\alpha \sigma_{w,z}) - \hat{\mathbb{P}}_{\hat{\mathcal{X}}^n \mid \mathcal{X}^n}(\hat{\gamma}_m(w, z) \le \mu_{w,z} + t_\alpha \sigma_{w,z})| \\
&\le D(\mathbb{P}_{\mathcal{X}^n}, \hat{\mathbb{P}}_{\hat{\mathcal{X}}^n \mid \mathcal{X}^n}) + L\hat{\xi}_m(w, z)
\end{aligned}
\tag{93}
$$

Where $\hat{\xi}_m(w, z) = |\gamma(w, z) - \hat{\gamma}_m(w, z)|$. Under the conditions of Theorem 6.1 in Singh et al. (2024), we have

$$\mathbb{P}\left(\hat{\xi}_m(w, z) \ge C \ln(4/\delta) m^{-\frac{1}{2}\frac{c-1}{c+1/b}}\right) \le 2\delta \tag{94}$$

where $(c, b) = \mathrm{argmin}_{(\tilde{c}, \tilde{b}) \in \{(b_1, c_1), (b_2, c_2)\}} \left( \frac{\tilde{c} - 1}{\tilde{c} + 1/\tilde{b}} \right)$ and $c_1, b_1, c_2, b_2$ are defined as in Singh et al. (2024). After-re-arranging, this gives:

$$\mathbb{P}(\hat{\xi}(w, z) \geq t) \leq 8 \exp(-\frac{t}{C} m^{\frac{1}{2} \frac{c-1}{c+1/b}}) \leq A \exp(-Btm^{\frac{1}{2} \frac{c-1}{c+1/b}}) \tag{95}$$

for $A, B > 0$ appropriately chosen. Now, taking expectations of Equation (93) we get

$$\|\mathbb{P}_{\mathcal{X}^n}(\gamma(w, z) \leq \mu_{w,z} + t_\alpha \sigma_{w,z}) - \hat{\mathbb{P}}_{\hat{\mathcal{X}}^n | \mathcal{X}^n}(\hat{\gamma}_m(w, z) \leq \mu_{w,z} + t_\alpha \sigma_{w,z})\|_{L_1(\mathbb{P}_{\mathcal{X}^m})} \tag{96}$$

$$\leq \|\mathcal{D}(\mathbb{P}_{\mathcal{X}^n}, \hat{\mathbb{P}}_{\hat{\mathcal{X}}^n | \mathcal{X}^n})\|_{L_1(\mathbb{P}_{\mathcal{X}^n})} + L\mathbb{E}\hat{\xi}_m(w, z) \tag{97}$$

Since $\mathbb{E}\hat{\xi}_m(w, z) = \int_0^\infty \mathbb{P}(\hat{\xi}_m(w, z) \geq t)dt$ we can write

$$\mathbb{E}\hat{\xi}_m(w, z) \leq A \int_0^\infty \exp(-Btm^{\frac{1}{2} \frac{c-1}{c+1/b}})dt \tag{98}$$

$$= \frac{A}{B} m^{-\frac{1}{2} \frac{c-1}{c+1/b}} \tag{99}$$

Which implies the result. $\qquad\square$

*Remark* C.10. The constants $c$ and $b$ describe the smoothness and effective dimensionality of the underlying regression problems Singh et al. (2024). Whilst sample splitting calibrates the posterior given only a subset of the dataset, we find sample splitting leads to better calibration performance (see Appendix B).

**Argmin consistency.** Theorem C.9 gives pointwise consistency of the calibration-loss estimator. In the grid-search implementation used in our experiments, the candidate set $\mathcal{B}$ is finite, so pointwise consistency over $\beta \in \mathcal{B}$ implies uniform consistency of $\hat{L}(\beta)$ over $\mathcal{B}$. Hence any minimizer $\hat{\beta} \in \arg\min_{\beta \in \mathcal{B}} \hat{L}(\beta)$ satisfies

$$L(\hat{\beta}) \to \min_{\beta \in \mathcal{B}} L(\beta),$$

in probability. For compact infinite-dimensional candidate families, the same conclusion follows under the usual stochastic-equicontinuity and continuity conditions for M-estimation.

### C.2.6. PROPERTIES OF NUCLEAR DOMINANT KERNELS

Here we present the proofs of Proposition A.10 and Proposition A.11 in Appendix A.6.1, which characterize the behaviour of the so-called 'nuclear-dominant' kernel construction used in previous work (Flaxman et al., 2016; Chau et al., 2021b) (and presented in (27)).

*Proof of Proposition A.10.* For nonnegative functions $a(t)$ and $b_{x'}(t)$, $\sup_{x'} \int a(t)b_{x'}(t)d\nu(t) \leq \int a(t) \sup_{x'} b_{x'}(t)d\nu(t)$ by monotonicity of integration. Applying this with $a(t) = k(x, t) \geq 0$ and $b_{x'}(t) = k(t, x')$ yields

$$\sup_{x'} r(x, x') \leq \int k(x, t) \sup_{x'} k(t, x') \, d\nu(t).$$

Since $k$ is stationary and bounded by 1, $\sup_{x'} k(t, x') = k(0) = 1$, giving $\sup_{x'} r(x, x') \leq \int k(x, t) \, d\nu(t)$. To show the limit, note that $\|x - t\| \to \infty$ for all fixed $t$ as $\|x - m\| \to \infty$, so $h(\|x - t\|) \to 0$. Because $0 \leq h(\|x - t\|) \leq 1$ and $\nu$ is finite, by dominated convergence $\int k(x, t) \, d\nu(t) \to 0$. This proves the result. $\qquad\square$

*Proof of Proposition A.11.* Given the form of $p$ and $k$, we have

$$r(x, x') = C_p \int_{\mathbb{R}^d} h(\|x - t\|)h(\|x' - t\|)h(\|m - t\|)dt$$

Because $h$ is nonincreasing and log-convex, it satisfies the product inequality

$$h(a)\,h(b) \leq h\left(\frac{a+b}{2}\right)^2 \leq h\left(\frac{a+b}{2}\right) \quad \text{for all } a, b \geq 0,$$

where the last inequality holds since $h(r) \leq 1$. By the triangle inequality, $\|x - m\| \leq \|x - t\| + \|t - m\|$, hence $\frac{1}{2}(\|x - t\| + \|t - m\|) \geq \frac{1}{2}\|x - m\|$, and by monotonicity of $h$,

$$h(\|x - t\|)\, h(\|t - m\|) \leq h\left(\tfrac{1}{2}\|x - m\|\right).$$

Therefore,

$$C_p \int h(\|x - t\|)h(\|x' - t\|)h(\|t - m\|)\, dt \; \leq \; h(\tfrac{1}{2}\|x - m\|) \int h(\|x' - t\|)\, dt \; = \; C_p C_p^{-1}\, h(\tfrac{1}{2}\|x - m\|),$$

where we note that $C_p = \int h(\|x - m\|) > 0$ by assumption, and is fixed w.r.t. $m \in \mathbb{R}^d$. Since the bound is independent of $x'$, this yields the claimed bound (29). $\qquad\square$

