# OpenReview forum: "Interventional Processes For Causal Uncertainty Quantification"
_ICML.cc/2026/Conference — ICML 2026 regular_

### Official Review · Reviewer_y9Es · 2026-02-22

**Soundness:** 2
**Presentation:** 3
**Significance:** 2
**Originality:** 3
**Overall Recommendation:** 4
**Confidence:** 3

**Summary:**

This paper proposes an improved Bayesian non-parametric method for estimation and uncertainty quantification of continuous treatment effects after causal identification. It first cast the estimation problem as kernel regression and then places linear GP prior on the RKHS functional and spectral GP priors on the RKHS elements. Authors conducted experiments on a toy example, a synthetic benchmark and a simulated healthcare example.

**Compliance With Llm Reviewing Policy:**

Affirmed.

**Final Justification:**

The authors addressed my concerns and therefore I increased my rating. I think it is overall a solid paper with the rebuttal additions.

**Key Questions For Authors:**

1. Can you be more specific with which practical challenge your method addresses (other than UQ itself) and design appropriate experiments to show that the UQ quality (with proper metrics) improves with your method compared to other baselines?

2. Can you include an actual real-world downstream application with a real-world relevant metric to show-case the effectiveness of the method in practice?

**Limitations:**

The authors did not discuss limitations. Please see weakness part for my suggested weaknesses.

**Strengths And Weaknesses:**

Strength:

1. The presentation is clear and the methodology is technically sound
2. The analysis of and comparison against BayesIMP is insightful

Weakness:

My major concern with the paper is that its experiments are too weak and superficial to support the claimed contributions.

1. The authors themselves also pointed out that casting the estimation problem as kernel ridge regression, using GP for uncertainty quantification, as well spectral representation of RKHS elements, are all existing ideas. Therefore, the major contribution of this paper in my opinion is the improvement over existing methods, which, if supported well by experimental results, would be significant and meaningful. However, the authors only made good comparisons against BayesIMP and the comparison against other worth-noting baselines(bayesian additive regression trees, continuous DR, generalized/orthogonal random forests, VCnet etc) are either missing or superficial. For example, continuous DR is missing from the toy example and the healthcare application.

2. Regarding the metric used, while the main focus of the paper is UQ, authors only reported calibration error on the two synthetic benchmarks, which is not sufficient for evaluating UQ quality. A model that constantly predicts large uncertainty will have 0 calibration error while being useless. I suggest the authors consider better metrics such as interval scores, or simply likelihoods when ground truth is known.

3. The experiments settings are too simplified and "toy-ish" and do not address practical causal inference challenges. Since the authors framed the paper as contributing to causal inference rather than "a novel bayesian non-parametric method for estimation/UQ," I would expect them to address practical challenges that arise in the estimation step (after causal identification) of observational causal inference, such as limited support coverage, heavy-tailed/skewed treatment distributions, non-smooth non-linearity, robustness against noise/misspecification etc, which can all affect both estimation quality and UQ quality. However, the current experiment set-up lacks these stress tests. Even the "application to healthcare" example is in fact only a simulated dataset based on linear models. This leaves it unclear how the proposed method works on complex real-world data. In my opinion, to demonstrate that IMPspec is a meaningful contribution to causal inference research, synthetic experiments specifically designed to show superiority of IMPspec over existing methods under challenging/adversarial settings and real-world data experiments that show-case the practical effectiveness are needed.

---

> ### Author Rebuttal · Authors · 2026-03-31
>
> Thank you for reviewing our work - your feedback has certainly strengthened the paper. Below we address your points.
>
> **(I) Request for other UQ Metrics:** Thank you for suggesting other UQ metrics. We note our definition of calibration error (Eq. 22) estimates average absolute coverage error across a grid of coverage levels and inputs. Thus, intervals that are systematically too wide are penalized through over-coverage, just as intervals that are too narrow are penalized through under-coverage.
>
> However, we agree reporting interval scores (IS) would strengthen the paper, as they jointly assess coverage and tightness. Below we report IS for all methods in Section 7.1 and 7.2 at nominal 90\% and 95\% levels. The IS at level $1-\alpha$ is computed as:
>
> $$\mathrm{IS}^{1-\alpha}= \frac{1}{|\mathcal{W}||\mathcal{Z}|}\sum_{w\in\mathcal{W}}\sum_{z\in\mathcal{Z}}\Big[(u_{w,z}-l_{w,z})+\frac2\alpha(l_{w,z}-\gamma(w,z))1(\gamma(w,z)<l_{w,z})+\frac2\alpha(\gamma(w,z)-u_{w,z})1(\gamma(w,z)>u_{w,z})\Big]$$
>
> with $[l_{w,z}, u_{w,z}]$ the  $(1-\alpha)$-prediction interval for $\gamma(w,z)$, and $\mathcal W \times \mathcal Z$ the test grid.
>
> **Section 7.1 Interval Scores**
> ||IS 90%|IS 95%|
> |---|---:|---:|
> |BayesIMP|1.80±0.18|2.50±0.36|
> |Sampling GP|1.59±0.09|2.20±0.16|
> |IMPspec-no-cal|1.10±0.05|1.36±0.07|
> |IMPspec-cal|**0.99±0.03**|**1.16±0.05**|
>
> **Section 7.2 Interval Scores (in-support)**
> ||IS 90%|IS 95%|
> |---|---:|---:|
> |Sampling GP|5.29±4.57|6.67±5.90|
> |BayesIMP|2.18±1.35|2.99±2.38|
> |IMPspec|**1.84±1.25**|**2.27±2.03**|
> |Continuous DR|6.32±10.57|8.37±13.53|
>
>
>
> **(II) Request for Additional Baselines:**
> 1. **ORF Baseline Added**: Thank you for the baseline suggestions. While BART is not for continuous treatments and VCNet does not cover uncertainty quantification, we have now implemented Orthogonal Random Forests (ORF) in the Benchmark Simulations in Sec 7.1, and report performance below (in-support) alongside our method. ORF performs significantly worse, which likely reflects its linear-in-treatment-effect assumption.
>
> ||RMSE|Cal.error|IS 90%|IS 95\%|
> |-|-:|-:|-:|-:|
> |ORF|0.84±0.16|0.16±0.07|6.58±0.76|10.34±0.95|
> |IMPspec|0.36±0.16|0.04±0.02|1.84±1.25|2.27±2.03|
>
>
> 2. **Clarifying Existing Baselines**.
> - Continuous DR and other methods you suggested are designed under the *back-door* criterion. Thus, they are not applicable in Sec 7.1 and 7.3, as those experiments mostly operate under settings analogous to the *front-door* criterion.
> - The baselines chosen for the BO experiment are were designed specifically for causal BO (i.e. CBO, BayesIMP).
> - The first experiment aims to compare against related GP-based methods - with a particular focus on the benefit over BayesIMP.
>
> **(III) Request for Additional Experiments**: We have added two ablations you suggested, using the design in Sec 7.1.
> 1. **Noise Mis-specification**: We re-ran all methods under heavy-tailed Student-T ($\nu = 3$) noise on $Y$ and $X$ for 50 trials. Below we report RMSE, cal-error and 90-95\% interval scores (IS). Our method still performs best, and uncertainty scores are only slightly worse than under Gaussian noise (see above and Table 2 main text).
>
> ||RMSE|Cal.Error|IS 90\%|IS 95\%|
> |---|---:|---:|---:|---:|
> |Sampling-GP|0.37±0.01|0.16±0.09|2.41±0.20|3.73±0.42|
> |BayesIMP|0.58±0.01|0.35±0.17|6.43±0.34|11.41±0.69|
> |IMPspec-nocal|0.35±0.02|0.15±0.09|2.31±0.26|3.47±0.50|
> |IMPspec-cal|**0.35±0.02**|**0.10±0.0**|**2.02±0.22**|**2.84±0.41**|
>
> 2. **Limited Support Coverage**: We re-ran all methods (50 trials) under a heavily skewed treatment distribution $Z \sim Beta(3,1/2)$, which concentrates its mass near $z=1$, making it challenging to extrapolate to the full range $[0,1]$. Our method's uncertainty estimates still scored best, but our RMSE is now only second best.
>
> ||RMSE|Cal.Error|IS 90\%|IS 95\%|
> |---|---:|---:|---:|---:|
> |Sampling-GP|**0.32±0.01**|0.10±0.05|1.40±0.06|1.74±0.10|
> |BayesIMP|0.45±0.02|0.19±0.12|2.92±0.31|4.47±0.61|
> |IMPspec-nocal|0.35±0.01|0.08±0.05|1.87±0.12|2.61±0.21|
> |IMPspec-cal|0.36±0.01|**0.06±0.04**|**1.43±0.07**|**1.70±0.10**|
>
> 3. **Real Dataset**: To demonstrate our method's applicability and scalability, we have now implemented it on a large-scale ($n \approx 10^4$) real dataset. Runtime + memory performance + learned causal effects can be found in point (III) of the response to reviewer xLo4.
>
>
> **(IV) Why we focus on Simulations**
> 1. We note causal effects are impossible to validate on real data without performing new experiments/interventions. Thus, almost all causal benchmark datasets in the literature are based on synthetic generation, where ground truth is known.
> 2. The simulation in Sec 7.2 and the healthcare application in Sec 7.3 are known benchmarks in the literature, and were originally used in previous work we compare against (e.g., see BayesIMP and CBO).
>
>
> If our clarifications and additional experiments have addressed your concerns, we would be grateful if you might consider updating your score.

---

> > ### Author Rebuttal · Reviewer_y9Es · 2026-04-01
> >
> > I thank the authors for including the additional experiments. They have addressed my concerns and I will increase my score to 4.

---

### Official Review · Reviewer_xLo4 · 2026-03-11

**Soundness:** 3
**Presentation:** 3
**Significance:** 4
**Originality:** 3
**Overall Recommendation:** 5
**Confidence:** 3

**Summary:**

This paper introduces a nonparametric prior, called the IMPspec process, for uncertainty quantification of causal estimands such as ATE, ATT, and CATE. Building on the inner-product representation of causal estimands and kernel-based estimators from Singh et al. (2024), the authors presume a surrogate observation model under which posterior moments become analytically tractable. Since the resulting posterior is not available in closed form, they approximate it by a Gaussian process with matching mean and covariance, and use this approximation to construct credible intervals. The paper also proposes procedures for hyperparameter tuning and calibration using held-out data. Empirically, the method is evaluated on synthetic datasets and GP-based surrogate experiments.

**Compliance With Llm Reviewing Policy:**

Affirmed.

**Final Justification:**

The authors adequately addressed my concerns and questions. I am raising my score to 5.

**Key Questions For Authors:**

1. Could the authors provide more details on the choice of the kernels (around line 1850)? Since the method does not appear to depend on a specific kernel family, it would be helpful to understand why Gaussian kernels were used in the experiments.
2. If Gaussian kernels are used, is the method susceptible to the same length-scale degeneracy issues that the authors discuss for radial kernels in Section 6 when comparing against Chau et al. (2021b)?
3. Related to Weakness 4, could the authors include an application to a real-world causal dataset without relying on a GP surrogate model?

**Limitations:**

The paper overall appears to be a solid methodological contribution to a problem relatively underexplored in the literature. Some of the practical limitations, such as cubic time scaling, length degeneracy and kernel hyperparameter sensitivities are standard limitations of most of the kernel based methodologies rather than shortcomings specific to this method. That being said, the paper could benefit from stronger justification of the proposed surrogate model and from empirical evaluation on more complex real world data.

**Strengths And Weaknesses:**

Strengths:
1. The proposed UQ procedure is designed to complement an existing point estimator with desirable theoretical properties, and is grounded in a closely related and coherent theoretical framework.
2. The posterior moments are analytically tractable under the proposed surrogate model.
3. The paper addresses a relatively underexplored problem, namely constructing interval estimators for causal estimands, and therefore its contribution is meaningful to the literature.

Weaknesses:
1. The proposed surrogate models in Eqs. 10 and 11 are lightly justified beyond their role in making posterior moments analytically tractable. While this is understandable, the paper would benefit from additional discussion as to its interpretation in real data applications.
2. The method is analytically tractable but computationally demanding. The cubic time cost in $n$ required to invert $n\times n$ matrices in Eqs 17-19 poses a clear limitation for scalability.
3. The overall procedure involves multiple stages of hyperparameter tunings and calibration, including bootstrap sampling. While this might be manageable in controlled simulated settings, its robustness and practicality in real data applications are less clear.
4. The numerical evaluation is limited to synthetic and simulated examples. Instead of GP surrogate model, a real world causal dataset would strengthen the empirical case for the method and help assess its robustness in practice.

---

> ### Author Rebuttal · Authors · 2026-03-30
>
> Thank you for your detailed review - we are glad to hear that you think our work provides a solid methodological solution to an important and underexplored problem. Below we address the main weaknesses you raised and queries.
>
>
> **(I) On Surrogate Models** We agree that the surrogate models eq(10-11) could be better explained in the main text. Below we expand on this.
> -  The role of these models is indeed to ensure tractability of the posterior moments. However, these particular models are also *necessary* to ensure the posterior mean of our estimator agrees exactly with the kernel estimator in Singh et al (2024) - which we aim to characterize uncertainty around. This is a consequence of known connections between GPs and RKHS's [4].
> - In real applications, eq(10) assumes an additive Gaussian noise model on outcomes $Y$. eq(11) is more difficult to interpret in full generality, but when the kernel $k_V$ is linear, is also equivalent to an additive Gaussian noise model $V = \mu(Z) + \xi$.
> - Our calibration procedure is designed to offset error introduced from surrogate mis-specification, and in experiments our method achieved best performance, even when the surrogates were mis-specified (e.g., eq(11) in Sec 7.1).
>
>
> **(II) On Computational Demands.** While kernel training can be easily scaled using minibatching, we agree the $\mathcal O(n^3)$ matrix inversion costs of our method at inference time can limit scalability. However, we have tested our method on datasets up to $n = 10^4$ (see below), and found end-to-end implementation on a single H100 GPU took $<40$secs and used $<16$ GB memory. For much larger datasets (i.e., $n\geq 10^5$) the kernel matrices at inference time can be approximated by using standard Nystrom or Random Feature approximations, which reduces time and storage complexity to $\mathcal O(n)$.
>
> **(III) On Request for Real World Dataset.** To demonstrate the scalability and practicality of our method, we have now implemented it on the well known 401k dataset [1], where the aim is to quantify causal effects of 401k pension plan eligibility $D \in \{0,1\}$ on asset accumulation (i.e., $Y =$ Net Financial Assets). The dataset contains $n\approx 10^4$ samples and $d = 9$ covariates $W$ (see [1] for details). We use our method to estimate the conditional average treatment effect curve:
>
> $$\text{CATE}(x) = \mathbb E[Y|do(D=1),X=x] - \mathbb E[do(D=0),X=x]$$
>
> and its uncertainty, where $X = \text{income}$. Thus, we apply our method as defined in the paper with the variable definitions $(W,V,Z) := (W,D,X)$ (LHS = variables in paper).
>
> Below we summari\e the runtime and memory footprint when running our method on a single NVIDIA H100, with minibatched gradients ($b=512$ batch sizes) for scalable training. The learned CATE and uncertainty bands from our method can be found [HERE](https://ibb.co/9HzcKX9d).
>
> |Stage|Batch Size|Sec|GPU Alloc (GB)|
> |-|-:|-:|-:|
> |Training|512|7.1|4.0|
> |Calibration|9915|29.9|14.2|
> |Inference|9915|1.86|14.8|
>
> Overall, we find the CATE is generally increasing in income levels, which agrees with previous work [3]. Moreover, the estimated CATE and its uncertainty bands by our method are similar to that of ORF (although our estimator is smoother due to the GP function class), which is a well-established frequentist method for causal uncertainty quantification.
>
> **(IV) Q1 Kernel Choice.** We used Gaussian kernels in experiments as they are the standard option in the literature, but any differentiable kernel can be used. To demonstrate, below we ran our method on the experiment in Section 7.1 using a $\gamma$-exponential kernel $k(x,x') = \tau\exp^{-\theta\|x-x'\|^\gamma}$ with tunable $\gamma$ (which generalizes the Gaussian and Exponential kernels), showing similar performance.
>
> |Method|Kernel|RMSE|Cal. Error|
> |---|---|---:|---:|
> |IMPspec-nocal|Gaussian|0.223 ± 0.046|0.098 ± 0.010|
> |IMPspec-cal|Gaussian|0.223 ± 0.046|0.065 ± 0.008|
> |IMPspec-nocal|Gamma-Exp|0.236 ± 0.006|0.088 ± 0.010|
> |IMPspec-cal|Gamma-Exp|0.238 ± 0.006|0.067 ± 0.008|
>
> **(V) Q2 Lengthscale Degeneracy:** We are not certain if the same lengthscale-degeneracy in BayesImp applies to our loss. However, we fix the parameters of $k_V$ when training $k_Z$ to maximize the weighted log-likelihood in eq(XX) regardless. This is because the kernel $k_V$ is learned to optimize the observational GP model
> $$Y = f(W,V)+\epsilon$$
> If we were to again optimize $k_V$ via the objective for training $k_Z$, this would reduce the quality of the observation model - and may also lead to the kernels "chasing eachother" (as in BayesIMP).
>
> If you find our clarifications and additional experiments satisfactory, we would be grateful if you might consider updating your score.
>
> [1] https://docs.doubleml.org/stable/examples/py_double_ml_pension.html
>
> [2] https://proceedings.mlr.press/v97/oprescu19a.html
>
> [3] https://www.mit.edu/~vchern/papers/ch_401k.pdf
>
> [4] https://arxiv.org/pdf/2506.17366

---

> > ### Author Rebuttal · Reviewer_xLo4 · 2026-04-02
> >
> > Thank you for addressing my concerns, espeically the computational budget and real date demonstration. On (V), could the authors elaborate on the answer, on what differences do the proposed method from BayesImp have that makes it less susceptible to lengthscale degeneracy (it seems like the main difference is in training procedure/objective?)? I did not quite follow the explanation about eq(XX). Could you explain more about what does "chasing each other" mean? Thank you.

---

> > > ### Author Response · Authors · 2026-04-04
> > >
> > > Thank you for your thoughtful follow-up question! We agree that we could have explained what we meant more clearly on lengthscale degeneracy. We were a bit short on characters, but now we can provide a more detailed answer.
> > >
> > > **General idea.** A useful way to think about our method is that (spectral representation aside) it is essentially learning probabilistic maps between the following feature spaces:
> > > $$\phi_Z(Z)\mapsto \phi_V(V), \tag{a}$$
> > >
> > > $$\phi_V(V)\otimes \phi_W(W)\mapsto Y. \tag{b}$$
> > >
> > > From this perspective, kernel hyperparameter tuning corresponds to learning the feature maps $\phi_V,\phi_W,\phi_Z$ in a way that maximizes the accuracy/likelihood of these mappings (recall that choosing the kernel is equivalent to choosing the feature map, via $k_V(v,\cdot) = \phi_V(v)$).
> > >
> > > In particular, the marginal likelihood in Eq. (20) represents (b) above and is used to learn feature maps $\phi_V,\phi_W$ so as to explain $Y$, while the weighted log-likelihood (WLL) in Eq. (21) represents (a) above and is used to learn feature map $\phi_Z$ so as to explain $\phi_V(V)$.
> > >
> > > The high-level concern is that, if one instead tries to learn both feature maps $\phi_V$ and $\phi_Z$ to maximise the accuracy/likelihood of the mapping $\phi_Z(Z)\mapsto \phi_V(V)$, the two feature maps may collapse towards each other in a way that leads to trivial or degenerate solutions. For example, one could in principle set:
> > > $$\phi_V(V) = \phi_Z(Z) = \text{constant}$$
> > > in which case the mapping $\phi_Z(Z)\mapsto \phi_V(V)$ can be solved perfectly, but such features would encode no information about $V$ and $Z$. This is what we meant before by kernels *"chasing each other"*.
> > >
> > > This concern was originally discussed in [1] in the context of vector-valued kernel regression, and subsequently referenced by BayesIMP.
> > >
> > > **Does our WLL Loss Converge to Such Degenerate Solutions?**
> > >
> > > We do not have a general result on whether our particular WLL loss Eq(21) actually converges to trivial solutions if jointly learning the lengthscales of the kernels on $Z,V$ (i.e., $\phi_Z,\phi_V$). To our knowledge, there is also no general result in the BayesIMP paper on whether their loss has such degeneracies, but the authors appeared to take the conservative route to avoid any problems.
> > >
> > > We therefore do not claim that such kernel degeneracies are impossible using our loss (or are less likely than using BayesIMP's loss), but rather that they can be avoided using a sensible training procedure. In our method, the middle representation $\phi_V$ is not learned through the weighted log-likelihood (WLL) in Eq. (21) - which is constructed from the regression $\phi_Z(Z)\mapsto \phi_V(V)$. Instead, $\phi_V$ is learned from the observational prediction problem for $Y$ via Eq. (20), and is then held fixed when optimizing Eq. (21) over the $Z$-side parameters. That is, we do not jointly learn $\phi_V$ and $\phi_Z$ via the WLL objective.
> > >
> > > Thank you again for your thorough review of our paper! If we have been able to resolve the last of your queries, we would be very grateful if you could reflect this in your score.
> > >
> > > [1] JF. Ton et al. (2020) Noise Contrastive Meta-Learning for Conditional Density Estimation using Kernel Mean Embeddings. AISTATS21.

---

### Official Review · Reviewer_J2uh · 2026-03-13

**Soundness:** 3
**Presentation:** 3
**Significance:** 2
**Originality:** 2
**Overall Recommendation:** 4
**Confidence:** 4

**Summary:**

The paper studies uncertainty quantification for causal effects under continuous treatments in a nonparametric setting. It begins from the observation that estimands such as ATE, CATE, and ATT can often be expressed through a common interventional functional,
$$
\gamma(w,z)=\int \mathbb{E}[Y \mid W=w, V=v]\, P_{V \mid Z}(dv \mid z),
$$
and represented using RKHS tools. Building on this, the authors propose **IMPSPEC** (spectral interventional mean process), a Bayesian method that places GP-style priors on the components needed to estimate this functional, while avoiding the difficulty of defining Gaussian process priors directly over RKHS-constrained objects.

Methodologically, the paper derives closed-form expressions for the posterior mean and variance of the causal functional and then uses moment matching to construct a GP approximation for credible intervals. It also introduces a closed-form hyperparameter training objective and a spectral calibration procedure to improve posterior coverage. Empirically, IMPSPEC is evaluated on a toy problem, a synthetic benchmark for continuous-treatment causal estimation, and a causal Bayesian optimization setting with a healthcare application. The reported results suggest improved calibration and competitive, and often superior, causal estimation and calibration performance to BayesIMP, a sampling-based GP baseline, continuous doubly robust estimation, and standard causal BO baselines.

**Compliance With Llm Reviewing Policy:**

Affirmed.

**Final Justification:**

The rebuttal addresses my main concerns. The authors provided clean answers. So I raise my score.

**Key Questions For Authors:**

1. Could the authors clarify which parts of the uncertainty pipeline are exact under the stated model, and which are approximate?

2. How sensitive is the calibration procedure to the choice of spectral family, bootstrap settings, and sample splitting?

3. Can the authors provide stronger evidence of practical scalability? Runtime or memory measurements, or experiments at larger $n$, would strengthen the paper’s practical significance.

4. How far should the empirical findings be expected to generalize beyond the current settings, i.e., in a stronger real or semi-synthetic setup?

5. On the experimental side, and since the paper positions the RKHS approach as avoiding density estimation and propensity weighting, could the authors clarify whether continuous-treatment baselines based on those ideas were considered, and if not, why they are not appropriate comparators here?

**Limitations:**

The paper does not include a dedicated discussion of limitations or broader societal impacts, despite motivating high-stakes applications such as healthcare. The authors should add a short paragraph discussing both methodological limitations (e.g., approximate inference, calibration dependence, scalability) and potential risks of deployment in such settings.

**Strengths And Weaknesses:**

## Strengths

*  The paper studies uncertainty quantification for causal effects under continuous treatments in a nonparametric setting. This is both practically relevant and technically challenging, especially when uncertainty estimates are needed not only for effect estimation but also for downstream decision-making.

* A notable strength is that the posterior mean is constructed to recover the kernel estimator of Singh et al., so the method provides uncertainty quantification around an already-motivated estimator rather than around a separate surrogate predictor.

* The experiments, while limited in scope, are directionally strong: IMPSPEC reports the best RMSE/calibration tradeoff on the toy and synthetic benchmarks and the best cumulative regret in the BO tasks considered.

**Weaknesses**

* The uncertainty story is more approximate than the framing suggests: Because the exact posterior for $\gamma$ is intractable, the method relies on surrogate observation models and a moment-matched GP approximation for credible intervals. This is reasonable, but it is better viewed as approximate Bayesian uncertainty quantification than as a direct posterior characterization of the causal functional.

* Calibration adds an extra empirical layer: The calibration procedure tunes the spectral measure through a bootstrap-based coverage objective, and its consistency relies on additional smoothness and sample-splitting assumptions. As a result, the strongest uncertainty guarantees depend on more than the base model alone.

* Scalability is not yet convincing: The method inherits the usual $O(n^3)$ GP cost, and the paper only briefly mentions minibatched stochastic optimization as a possible remedy. Given the relatively small-scale experiments, practical scalability remains unestablished.

* Some conceptual distinctions need clearer explanation: In particular, the difference between the true posterior over $\gamma$ and the moment-matched GP approximation, as well as the role of the surrogate models in equations (10)–(11), should be explained more plainly.

* The empirical scope is still limited: The synthetic benchmark uses only $n=100$ samples, and the healthcare application remains simulator-based, albeit grounded in real data and a known causal graph. The results are encouraging, but they do not yet demonstrate robustness in more realistic observational settings.

---

> ### Author Rebuttal · Authors · 2026-03-30
>
> Thank you for your thorough and detailed review. We are glad you recognize the challenges of the continuous treatment setting, and some key benefits and performance of our method. Your comments have certainly helped us strengthen the paper. Below we address your main points:
>
> **(I) Distinguishing Method Components + Approximations** We agree more clarity on the role of different modeling assumptions in the pipeline would be helpful. We will make the below clearer to the revised version:
>
> - The surrogate observation models eq(10-11) are designed to induce closed-form posterior moments (Theorem 4.1) with a posterior mean that agrees with the estimator in Singh et al (2024).
> - Thus, the posterior mean and variance (in Theorem 4.1) are exact under the stated model.
> - The credible intervals use a GP-posterior approximation centered at the posterior moments.
> - The true posterior over $\gamma$ is essentially a weighted infinite sum of product-of-normals, so is likely heavier tailed than the Gaussian approximation (Fig 2 shows prior samples).
>
> **(II) Requested Ablations of Calibration Procedure**
>
> - **Sample Splitting**: We explore the effect of sample splitting on calibration performance in Table 4 in Appendix C.1, finding that sample splitting improved performance. This is consistent with our calibration consistency result Thm B.9 in Appendix B.2.5, which relies on sample splitting.
> - **Bootstrap Settings**: We re-ran our method on the experiment in Sec 7.1 varying with varying \# bootstrap iterations used in calibration (50 trials on a NVIDIA Quadro P5000 / RTX A4500 GPU). Below we find performance and runtimes are relatively unaffected by this parameter. This may be due to the fact that the uncalibrated model was overconfident in this experiment (see Fig 3, right) so even a few bootstrap replications give a strong enough signal to improve calibration.
>
> |Boot-reps $\to$|B=10|B=20|B=50|B=100|
> |-|-:|-:|-:|-:|
> |RMSE| 0.23 ± 0.01 |0.23 ± 0.01|0.23 ± 0.01|0.23 ± 0.01|
> |Cal error |0.07 ± 0.01|0.08 ± 0.01|0.08 ± 0.01|0.07 ± 0.01|
> |Wallclock time|19.11 ± 1.72|16.10 ± 1.27|20.10 ± 1.94|19.03 ± 1.65|
>
> - **Spectral Family:** We re-ran our method on the experiments in Sec 7.1 using a Laplace distribution $\nu$ with tunable scale parameter $\lambda$, instead of a Gaussian distribution. The results remain similar to the above in this case) below uses $B=20$. More generally, any family with a variance hyperparameter that can be tuned to affect posterior variance will enable the calibration procedure to target reasonable coverage levels.
>
> ||RMSE|Cal Error|
> |-|-:|-:|
> |$\nu = \mathrm{Lap}(\mu_V, \sigma^2)$ |0.23 ± 0.01|0.09 ± 0.01|
>
> **(III) Evidence of Scalability**: To strengthen evidence of the scalability of our method, we have implemented it on a **large-scale real dataset ($n \approx 10^4$)**, where we estimate how the average effect of 401k pension plan eligibility on Net Financial Assets varies with income levels.
>
> Details and results for this problem can be found in point (III) of the response to **reviewer xLo4**. The method ran in $<40$secs on a single GPU, demonstrating feasibility in large-scale settings.
>
> **(IV) Clarifying Density/Weighting Baselines**
>
> - In the Synthetic Benchmarks experiments in Sec 7.2, we compared against the frequentist doubly-robust method for continuous treatments by Kennedy et al. (2017) (cited in the paper). This method precisely requires inverse propensity weighting and density estimation approaches.
> - The Sampling-GP approach of Witty et al. (2020), which we compared against in both the toy example (Sec 7.1) and synthetic benchmark (Sec 7.2) also uses explicit (GP-based) conditional density estimation.
>
> **(V) Requested Limitations/Impact**
>
> - We propose the following limitations paragraph:
>
> *"Two key limitations of our method are (i) it relies on Gaussian noise assumptions for closed form posterior moments, and (ii) it uses a Gaussian posterior approximation for tractable intervals. In practice, both modeling assumptions and posterior approximations may introduce error into the uncertainty estimates. Our calibration procedure is designed to mitigate this, by explicitly adapting the posterior to target good frequentist coverage rates. However, we cannot guarantee to achieve perfect calibration even asymptotically, as the calibration parameter may not have a large enough effect on posterior variance to perfectly match the true asymptotic distribution, and the asymptotic distribution may be non-Gaussian."*
>
> - We propose the following impact paragraph:
>
> *"The method developed in this work may be useful in helping practitioners to assess the risk and uncertainty of policies or treatments. It may also be useful in intervention search problems, by enabling researchers to discover good treatment dosages faster, in high-stakes applications like healthcare."*
>
> If you find our clarifications and additional experiments satisfactory, we would be grateful if you might consider updating your score.

---

> > ### Author Rebuttal · Reviewer_J2uh · 2026-04-04
> >
> > The authors addressed my main concerns in a clean, straightforward way. I have raised my score by one point.

---

### Official Review · Reviewer_NN4N · 2026-03-13

**Soundness:** 2
**Presentation:** 3
**Significance:** 3
**Originality:** 3
**Overall Recommendation:** 4
**Confidence:** 3

**Summary:**

This paper proposes IMPSPEC, a Gaussian process–based framework for uncertainty quantification of causal effects under continuous treatments. The key idea is to represent the conditional expectation function as a linear functional over a tensor-product RKHS and place a Gaussian process prior on this functional rather than on the RKHS function itself. This formulation relaxes standard boundedness constraints while enabling closed-form posterior inference and recovering the kernel ridge estimator of Singh et al. (2024) as the posterior mean.
To make the model computationally tractable, the authors use a spectral decomposition of the RKHS-valued feature map based on Mercer's theorem, which converts the problem into Gaussian processes over scalar-valued coordinates. This leads to closed-form posterior moments for the causal effect and enables practical uncertainty quantification via a moment-matching GP approximation. The paper further introduces tractable hyperparameter learning objectives and a bootstrap-based calibration procedure.
Experiments on synthetic datasets, toy examples, and a healthcare simulator demonstrate improved calibration and predictive accuracy compared with existing approaches such as BayesIMP, with additional downstream improvements in causal Bayesian optimization tasks.

**Compliance With Llm Reviewing Policy:**

Affirmed.

**Final Justification:**

An interesting work needs some revision for clarification.

**Key Questions For Authors:**

1.  Regarding the surrogate model: Equation (11) models $\phi_{V,i}(V) = \mu_i(Z) + \xi_i$ with independent $\xi_i$. Could you please elaborate on the justification for this modeling choice? More concretely, how might this "likelihood approximation" affect the accuracy of the posterior variance and, consequently, the calibration of the final credible intervals? Do you have any theoretical or simulation-based analysis that quantifies the bias introduced by this assumption?
2.  Regarding non-negative variance: The posterior variance in Theorem 4.1 relies on a Monte Carlo approximation of $\tilde{K}_V$. Can you provide theoretical conditions or discuss practical numerical strategies to ensure that the estimated $S_1+S_2+S_3$ is non-negative? If negative values do occur in practice, how do you suggest handling them, and what would be the impact on downstream tasks like CBO?
3.  Regarding theoretical guarantees: The paper currently lacks a theoretical analysis of the frequentist coverage properties of the proposed intervals. Can you comment on whether, under certain conditions (e.g., correct model specification, regularity conditions), the IMPSPEC posterior intervals might be expected to achieve asymptotically correct coverage? Or do you view the method strictly as a Bayesian procedure, with the calibration algorithm serving as a practical, albeit heuristic, bridge to good frequentist performance?
4.  Regarding computational scalability: The paper mentions stochastic gradients as a potential path to scaling, but provides no evidence. Could you report the runtime and memory footprint for the dataset sizes used in the experiments (n ~ 100-1000)? For a dataset with n ~ 10^5, which type of sparse approximation (e.g., inducing points) do you think would be most suitable for IMPSPEC, and what might be the potential impact on calibration?

**Limitations:**

yes

**Strengths And Weaknesses:**

## Strengths

- The paper studies an important problem: uncertainty quantification for causal effects under continuous treatments, which is highly relevant in high-stakes decision-making settings.

- The methodology is technically interesting and well-developed. In particular, the use of a GP prior on the linear functional, together with the spectral decomposition of the RKHS-valued feature map, provides a novel and elegant way to obtain tractable posterior inference while recovering the kernel ridge estimator of Singh et al. (2024) as the posterior mean.

- The paper is generally well written and clearly structured. The critique of prior GP-on-RKHS approaches is insightful, and the empirical results on synthetic tasks and downstream causal Bayesian optimization suggest meaningful practical benefits.


## Weaknesses

- A central concern is the surrogate observation model in Eq. (11), which treats each spectral coordinate $\phi_{V,i}(V)$ as a noisy observation of $\mu_i(Z)$ with independent noise. This appears to be a strong modeling simplification, and the paper does not clearly justify its adequacy or analyze the approximation error it may introduce.

- The posterior variance depends on a Monte Carlo approximation of $\tilde{K}_V$, but the paper does not discuss whether this approximation can lead to numerical instability (e.g., negative variance estimates in finite precision) or how such issues would be handled in practice.

- While the paper proposes a calibration procedure for credible intervals, it does not provide a clear theoretical account of whether the resulting intervals should be interpreted purely as Bayesian credible intervals or as approximations with reliable frequentist coverage guarantees.

---

> ### Author Rebuttal · Authors · 2026-03-30
>
> Thank you for providing a detailed review - we are glad you found our method novel and elegant, and the paper well written/structured. Your comments have helped us strengthen the manuscript. Below we address your points:
>
> **(I) Surrogate Observation Model eq(11)**
>
> - **Justification:**  Eq. (11) should be interpreted as a surrogate likelihood used for closed form posterior moments - it is the same  approximation used in BayesIMP. However, in our case it is precisely the model that ensures the posterior mean agrees with with Singh et al's kernel estimator, thus ensuring our model characterizes uncertainty around a SOTA estimator.
> - **Calibration as Compensation:**  As $\phi(V)$ is generally not known in closed form, any posterior approximation error induced by eq(11) is difficult to quantify in full generality. Nonetheless, our calibration procedure precisely aims to offset error from model mis-specification and posterior approximation (see below for details).
> - **Empirical Performance:** The assumption eq(11) has not been to the detriment of our method, as we generally achieved SOTA calibration and predictive accuracy in all experiments (including the new  ablations performed for other reviewers).
>
>
> **(II) Variance Non-negativity:** Our MC approximation for $\tilde K_V$ guarantees it is positive semi-definite. In particular, since
> $$\tilde k_V(v,v') = \int k_V(v,t)k_V(v',t) d\nu(t)$$
> our finite sample approximation is
> $$\widehat{\tilde k_V}(v,v') = \sum_{s=1}^S k_V(v,v_s)k_V(v',v_s) \quad , \quad (v_s)_{s=1}^S \sim \nu$$
> which is PSD. Since $S_2$ is quadratic in $\tilde K_V$, this ensures $S_1+S_2+S_3 \geq 0$. We will clarify this in the revision.
>
> **(III) Uncertainty Interpretation and Guarantees**
> - **Interpreting Method:** We view our method as having properties of both perspectives you put forward: it is a closed-form approximate bayesian inference procedure, yet with the calibration algorithm directly targeting good frequentist performance. For pure Bayesians, the main contribution is developing a GP-based method for continuous treatments that can avoid the pathologies of earlier GP-on-RKHS methods (BayesIMP). For frequentists, the main contribution is closed-form uncertainty estimates around the estimator in Singh et al. (2024) with good calibration performance.
> - **Calibration guarantees:** We do not view the calibration procedure as purely heuristic. Theorem B.9 in Appendix B shows that our estimator of the calibration loss used to select the calibration hyperparameter is a consistent estimator under the stated assumptions, whenever the bootstrap estimator of the distribution and posterior mean is consistent. Under standard regularity assumptions, this transfers to consistency of the selected calibration hyperparameter to the argmin.
> In this case, our method asymptotically selects the best-calibrated posterior within the chosen spectral family. If the spectral family induces sufficient variation in the posterior variance and the limiting asymptotic distribution of $\sqrt{n}(\hat \gamma - \gamma)$ is Gaussian pointwise, this implies frequentist posterior consistency (although we cannot guarantee this is always true).
>
> We initially omitted these details from the main text for readability, but we will revise the paper to make more explicit reference to the Appendix results.
>
> **(IV) Computational Scalability**. We appreciate the paper could do with more evidence of the scalability of our proposed method.
>
> 1. **Runtime/Memory in Experiments**: Below we report wall-clock runtimes and memory footprint for the experiment in Section 7.1, for $n \in \{100,1000\}$, split by training, calibration, and inference. All results are averaged over 50 trials and were run on either a NVIDIA Quadro P5000 or RTX A4500 GPU, using minibatches of size \(b=512\) for kernel hyperparameter training in the $n=1000$ case. Runtime is $<30$secs total and GPU ram usage $<2$GB at $n=1000$.
>
> |Sample size|train(sec)|cal(sec)|inf(sec)|peak_gpu_ram(GB)|
> |---|---:|---:|---:|---:|
> |n=100|8.86|9.03|0.02|0.20|
> |n=1000|10.53|14.03|0.04|1.60|
>
> 2. **Large-scale Real Dataset**: We have additionally run our method on a real dataset of size $n\approx 10^4$ and report runtime and memory results in point (III) of the response to **reviewer xLo4**. Even at that size our method runs in $<40$secs and uses $<14$GB GPU memory with exact kernel computations at calibration and inference time.
>
> 3. **Approach to $n \gg 10^4$**: For much larger datasets, one could use standard kernel approximations such as Random Fourier Features or Nystrom's method, which, when using standard matrix identities (e.g., Woodbury) would reduce time and storage complexity to $\mathcal O(n)$. While this may affect raw calibration performance, we do not expect these approximations to negatively affect our calibration optimization procedure.
>
>
> If you find our clarifications and additional experiments satisfactory, we would be grateful if you might consider updating your score.

---

> > ### Author Rebuttal · Reviewer_NN4N · 2026-04-04
> >
> > Thank you for addressing my concerns. Most of my concern has been solved.
> >
> > In Question 1, I still have some unclear points. Can you provide further discussion on how this approximation may affect the posterior uncertainty in a more concrete way?

---

> > > ### Author Response · Authors · 2026-04-04
> > >
> > > Thank you for the follow-up - we are glad our previous response addressed most of your concerns! We have now analyzed your remaining question in detail and can provide a more concrete answer.
> > >
> > > **How Eq. (11) may not be exact**
> > > Recall Eq. (11) is a surrogate likelihood model for the spectral coordinates,
> > > $$
> > > \phi_{V,i}(V)=\mu_i(Z)+\xi_i,\qquad \xi_i\sim N(0,\eta^2),
> > > $$
> > > introduced so that posterior moments are tractable and the posterior mean agrees with the estimator of Singh et al. (2024). This model may not be exact if:
> > >
> > > 1. The true noise is non-Gaussian, or
> > >
> > > 2. $\phi_V(V)$ is not well represented by an additive-noise model (ANM).
> > >
> > > Below we examine the effect this modeling assumption can have on posterior uncertainty by studying how deviation from a well-specified likelihood can propagate to the posterior.
> > >
> > > **Finite-Sample Analysis of Posterior Approximation Error**
> > >
> > > In what follows, we summarize some simple results we derived in the pdf at https://anonymous.4open.science/r/Submission16139. This analysis is for a fixed dataset $D = (V_i,Z_i,W_i,Y_i)_{i=1}^n$ and is therefore non-asymptotic.
> > >
> > > **Setup**: Let $\hat L(f,\mu) = \hat p(D|f,\mu)$ denote the data likelihood implied by Eq.(11) and the models in the main text. Let $L(f,\mu) = p(D|f,\mu)$ denote a *well-specified*, but possibly intractable likelihood (i.e., there exists a latent pair $(f^\star,\mu^\star)$ such that the true conditional distribution of the observed data is exactly $L(\cdot\mid f^\star,\mu^\star)$). Let $P$ and $\hat P$ denote the induced posteriors on $(f,\mu|D)$ under $L$ and $\hat L$ respectively, and $P_{w,z}$ and $\hat P_{w,z}$ the corresponding posteriors of $\gamma(w,z)|D$, via the representation in Eq.(6) and Eq.(9).
> > >
> > > **Bounds**: If the normalized second moment of the likelihood ratio is bounded as
> > > $$\frac{\mathbb E_{P}[R(f,\mu)^2]}{\mathbb E_P[R(f,\mu)]^2}\le 1+\varepsilon,\quad R(f,\mu) :=\hat L(f,\mu)/L(f,\mu)$$
> > > then one can show using standard inequalities that the posterior approximation error satisfies the $\xi^2$-divergence bound
> > > $$\chi^2(\hat P_{w,z}|P_{w,z})\leq\varepsilon$$
> > >
> > > Under suitable moment assumptions, this control propagates to the posterior moments used to form our Gaussian credible intervals. For example, if $\gamma_{w,z}\in L^4(P_{w,z})$, then
> > > $$\Big|\mathbb E_{\hat P_{w,z}}[\gamma_{w,z}]-\mathbb E_{P_{w,z}}[\gamma_{w,z}]\Big|\le\sqrt{\varepsilon}\sqrt{Var_{P_{w,z}}(\gamma_{w,z})}$$
> > > and we get a similar bound of order $O(\varepsilon+\sqrt \varepsilon)$ for the variance (see eq(3) in pdf).
> > >
> > > **Example.** We give two examples where we expect the surrogate model may be a good approximation, under the ANM setting
> > > $$V = h(Z) + \xi.$$
> > > 1. **Exact case.** Under a linear-kernel $k_V$, one may take spectral coordinates of the form
> > > $\phi_{V,i}(v)=u_i^\top v$ for orthonormal directions $\{u_i\}$. If $\xi\sim N(0,\eta^2 I)$, then
> > > $$
> > > \phi_{V,i}(V)=u_i^\top h(Z)+u_i^\top \xi,
> > > $$
> > > and the noise terms $u_i^\top \xi$ are independent $N(0,\eta^2)$. Thus Eq. (11) is exactly specified for $\mu_i(z) = u_i^\top h(Z)$, so one may take $L=\hat L$.
> > > 2. **Approximate case.** Suppose the spectral coordinates $\phi_{V,i}$ are non-linear but smooth. Then by a first-order Taylor approximation,
> > > $$\phi_{V,i}(V) = \phi_{V,i}(h(Z)) + \nabla \phi_{V,i}(h(Z))^\top \xi + O(||\xi||^2)$$
> > > If $||\xi||$ is small enough with high probability, $\phi_{V,i}(V)$ is well approximated by an ANM. If the noise is close to a Gaussian, then Eq.(11) is a close approximation (for an appropriate noise scale $\eta$), and so $\hat L$ may be close to some well-specified $L$ on average.
> > >
> > > **Why approximation error does not preclude good calibration:** We note that even when Eq.(11) is only an approximate working model, this does not by itself prevent good frequentist calibration:
> > > 1. The posterior mean is anchored to the estimator of Singh et al. (2024), which is consistent under general conditions that do not require exactness of the surrogate likelihood.
> > > 2. Our calibration step is designed to reduce coverage error in the uncertainty estimates, induced by the model or posterior approximation.
> > >
> > >
> > > **Empirical Evidence**
> > >
> > > We ran an ablation of the experiment in Section 6.1 under heavy-tailed Student-$t_{\nu=3}$ noise. Even under Gaussian noise, Eq.(11) is not exact there under a nonlinear kernel, but we expect heavy tails to worsen approximation error. The results below show slightly increased RMSE and calibration error relative to Gaussian noise (see Table 1 main text), but our method remains best-calibrated, and our calibration algorithm reduced coverage error to within $0.1$.
> > >
> > > |Method|RMSE|Cal.Error|
> > > |---|---:|---:|
> > > |Sampling-GP|0.37±0.01|0.16±0.09|
> > > |BayesIMP|0.58±0.01|0.35±0.17|
> > > |IMPspec-nocal|0.35±0.02|0.15±0.09|
> > > |IMPspec-cal|0.35±0.02|0.10±0.04|
> > >
> > >
> > > **Thank you** again for your thorough review of the paper - your comments have helped us strengthen the manuscript, and we will make sure to add the above discussion to the Appendix and reference it in the main text.

---

### Decision · Program_Chairs · 2026-04-30

**Decision:**

Accept (regular)

**Comment:**

Uncertainty quantification in causal inference for continuous treatment is still an open problem. The reviewers appreciate the relevance, and commend on the rigor. Overall, the reviewers feel that this submission is technically strong and timely. For completeness, there are also conformal methods for continuous treatments (https://arxiv.org/abs/2407.03094).

 The proposed method is novel, well motivated, and supported by empirical results. The scalability issue is often common in GPs, and not necessarily negative as it comes with theoretical properties and mathematical elegance. Overall, I believe this paper is of interest to the Causal ML community and a clear candidate to be accepted. congrats!